# A brainstem map for visceral sensations

Chen Ran[1], Jack C. Boettcher[1], Judith A. Kaye[1], Catherine E. Gallori[1] & Stephen D. Liberles[1✉]

The nervous system uses various coding strategies to process sensory inputs. For example, the olfactory system uses large receptor repertoires and is wired to recognize diverse odours, whereas the visual system provides high acuity of object position, form and movement[1–5]. Compared to external sensory systems, principles that underlie sensory processing by the interoceptive nervous system remain poorly defined. Here we developed a two-photon calcium imaging preparation to understand internal organ representations in the nucleus of the solitary tract (NTS), a sensory gateway in the brainstem that receives vagal and other inputs from the body. Focusing on gut and upper airway stimuli, we observed that individual NTS neurons are tuned to detect signals from particular organs and are topographically organized on the basis of body position. Moreover, some mechanosensory and chemosensory inputs from the same organ converge centrally. Sensory inputs engage specific NTS domains with defined locations, each containing heterogeneous cell types. Spatial representations of different organs are further sharpened in the NTS beyond what is achieved by vagal axon sorting alone, as blockade of brainstem inhibition broadens neural tuning and disorganizes visceral representations. These findings reveal basic organizational features used by the brain to process interoceptive inputs.

Sensory circuits transform basic physical inputs—photons of light, sound waves, chemicals and mechanical forces—into complex stimulus representations and perceptions. Landmark discoveries have provided insights into how neuronal circuits achieve these transformations for our external sensory systems. Examples include the olfactory receptor–guided map in the olfactory bulb[1,2], the cortical homunculus in the somatosensory system[6] and visual system maps that extract increasingly complex stimulus features as information ascends[3–5]. By contrast, less is understood about how interoceptive signals are processed.

The brain receives vital sensory information from internal organs within the body and uses this information to orchestrate crucial autonomic functions such as breathing, heart rate, blood pressure and gut motility, to ensure airway integrity and to modulate feeding, drinking and nausea behaviours[7–15]. Major respiratory, cardiovascular and digestive signals are primarily transmitted to the brain by the vagus nerve, which contains dozens of spatially intermingled sensory neuron types in sensory ganglia[11,12,14–16]. For example, sensory neurons in the gut detect chemicals and stretch to inform on the quality and quantity of ingested food, provide signals of nutrient reward, orchestrate systemic metabolism and contribute to the feeling of satiety after a meal[9,10,13,15,17]. Larynx-innervating vagal sensory neurons detect similar chemical and mechanical cues and initiate protective reflexes that guard the airways against aspiration[12,18]. Identical stimuli applied to the larynx or gastrointestinal tract induce distinct physiological and behavioural responses. This suggests that the location of a stimulus within the body is a key feature that must be decoded by downstream neural circuits.

Vagal sensory axons cross the skull and primarily target the NTS, a large sensory hub in the brainstem for interoceptive and gustatory information[19,20]. The central axons of vagal and other cranial afferents display some topography in their NTS projections, as visualized by genetic techniques or tissue-targeted dye injection[7–9,14,15,21,22]. For example, gustatory information is processed rostrally, whereas interoceptive information is processed caudally[19]. However, tracing of vagal axon tracts does not reveal response properties and input transformations that may occur in NTS neurons, which have elaborate dendritic arbours and potentially contact sensory axons from a distance[20,23,24]. Other classical approaches such as in vivo electrophysiology and cFos immunohistochemistry have provided important insights into NTS responses[20,23–27]. However, owing to technical limitations, these studies have produced conflicting conclusions about the organization of NTS neurons that are responsive to different interoceptive cues[20,24,28]. Here we focus on classical, well-defined sensory inputs from the gastrointestinal tract and upper airways to reveal basic features of visceral sensory coding.

## Neuronal tuning to internal stimuli

We developed an in vivo two-photon calcium imaging preparation in the mouse to enable a massively parallel, spatially resolved and real-time analysis of the responses of individual NTS neurons to multiple internal organ stimuli. We used a nuclear-localized, red-shifted calcium indicator (jRGECO1a fused with histone H2B)[29,30], administered through an adeno-associated virus (AAV) with a promoter that drives constitutive expression in neurons. Nuclear-localized calcium indicators are widely used, remove background signal from neuropil and display response amplitudes and kinetics compatible with measuring interoceptive responses[29,30] (Extended Data Fig. 1a–f). Brain motion artefacts precluded imaging in an awake mouse, but were negligible under anaesthesia after implanting a cranial window (Supplementary Video 1). The dorsal surface of the brainstem was exposed after surgical removal of part of the cerebellum, and two-photon microscopy was performed over a large caudal NTS area and depth (509 × 509 × 320 μm).

[1]Department of Cell Biology, Howard Hughes Medical Institute, Harvard Medical School, Boston, MA, USA. ✉e-mail: Stephen_Liberles@hms.harvard.edu

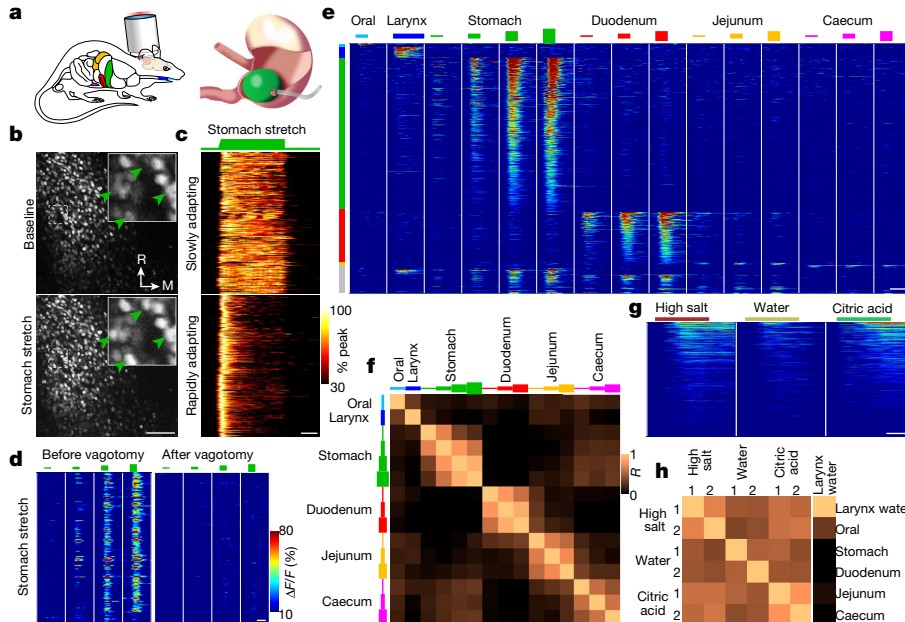

**Fig. 1 | Tuning of NTS neurons to gut and upper airway stimuli. a**, Cartoons depicting NTS two-photon calcium imaging (left) and gastric balloon distension (right). **b**, Representative two-photon NTS images of H2B-jRGECO1a fluorescence at baseline and during stomach stretch, transverse plane. Green arrowheads indicate responsive neurons. M, medial; R, rostral. **c**, Heatmaps depicting all observed NTS neuron responses (peak $\Delta F/F$ normalized) to a 40-s stomach stretch (600 µl). *k*-means clustering defined slowly adapting neurons (top: 548) or rapidly adapting neurons (bottom: 696) from 19 mice. **d**, Heatmap depicting NTS responses (188 neurons, 2 mice) to gastric distension (150, 300, 600 and 900 µl) before and after subdiaphragmatic vagotomy. **e**, Heatmap depicting all observed NTS neurons responses (1,133 neurons, 6 mice) to a stimulus series of oral balloon distension (200 µl), laryngeal water perfusion,

stomach distension (150, 300, 600 and 900 µl), duodenum distension (90, 115 and 140 µl), jejunum distension (90, 115 and 140 µl) and caecum distension (100, 250 and 350 µl). **f**, Correlation coefficient matrix of maximal $\Delta F/F$ for each stimulus pair across all neurons responsive to any stimulus in **e**. **g**, Heatmap depicting all observed NTS neuron responses (484 neurons, 4 mice) to successive laryngeal perfusion of high salt (10× PBS), water or citric acid (25 mM, pH 2.6). Responses are of two trials averaged. **h**, Correlation coefficients ($R$) of maximal $\Delta F/F$ for each stimulus pair across all responsive neurons in **g** (left, 6 × 6 box) or for stimulus pairs from **e** that included laryngeal water and the highest stretch stimulus in each organ across all neurons responsive to those stimuli (right, 1 × 6 column). Scale bar, 100 µm (**b**). White bars, 10 s (**c**–**e**,**g**).

This platform enabled the simultaneous recording of about 2,800 neurons and three-dimensional mapping of organ representations (Supplementary Video 2).

We first validated NTS calcium imaging by observing responses to gastric distension (Fig. 1a–d), a classical interoceptive stimulus that activates NTS neurons[8,13]. Mechanical distension of the glandular stomach through a surgically implanted balloon induced calcium transients in NTS neurons. Responses were highly repeatable and occurred at physiological levels comparable to the volume of a mouse meal (about 300 µl) (Fig. 1d and Extended Data Fig. 2a,b). Responses were dose-dependent (Fig. 1d), and enhancing gastric distension increased both the number of responding neurons and the response intensity in individual neurons. *K*-means clustering revealed two response types that could be assigned as either rapidly or slowly adapting, and response kinetics for individual neurons were consistent across trials (Fig. 1c and Extended Data Fig. 2). All gastric distension responses observed in the caudal NTS were abolished after bilateral subdiaphragmatic vagotomy (Fig. 1d), which indicates that the observed responses are transmitted by vagal sensory neurons.

Next, we investigated the tuning properties of individual NTS neurons by examining responses to multiple interoceptive stimuli. The visceral NTS receives diverse mechanical, chemical, osmotic and thermal stimuli from numerous body locations. Therefore, we initially varied the location of the stimulus while keeping the modality of the stimulus constant when possible. The position, timing and magnitude of mechanosensory stimuli can be precisely controlled and repeatedly applied. Therefore, we activated mechanosensory neurons in the oral cavity, the stomach, the duodenum, the jejunum and the caecum in the same mouse through localized tissue distension. Distension of

the larynx was not achieved owing to its small size in the mouse, so laryngeal neurons were instead activated through the perfusion of water, a distinct sensory modality that has been well characterized to activate vagal afferents[12,18,27]. Responses were observed for each stimulus in the caudal NTS, with more neurons detecting stomach distension, duodenum distension or laryngeal water (approximately 62%, 20% and 4% of all responding neurons, respectively). Fewer neurons responded to distension of the oral cavity, the jejunum or the caecum (about 1% each; Fig. 1e and Extended Data Figs. 3a–e and 4). Notably, the majority of responsive neurons (89.8%, 1,018 out of 1,133 neurons, 6 mice) were selectively stimulated by signals emanating from a specific organ (Fig. 1e and Extended Data Fig. 3a–e). Neurons that responded to two or more organ inputs were reliably observed across experiments, but were rare and were the most common dually tuned neurons responding to both stomach and duodenum distension (Extended Data Fig. 3d). Moreover, even neurons that showed significant responses to two or more organ inputs usually responded more strongly to one of them (Extended Data Fig. 3f,g). Neuronal responses to stimuli of different intensity in the same organ were more correlated ($R = 0.60$) than responses to stimuli in different organs ($R = 0.10$) (Fig. 1f).

Single-cell transcriptomics studies revealed that the NTS contains diverse cell types, including several classes of excitatory and inhibitory neurons[31]. Inhibitory neurons in other sensory systems are often broadly tuned[32,33]. Therefore, to distinguish the contributions of NTS inhibitory and excitatory neurons, calcium transients were measured in *Slc32a1-ires-cre* (also known as *Vgat-ires-cre*) mice that also contain a Cre-dependent *Gfp* allele. Inhibitory neurons were engaged by each internal organ stimulus tested and displayed markedly similar tuning

properties to GFP-negative neurons or the global NTS population (Extended Data Fig. 5). Thus, sensory inputs from different organs activate largely discrete ensembles of both excitatory and inhibitory NTS neurons.

We also obtained the following collection of nine Cre mice that label different subpopulations of NTS neurons and/or neuron types that have been previously linked to particular physiological functions: (1) *Th-cre*, (2) *Cartpt-ires-cre*, (3) *Calcr-ires-cre*, (4) *Pdyn-ires-cre*, (5) *Tac1-ires-cre*, (6) *Penk-ires-cre*, (7) *Gcg-cre*, (8) *Sst-ires-cre* and (9) *Crhr2-ires-cre*. Stomach stretch and intestine stretch activated NTS neurons labelled in each Cre line (Extended Data Fig. 6), although quantitative differences were noted in a few cases. In particular, neurons labelled in *Crhr2-ires-cre* mice were enriched medially and responded more frequently to duodenum stretch. By contrast, neurons labelled in *Th-cre* mice were enriched more laterally and responded less frequently to duodenum stretch. Taken together, these results show that different visceral inputs can engage a heterogeneous assortment of cell types, and each of these Cre-defined cell types is recruited across different sensory representations.

Next, we asked how the NTS represents different sensory modalities from the same organ. We focused on the larynx, which enabled comparisons with previous in vivo electrophysiological studies[18,27], and the intestine. Larynx-innervating vagal sensory neurons detect various chemical challenges, including water, salt and acid, and initiate protective reflexes that guard the airways against aspiration[12]. During an NTS imaging session, a cannula was implanted below the vocal folds, and saline solution was perfused through the larynx at a slow rate (500 µl min$^{-1}$), which did not trigger background mechanical responses. Robust and repeatable NTS neuron responses were observed when the stimulus solution was switched to water, high salt (10× PBS) or citric acid (25 mM, pH 2.6) without altering the flow rate. Although rare NTS neurons displayed selective responses to specific laryngeal stimuli (Extended Data Fig. 3h), the majority were engaged by all tested stimuli (Fig. 1g,h and Extended Data Fig. 3h,i). Neuronal responses were better correlated between repeated applications of the same ($R = 0.62$) or different ($R = 0.47$) laryngeal stimuli than between laryngeal water and stimuli in other organs ($R = 0.07$). NTS neurons are more broadly tuned to laryngeal stimuli than vagal sensory neurons[12,18], which indicates a convergence of sensory information as signals ascend into the brain. Integration of laryngeal signals may provide a circuit-based explanation for how different airway threats can induce a common motor programme of airway defence.

Next, we compared NTS representations of intestinal stretch and intestinal nutrients, two stimuli that induce the sensation of satiety[9,10,13,15,17,34]. Acute glucose application into the proximal duodenum (300 mM in Hank's balanced salt solution (HBSS), 100 µl, exit port at 1.5 cm distally) consistently activated NTS neurons across animals (Extended Data Fig. 4c). By contrast, HBSS alone did not, which suggests that this small bolus was insufficient to trigger mechanosensory neurons. We note that intestinal glucose may activate both a dedicated response pathway and other non-mechanosensory neurons that are more broadly tuned, for example, to osmotic stimuli[8,10,34]. In the NTS, we observed some convergence of responses to intestinal glucose and intestinal stretch ; 42.2% of glucose-responsive NTS neurons were also activated by duodenum distension, and 31.0% of duodenum distension-responsive neurons were also activated by glucose (Extended Data Fig. 4d). We also noted that a smaller group of glucose-responsive NTS neurons detected stomach distension. However, glucose-responsive neurons were 6.8-fold more likely to respond to duodenal stretch than stomach stretch compared with NTS neurons at large. This result indicates that signal convergence is not random, but instead displays some organ selectivity. Mechanical and chemical signals in the duodenum activate different vagal sensory neurons but overlapping NTS neurons[8,14,34], which indicates high-level convergence that arises in the brainstem.

Such partial signal convergence may reflect an economical design whereby polymodal neurons mediate common physiological and behavioural effects induced by both stimuli, whereas more selective NTS neurons preserve modality discrimination. Together, our data indicate that the NTS contains largely discrete neuronal populations that encode stimuli from distinct organs and in some cases at least partially overlapping ensembles that are engaged by multiple stimuli from the same organs.

## A spatial map of the viscera in the NTS

In addition to revealing neuronal tuning properties, in vivo calcium imaging provides precise spatial information for thousands of responding neurons across a large NTS region in a single experiment. In vagal sensory ganglia, neurons that detect inputs from different organs are intermingled in a salt and pepper manner[8,14]. By contrast, we observed here a remarkable spatial order that arises in the NTS, with nearby neurons displaying similar response properties. Stomach stretch-responsive neurons were clustered in an NTS region that was bordered laterally by the dorsal column nuclei, which responded to cutaneous stimuli that did not activate the NTS (Extended Data Fig. 7a), and along the anterior–posterior axis were most prevalent near the anterior boundary of the area postrema (Fig. 2a).

The positions of stomach stretch-responsive NTS neurons provided an anatomical landmark for comparing the locations of neurons responsive to other stimuli. Neurons responsive to duodenum stretch were situated more medially near the border of the NTS and area postrema (Fig. 2a). Analysis of the spatial distribution of all neurons that were selectively responsive to stomach and duodenum stretch revealed an approximately 60 µm centroid segregation per a plane of view, and segregation became more pronounced in deeper, more ventral NTS regions (Extended Data Fig. 7c–e). The relative positions of neurons responsive to stomach and duodenum stretch were stereotyped, easily located across animals and conserved across anaesthetics (Fig. 2a and Extended Data Fig. 1g–i). As is commonly observed in other brain maps[35], we observed mesoscale order and some microscale heterogeneity as stomach-recipient and intestine-recipient neurons were intermingled at the border between domains. A similar analysis revealed that neurons activated by stimuli from the oral cavity, the larynx, the jejunum and the caecum were also located apart from neurons responsive to stomach stretch (Fig. 2b and Extended Data Fig. 7b,f). Generally, organ representations in the NTS reflected their physical positions within the body, with more anterior organs represented in the more rostrolateral area of the NTS (Fig. 2b,c). Thus, visceral inputs from the gastrointestinal tract and upper airways are organized into a brainstem map that takes a shape analogous to a homunculus.

In contrast to the spatial separation between organ representations, multiple inputs from a single organ were observed to converge. Neurons that selectively responded to intestinal glucose and duodenum stretch were intermingled in the same domain and generally separated from neurons responsive to stomach stretch (Fig. 2d,e). Furthermore, NTS neurons responsive to stretch of more remote intestinal regions in the jejunum showed low spatial separation from neurons responsive to duodenal stretch, even though jejunum stimulus applications were about 40 and 90 mm apart from duodenum stimuli (Fig. 2f–h). This is in sharp contrast to the more segregated representations of duodenum and stomach stretch, which were only about 3 mm apart. Similarly, neurons responsive to distension of different stomach regions that were about 3 mm apart also showed low spatial separation (Extended Data Fig. 7g–i). Thus, stimuli from different organs can be segregated while different inputs from the same organ can converge. Mechanosensory signals from the gastrointestinal tract are represented more prominently by organ than absolute position in space, which suggests that the NTS uses a discrete map rather than a continuous map for at least some visceral stimuli[36].

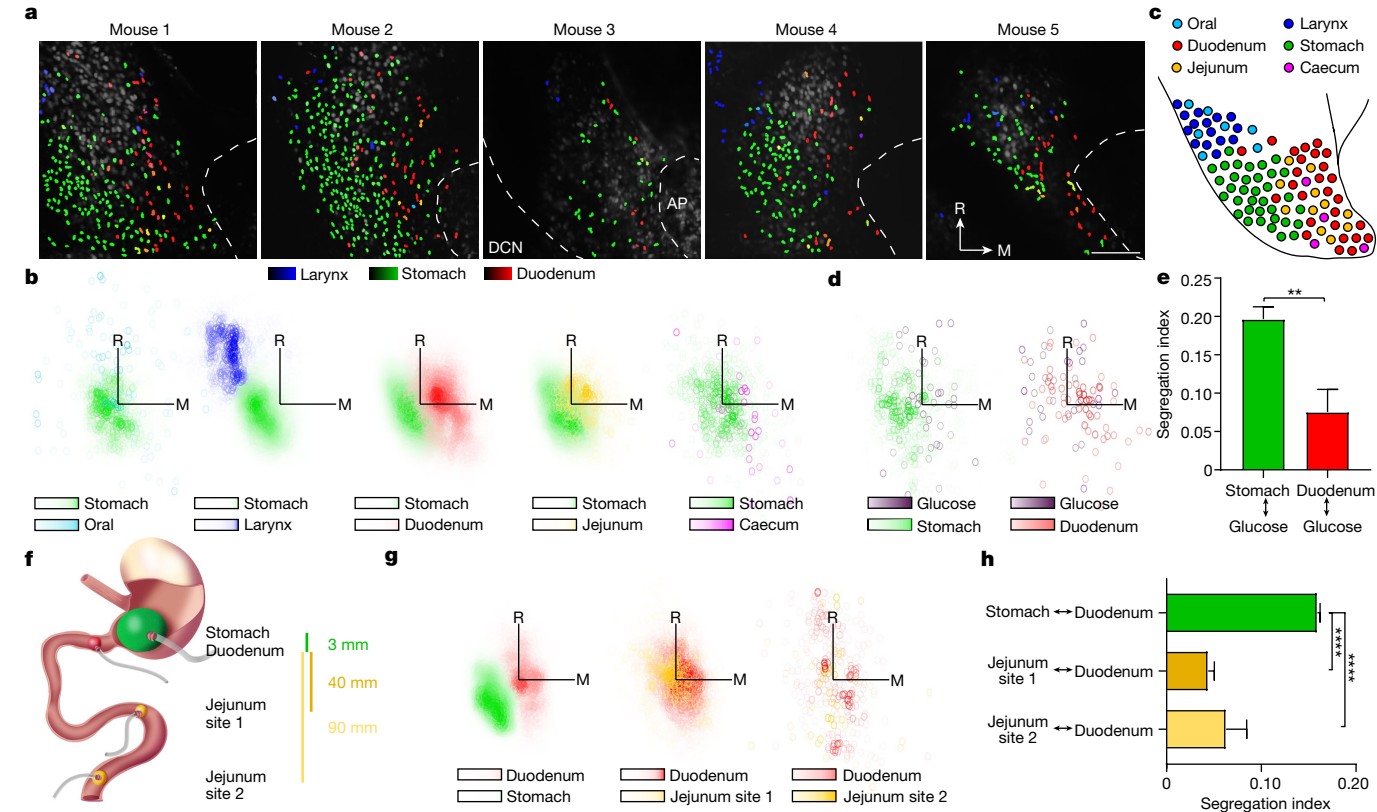

**Fig. 2 | A sensory map within the NTS. a**, Representative two-photon NTS images of H2B-jRGECO1a fluorescence (transverse view, images are similarly oriented), with neurons colour-coded based on maximal responses to laryngeal water (100 µl), stomach stretch (150, 300, 600 and 900 µl) or duodenal stretch (90, 115 and 140 µl). DCN, dorsal column nuclei; AP, area postrema. Scale bar, 100 µm (**a**). **b**, Neuron positions are charted by response type. Axis origin: centroid for stomach stretch-responsive neurons. The following number of neurons and mice were analysed: oral–stomach: 1,204 neurons, 11 mice; larynx–stomach: 10,610 neurons, 57 mice; duodenum–stomach: 28,556 neurons, 107 mice; jejunum–stomach: 13,050 neurons, 66 mice; caecum–stomach: 415 neurons, 9 mice. Colour scale depicts neuron density (Methods). Axis length, 150 µm. M, medial; R, rostral. **c**, Schematic depicting the relative location of different organ response domains in the NTS. **d**, Neuron positions are charted by response type. Axis origin: centroid of

neurons responsive to duodenal glucose (300 mM, HBSS). $n = 425$ neurons (left) or 97 neurons (right), 2 mice. **e**, Quantification of the segregation of neurons in **d** responsive to duodenum glucose from neurons responsive to duodenum or stomach stretch. Mean ± s.e.m., **$*P < 0.0001$, two-tailed Mann–Whitney test. See Methods for segregation index calculation. **f**, Cartoon depicting sites of balloon distension in the gastrointestinal tract. **g**, Neuron positions are charted by response type and axis origin: centroid of duodenum stretch-responsive neurons. The following number of neurons and mice were analysed: left: 14,981 neurons, 71 mice; middle: 2,846 neurons, 53 mice; right: 280 neurons, 7 mice. Axis length, 150 µm. **h**, Quantification of the segregation of neurons in **g** responsive to duodenum stretch from neurons responsive to other stimuli. Mean ± s.e.m., ****$P < 0.0001$, Dunn's multiple comparisons test following Kruskal–Wallis test of significance.

## Inhibition shapes NTS representations

We next sought to identify mechanisms that underlie the spatial segregation of visceral representations in the NTS. First, we compared the locations of vagal sensory axons from an organ with the NTS domain responsive to that organ. NTS calcium imaging was performed on mice in which the presynaptic terminals of vagal axons were visualized by injecting *AAV1-Cag-Flex-synaptophysin-Gfp* into the stomach, the intestine or the larynx of *Slc17a6-ires-cre* mice (also known as *Vglut2-ires-cre* mice), which enabled genetic access to >99% of vagal sensory neurons[7]. The central terminals of vagal axons from each organ displayed some organization in the NTS, but NTS neurons responsive to stimuli from that organ could not be predicted solely by the position of the corresponding axon terminals (Extended Data Fig. 8a–g). Vagal sensory neurons in the stomach, the larynx and the intestine include various mechanoreceptors and chemoreceptors with different NTS targeting patterns, so we imaged NTS responses while simultaneously visualizing axons of genetically defined vagal sensory neuron subtypes. In particular,

GLP1R and GPR65 label discrete populations of vagal sensory neurons that (1) predominantly function as gut mechanoreceptors and chemoreceptors, respectively, and (2) display spatially discrete NTS projections[8]. We injected *Glp1r-ires-cre* or *Gpr65-ires-cre* mice with *AAV1-Cag-Flex-synaptophysin-Gfp* in the stomach and *AAV1-Syn-H2b-jRGECO1a* in the NTS. We observed that stomach stretch-responsive neurons were closer to axonal boutons from stomach GLP1R neurons than stomach GPR65 neurons (Extended Data Fig. 8h–k). However, the positions of vagal axons and responsive NTS soma were not perfectly aligned (Extended Data Fig. 8l). These findings raise the possibility that higher-order processing and dendrite organization in the NTS also contribute to input segregation.

Inhibitory neurons contribute to the sharpening of neuronal responses across brain regions[32,33], and inhibitory neurons are distributed across the entire NTS (Extended Data Fig. 5). To examine the role of inhibitory neurons in patterning visceral representations, we first activated inhibitory neurons in the context of NTS imaging. We injected the NTS of *Vgat-ires-cre;Rosa26-lsl-Gfp-L10a* mice with an AAV containing a Cre-dependent gene encoding a designer receptor (hM3Dq)

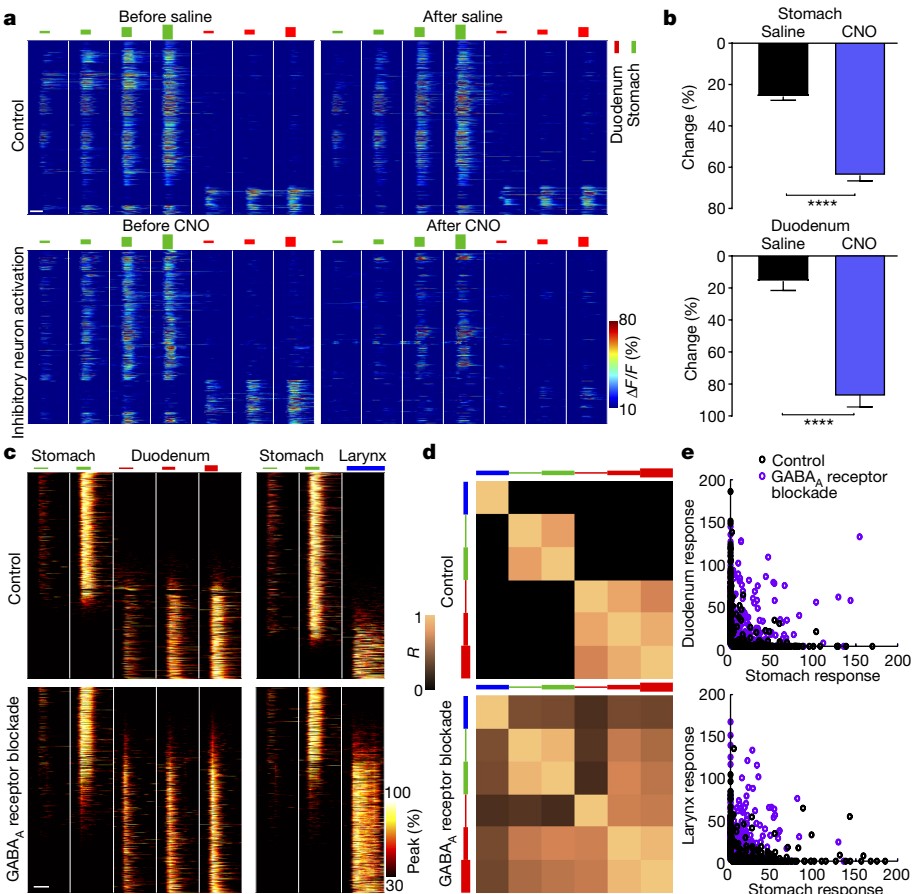

**Fig. 3 | Inhibition sharpens NTS responses. a**, *Vgat-ires-cre;Rosa26-lsl-Gfp-L10a* mice were injected in the NTS with *AAV8-Syn-DIO-HA-hM3dq-ires-mCitrine* and *AAV1-Syn-H2b-jRGECO1a*. NTS imaging was performed on GFP-negative excitatory neurons before and after CNO administration (bottom). Control mice (top) lacked expression of hM3Dq and were instead injected with saline. Heatmaps depict responses to stomach stretch (150, 300, 600 and 900 μl) and duodenum stretch (90, 115 and 140 μl). *n* = 869 neurons, 3 mice (top) or 673 neurons, 5 mice (bottom). **b**, Average change in peak Δ*F*/*F* of neurons from **a** after injection of saline or CNO. Samples sizes (left to right, top): 726 neurons, 3 mice; 507 neurons, 5 mice; (left to right, bottom): 94 neurons, 3 mice; 92 neurons, 5 mice. ****$P$ < 0.0001, two-tailed Mann–Whitney test. **c**, Heatmaps depicting all observed NTS responses without (top left: 16,222 neurons, 103 mice; top right: 7,919 neurons, 73 mice) or with (bottom left: 1,168 neurons, 5 mice; bottom right: 1,442 neurons, 5 mice) NTS-localized administration of the GABA_A receptor antagonist bicuculline. Responses were measured for gastric distension (150 and 300 μl), duodenum distension (90, 115 and 140 μl) and laryngeal water perfusion (100 μl), with the peak Δ*F*/*F* of each responding neuron normalized to 100%. Neurons in each heatmap are sorted by their response bias to stomach stretch (top) over duodenum stretch/laryngeal water (bottom). **d**, Correlation coefficient matrices of maximal Δ*F*/*F* for each stimulus pair across all responsive neurons from **c** (top: 11,795 neurons, 73 mice; bottom: 1,806 neurons, 5 mice). **e**, Responses (maximal Δ*F*/*F* above thresholds; Methods) of NTS neurons from **c**. *n* = 400 randomly depicted neurons per condition, 2 out 1,600 out of range. White bars, 10 s (**a**,**c**).

responsive to clozapine-*N*-oxide (CNO) and a second AAV containing a Cre-independent *H2b-jRGECO1a* allele. NTS responses were recorded in GFP-negative neurons to visceral stimulation before and after CNO injection in the same mouse. We observed that chemogenetic activation of NTS inhibitory neurons suppressed both the amplitude and number of neurons responsive to both stomach and duodenum stretch (Fig. 3a,b). This result is consistent with the notion that local inhibition contributes to vagal input gating[37].

We next blocked inhibition by imaging NTS neuron responses with or without NTS-localized injection of the GABA_A receptor antagonist bicuculline. Blockade of inhibition broadened NTS responses, causing many neurons to respond to both stomach and duodenum stretch or to both stomach stretch and laryngeal water (Fig. 3c–e). The mean correlation coefficient between all pairs of stimuli from different organs increased from −0.12 to 0.42 in bicuculline-treated animals. Response broadening was similarly observed in both excitatory and inhibitory neurons (Extended Data Fig. 9). As reported before[38], GABA_A receptor blockade in the NTS substantially reduced breathing rate; perhaps the engagement of breathing control circuits by ectopic sensory inputs contributes to altered breathing patterns.

To probe how inhibition is naturally engaged, we simultaneously stretched the stomach and duodenum and compared their responses to distending each organ alone. We observed that suppression was a prominent feature in many neuronal responses. Responses to duodenum distension were consistently suppressed in 29.0% of neurons if the stomach was simultaneously distended. Conversely, 17.4% of stomach-responsive neurons were inhibited by duodenum stretch (Fig. 4a). Response suppression was dose-dependent (Fig. 4b), with enhanced stomach stretch leading to stronger suppression of duodenum responses. Neurons prone to inhibition were intermingled with other neurons responsive to the same stimulus, without apparent positional segregation (Extended Data Fig. 10a). Cross-inhibition studies involving other organ pairs revealed that gastric distension similarly suppressed responses to jejunum distension but was less effective at suppressing laryngeal water responses, suggestive of at least some selectivity in cross-inhibition (Extended Data Fig. 10). These findings indicate that NTS neurons display a marked spatial patterning that is based on the organ they receive signals from. Spatial patterning is enhanced beyond what can be achieved by vagal axon sorting alone, and lateral inhibition is a key feature that contributes to the selective tuning of at least some NTS neurons.

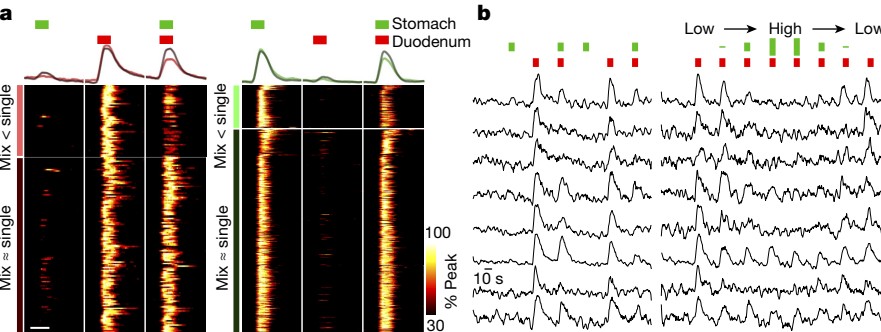

**Fig. 4 | NTS response suppression by paired inputs. a**, Calcium responses were recorded in NTS neurons during successive application of stomach stretch (600 µl), duodenal stretch (115 µl) and simultaneous stomach–duodenal stretch, with the stimulus series repeated and responses averaged. Neurons that selectively detected duodenal stretch (left, 176 neurons) or stomach stretch (right, 544 neurons) were classified into two groups that exhibited inhibition from the simultaneous stomach–duodenal stretch (mix < single; left: 51 neurons; right: 94 neurons) or did not (mix ≈ single;

left: 125 neurons; right: 450 neurons, 5 mice). Average traces (top) or heatmaps (bottom) of responding neurons in each class are separately depicted, with the peak Δ*F*/*F* of each responding neuron normalized to 100%. White bar, 10 s. **b**, Representative traces depicting normalized Δ*F*/*F* over time for individual neurons from **a** that showed responses to the six stimuli described (left) and another series of eight duodenal stretches with simultaneous application of 0, 300, 600, 900, 900, 600, 300 and 0 µl stomach stretch (right).

## Discussion

Neuronal maps are hallmark circuit motifs used by several sensory systems to represent different stimulus features[4–6,35,39–44]. Spatial grouping of similarly responding neurons presumably facilitates and economizes neural circuit wiring[45]. Some maps encode stimulus position, such as the somatosensory homunculus or whisker representations in the rodent barrel cortex[6,41]; other maps encode stimulus identity or quality, such as auditory maps that report on the tone and structure of sounds, or colour maps in the visual cortex[40,44]. Here, we revealed a striking topographical map in the brainstem for visceral inputs from the gastrointestinal tract and the larynx, with neuron location in the brainstem reflecting the site of sensation within the body. We also revealed a key role for inhibition in sharpening NTS responses and amplifying the contrast among nearby neurons that transmit distinct visceral signals. Inhibition in other sensory systems allows efficient stimulus contrast and top-down control[32]. Here, blocking inhibition broadens neuronal tuning, which indicates that many NTS neurons possess the latent capacity to process multiple streams of information.

The NTS map arises from distributed neuronal representations in vagal ganglia[8,14]. The genesis of spatial organization in the brainstem shares similarities with peripheral processing in the olfactory system, in which primary olfactory sensory neurons that express the same chemosensory receptor are distributed, and spatial order subsequently arises de novo in the olfactory bulb[1,2]. Furthermore, the responses of second-order olfactory neurons (mitral cells) are primarily defined by space, by the olfactory bulb glomerulus they innervate and the olfactory receptor they receive inputs from. Here we showed that location is also a key determining factor in NTS neuron responses. Many important studies have used genetic tools to delineate NTS cell types[10,46–55]. Here we report that each stimulus activates a heterogeneous assortment of NTS cell types, although the frequencies of recruited cell types can vary across stimuli and are particularly notable in two cell types with regionalized NTS locations. Therefore, we suggest that it will be additionally important to consider cell position as well as transcriptional identity in understanding the functions of NTS neurons. Our data indicate that the NTS contains spatially defined domains for particular sensory inputs, with each input-defined domain composed of diverse and commonly shared types of transcriptome-defined cells. Our findings are reminiscent of the somatosensory and motor cortices and the spinal cord, all of which contain diverse cell types horizontally distributed across columns that correspond to distinct somatotopic areas[56–58].

Across sensory systems, maps can change in complexity as information ascends through neural circuits. Neurons in the visual system extract increasingly complex stimulus features as information is transmitted from the retina to the visual cortex[3–5]. By contrast, the olfactory system forms elaborate receptor-guided maps in the olfactory bulb but discards spatial structure in the olfactory cortex[1,2,43]. Here we showed that the interoceptive sensory system also generates remarkable spatial organization in the brainstem. Understanding how maps are transformed and key coding features that arise or dissipate in each brain region is essential for determining the role of that brain region in circuit computations, and more generally, how the brain encodes information.

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

# Methods

## Animals

All animal procedures followed the ethical guidelines outlined in the NIH Guide for the Care and Use of Laboratory Animals, and all protocols were approved by the Institutional Animal Care and Use Committee at Harvard Medical School. Animals were maintained under constant temperature ($23 \pm 1\,°C$) and relative humidity ($46 \pm 5\%$) with a 12-h light–dark cycle. Mice used in this study were from both sexes and of a mixed genetic background. *Glp1r-ires-cre* (029283), *Gpr65-ires-cre* (029282) and *Crhr2-ires-cre* (033728) mice were previously made in the laboratory and are now available at the Jackson Laboratory. *Slc17a6-ires-cre* (*Vglut2-ires-cre*, 028863), *Slc32a1-ires-cre* (*Vgat-ires-cre*, 028862), *Rosa26-lsl-Gfp-L10a* (024750), *Tac1-ires-cre* (021877), *Pdyn-ires-cre* (027958), *Penk-ires-cre* (025112), *Cartpt-ires-cre* (028533), *Sst-ires-cre* (013044), *Th-cre* (021877) and *Gcg-cre* (030542) mice were purchased from the Jackson Laboratory.

## AAV generation

To construct the AAV for neuronal expression of nuclear-localized jRGECO1a (*AAV1-Syn-H2b-jRGECO1a*), the sequence encoding jRGECO1a was cloned from *pAAV-Syn-NES-jRGECO1a* (Addgene, 100854) and inserted into *pAAV-Syn-H2b-Gcamp6f* (Addgene, 74144) by replacing the sequence encoding GCaMP6f. In brief, the *jGRECO1a* gene was amplified by PCR (forward primer: ATAATAGGATCCACCAGTCGCCAC-CATGGGTTCTCATCATCATCATCATCATGGTATGG; reverse primer: ATAATAAAGCTTCTCGAGTGGATCATCTGCAGAATTCTCTAGATCGCGAAC-TAGTGATCCGGTCACTTCGCTGTCATCATTTGTACAAACTC). The *jGRECO1* PCR product and *pAAV-Syn-H2b-Gcamp6f* were cut (BamHI and HindIII) and ligated together. The resulting plasmid was sequenced for verification and used to prepare virus by the Boston Children's Hospital Viral Core Facility.

## Single-cell transcriptomics analysis

Published transcriptomics data[31] of single nuclei from the mouse dorsal vagal complex were analysed (Extended Data Fig. 6) using eight libraries with the greatest sequencing depth (31S, 20W, 34W, 14V, 36S, 40S, 39S and 25V), which corresponded to 19,481 neurons, including 13,630 NTS neurons. Neuronal clusters were assigned to the NTS or other caudal brainstem regions based on the expression patterns of signature genes revealed in RNA in situ hybridization data from the Allen Brain Atlas. Count matrices of unique transcripts for each library were normalized using regularized negative binomial regression with the function SCTransform in Seurat[59]. The transformed count matrices for each library were then merged, and principal component analysis, dimensional reduction and cluster identification were performed. Analysis and data visualization were conducted using Seurat (v.4.0.5) in R (v.4.1.1)[60,61].

## Stereotactic NTS injections

Young (2–5 week old) mice were anaesthetised (65 mg kg$^{-1}$ ketamine, 13 mg kg$^{-1}$ xylazine) and placed in a small animal stereotaxic frame (David Kopf Instruments). Direct NTS exposure caused minor but unavoidable tissue inflammation, so the NTS was accessed by oblique injection (25° from perpendicular, angled posteriorly). Three injections (unilateral) or five injections (bilateral) were made 0.36 mm, 0 mm and 0.12 mm lateral to the midline and 4.40 mm, 4.65 mm and 4.90 mm posterior to bregma, respectively. For young mice, bregma-to-lambda distances were measured, and injection coordinates along the rostral–caudal axis were scaled to correspond to coordinates for a bregma-to-lambda distance of 4.2 mm. A glass pipette was slowly lowered to a depth of 4.8–5.0 mm for the most rostral injection, and 4.3–4.8 mm for other sites. Virus solutions containing *AAV1-Syn-H2b-jRGECO1a* ($0.6–1.3 \times 10^{13}$ genomes per ml), *AAV8-Syn-DIO-HA-hM3dq-ires-mCitrine* (Addgene 50454-AAV8, $1.9 \times 10^{13}$ genomes per ml), *AAV1-Syn-Gcamp6m* (Penn Vector Core, CS1114, $7.0 \times 10^{12}$ genomes per ml) or *AAV9-Cag-Flex-Gfp* (Addgene 51502-AAV9, $3 \times 10^{12}$ genomes per ml) were injected (150 nl per injection, 2 nl s$^{-1}$) using a Nanoject III injector (Drummond). Mice recovered for 10–21 days before functional imaging. Control experiments were also performed in animals injected with AAVs at 5 weeks of age, and similar NTS responses were observed during in vivo imaging (Extended Data Fig. 1d–f).

## Organ injections

*AAV1-Cag-Flex-synaptophysin-Gfp* was packaged by the Baylor College of Medicine Intellectual and Developmental Disabilities Research Center Neuroconnectivity Core Facility and the Boston Children's Hospital Viral Core Facility. Serotype 1 AAVs retrogradely label other peripheral sensory neurons with high efficiency[62–64]. Young *Vglut2-ires-cre*, *Gpr65-ires-cre* or *Glp1r-ires-cre* mice were anaesthetised (65 mg kg$^{-1}$ ketamine, 13 mg kg$^{-1}$ xylazine). Stomach and duodenum were exposed through a small abdominal incision. For the stomach, approximately 40 injections (10 µl total) of *AAV1-Cag-Flex-synaptophysin-Gfp* were made into the glandular stomach wall using a Hamilton syringe. For the duodenum, approximately 5 injections (500 nl total, 5 nl s$^{-1}$) of *AAV 1-Cag-Flex-synaptophysin-Gfp* were made into the first 1–1.5 cm of the duodenum using a Nanoject III injector (Drummond). The larynx was exposed through a small midline incision on the ventral side of the neck, and the sternohyoid muscle was retracted. Approximately 5 injections (300 nl total, 5 nl s$^{-1}$) were made into the laryngeal wall between the anterior edge of the thyroid cartilage and the first ring below the cartilage. After injection, tissue was cleaned with a surgical sponge to absorb spilled fluid. Incisions were closed with sutures, and the mice recovered for at least 20 days before calcium imaging.

## Two-photon calcium imaging

Mice were anaesthetised with urethane (2 mg g$^{-1}$ over 2 intraperitoneal injections at least 20 min apart and surgery beginning at least 20 min after the second urethane injection) or with isoflurane inhalation (1.5–2.5% throughout the procedure, hours after initial intraperitoneal injection with 65 mg kg$^{-1}$ ketamine and 13 mg kg$^{-1}$ xylazine). Deep anaesthesia was ensured by assessing leg pinch and corneal reflexes during the experiment, with additional urethane (0.2–0.6 mg g$^{-1}$) given as needed. Mice were warmed on a custom-made heated platform, and a tracheotomy was performed to insert a breathing tube. Cranial surgery was performed with the head fixed at an approximately 30° downward pitch. The cervical paraspinal muscles were removed, exposing the skull and the first cervical vertebra. The skull on top of cerebellar lobules VI–IX was cut with a dental drill, and the posterior part of the cerebellum and approximately lobules VII–X were gently aspirated to expose the dorsal surface of the brainstem. After bleeding stopped, a custom-made titanium headpost was mounted on the skull parallel to the surface of the area postrema. The anterior portion of the post was affixed to the skull with adhesive cement (C&B Metabond; Parkell), and the posterior portion was sealed to the skin with adhesive. A small drop of KwikSil adhesive (World Precision Instruments) was applied to the surface of the brainstem, and a fan-shaped cranial window, cut from a round coverslip (Warner Instruments, size no. 0, 5 mm), was lowered to the surface of the brainstem. The angle of the cranial window was adjusted so that it was parallel to the surface of the area postrema, and pressure was applied using a micromanipulator (World Precision Instruments). Blood flow in the vein of the inferior cerebellar peduncle was carefully monitored, as a stable preparation free from motion artefacts requires that sufficient pressure is applied to the window to impede blood flow at this step. The cranial window was further secured to the skull and headpost with Metabond cement. The micromanipulator was removed after the cement hardened, as determined through a partial release in pressure on the brain such that blood flow in the vein of the inferior cerebellar peduncle was no longer obstructed. To increase the surgery success rate, mice were often given an intravenous injection of PBS (300 µl) early during the surgery, and/or given oxygen through a nose cone.

Two-photon calcium imaging was performed with an Olympus FVMPE resonant-scanning two-photon microscope equipped with a piezoelectric

objective Z-stepper (P-915, Physik Instrumente) and an Olympus ×25 water-immersion objective (NA of 1.0, WD of 8 mm). Olympus FluoView software was used to collect two-photon images. A Ti:sapphire laser with dispersion compensation (MaiTai eHP DeepSee, SpectraPhysics) was tuned to 975–1,020 nm, and fluorescence emission was filtered with a 570 nm long-pass dichroic and a 495–540 nm bandpass filter for the GFP and GCaMP6m signals and either a 575–645 nm or a 575–725 nm bandpass filter for the jRGECO1a signal. Volumetric imaging (1.25 Hz, resolution of 512 × 512 pixels) typically consisted of five focal planes 80 or 100 μm apart, covered a large NTS area (509.12 × 509.12 × 320 μm) and enabled the simultaneous recording of about 2,800 neurons per experiment. Neurons in Extended Data Fig. 1a–c were imaged on a single focal plane at 10–12.5 Hz. Laser power measured at the front aperture of the objective was 35–90 mW and depended on the quality of the cranial window and imaging depth. Mice were carefully positioned onto headpost clamps for parallel positioning of the cranial window and the front lens of the microscope objective. To ensure that the anatomical axes were in alignment with the microscope detectors, the yaw and roll dimensions were further finely adjusted using the area postrema as the landmark.

Mechanical organ distensions were achieved by inflation of a latex balloon (Braintree Scientific, 73-3479 for the stomach and 73-3478 for all other organs), as previously described[7]. A small rodent feeding needle (Cadence Science, 9920) was affixed inside the balloon and connected to a syringe by silicon tubing, which enabled precise volume control by liquid infusion. Distension of the oral cavity was achieved by inflating a balloon placed in the mouth against the cranial base. For gastric distension, the content of the stomach was removed, and a small incision was made in the greater curvature of the glandular stomach. The balloon was advanced into the antrum of the stomach and was secured by a suture around the incision site. In Extended Data Fig. 7g–i, anterior stomach distension involved inflation of a balloon implanted in the forestomach. Duodenum distension was achieved by placing the balloon into the duodenum bulb through an incision at the pyloric sphincter and was secured at the sphincter by a suture. To ensure that mechanical stimuli were isolated only to the duodenum or stomach, a near-complete circular incision was made around the pyloric sphincter and then a suture was tied around the sphincter, which also prevented spillage of stomach contents. Jejunum balloons were placed in the small intestine about 40 mm (around the ligament of Treitz) and 90 mm distal to the pyloric sphincter. Caecum balloons were placed through an incision at the caecum–colon junction and secured by a suture. Tubing affixed to all balloons was secured to the skin with adhesive. For Fig. 4a,b and Extended Data Fig. 10, a stomach stretch of 600 μl, a duodenum stretch of 115 μl, a jejunum stretch of 115 μl and a laryngeal water perfusion of 8 s were applied to mice before imaging.

Laryngeal perfusions were based on a previous protocol with modifications[12]. A tracheotomy was performed approximately five cartilaginous rings below the thyroid cartilage, and a cannula (PE 50 tubing, Braintree Scientific) was advanced through the incision site towards the thyroid gland. One additional cartilaginous ring between the perfusion cannula and the breathing tube was removed to reduce tension. Additional Kwik-Sil silicone adhesive was applied around the trachea, the cannula and the breathing tube to secure their positions, avoiding mechanical displacement during stimulation, and tissue glue was applied to affix the cannula to the skin. For stimulations in Fig. 1g,h and Extended Data Figs. 3h,i and 10, PBS was constantly and slowly perfused through the laryngeal cannula using a peristaltic pump. For stimulation, the perfusate was switched to either high salt (10× PBS), 25 mM citric acid (pH 2.6) or water for 24 s (Fig. 1g,h and Extended Data Fig. 3h,i) or for 8 s (Extended Data Fig. 10) without changing the flow rate. The intervals between stimuli were 96 s (Fig. 1g,h and Extended Data Fig. 3h,i) or 40 s (Extended Data Fig. 10), and each stimulus was repeated twice in the same mouse. For laryngeal stimulation in other figures, a 100 μl water bolus was delivered through a syringe connected to the laryngeal cannula with silicon tubing.

For application of chemicals in the duodenum, cannulas were inserted into the duodenum bulb through an incision at the pyloric sphincter and secured with a suture around the sphincter. An exit port, made by silicon tubing, was created 1.5 cm distal to the sphincter. Stimuli including saline (HBSS) and glucose (0.3 M in HBSS) were delivered through individual cannulas (100 μl over 40 s). Saline responses were examined at the end of each imaging session to ensure a lack of background responses due to exit port obstruction and associated mechanical distension (Extended Data Fig. 4c,d).

Bicuculline methiodide (Millipore-Sigma, 14343) was injected twice (each 100 pmol in 50 nl Ringer's solution, 3 nl s⁻¹) using a Nanoject III injector, with one injection 100 μm beneath the obex and a second injection 100 μm below the brainstem surface about 0.3 mm more caudal and 0.3 mm more lateral to the obex. Mice receiving bicuculline or CNO injections were mechanically ventilated throughout the rest of the experiment, and imaging was performed within 30 min of bicuculline injection. CNO (0.1 mg ml⁻¹ in PBS, 2 mg kg⁻¹ intraperitoneal injection, Tocris) was injected 15 min before calcium imaging, and 1 mouse (out of 9) that did not display CNO-induced apnoea was excluded. Subdiaphragmatic vagotomy (Fig. 1d) was performed bilaterally using spring scissors (Fine Science Tools, 15000-04), with cuts made above the hepatic branches of the left and right vagal nerves. Calcium imaging experiments were done on mice containing various *cre* alleles, and results were pooled.

## Calcium imaging data analysis

Lateral motion was corrected by registering the time series to a reference image using Suite2P and/or TurboReg[65,66]. An image averaged across the corrected time series was used to generate a template to delineate the outlines of jRGECO1a-labelled neuronal nuclei, which were semi-automatically detected using a fast-normalized cross-correlation routine as previously described[39,67]. In brief, the averaged images were cross-correlated against a kernel with a size approximating that of an average nucleus. The image was then used to generate a binary mask that demarcated the jRGECO1a-labelled nuclei. The mask was then visually examined, and errors were manually corrected. Fluorescence intensity ($F_t$) was extracted by averaging the intensity of all pixels within the boundaries of the regions of interest. Baseline fluorescence ($F_0$) was determined by averaging jRGECO1a fluorescence over a 24 s period before stimulation onset, and the standard deviation of this pre-stimulus baseline period ($s_0$) was determined. $\Delta F/F$ was calculated as follows:

$$\frac{\Delta F}{F} = \frac{(F_t - F_0)}{F_0}$$

$\Delta F/F$ traces were detrended, except for duodenal chemical perfusion, to remove non-stationary effects such as photobleaching. The response threshold ($\theta$) of each neuron was set as $2.5 \times s_0 + F_0$. A neuron was considered to display a positive response to a mechanical stimulus if $\Delta F/F$ during the stimulation period was (1) above $\theta$ for more than three continuous frames and (2) higher than two times the standard deviation ($s_0'$) above the averaged fluorescence intensity ($F_0'$) of the seven frames before stimulation onset and $\theta$ for at least two continuous frames. A neuron was considered to display a positive response to a chemical stimulus if $\Delta F/F$ during the stimulation period plus the 20 frames after stimulus offset was (1) above $\theta$ for more than four continuous frames and (2) higher than two times the standard deviation ($s_0''$) above the averaged fluorescence intensity ($F_0''$) of the 25 frames before stimulation onset and $\theta$ for at least two continuous frames. Responses were manually examined, and rarely (2.94%), responses were excluded owing to motion artefacts, substantial baseline drifting that could not be corrected by detrending or by physical obstruction of the light path by debris. The responses reported in Fig. 3c and Extended Data Fig. 3f,g,i were maximal $\Delta F/F$ minus $\theta$ during the corresponding period (stimulation period for mechanical stimuli and stimulation

period plus the 20 frames after stimulus offset for chemical stimuli) after smoothing the $\Delta F/F$ using a moving average of five frames. In experiments involving duodenal glucose, fields of view (FOVs) with at least five neurons responsive to every stimulus were used for further analysis. For all other experiments, FOVs with at least two responsive neurons to all stimuli analysed were used.

Spatial distributions of neurons were depicted relative to the centroid of all stomach stretch-responsive neurons (Fig. 2b and Extended Data Figs. 1, 7b,d and 8), stomach site 2 stretch-responsive neurons (Extended Data Fig. 7h), duodenal glucose-responsive neurons (Fig. 2d), duodenal stretch-responsive neurons (Fig. 2g), jRGECO1a-positive, Cre-negative neurons (Extended Data Fig. 6b), slowly adapting neurons (Extended Data Fig. 2e) or neurons not sensitive to cross-inhibition (Extended Data Fig. 10) in the same FOV. Density scatter plots (Fig. 2b,d,g and Extended Data Figs. 1e,h, 2e, 6b, 7b,d,h, 8c,d,i,j and 10a,d,g) were generated using a previously published algorithm[68] to reveal the segregation between stimulus pairs. These plots include neurons that were selectively activated by one of the two stimuli, and colour scales depict the density of neurons that selectively responded to the stimulus indicated, adjusted individually between different organs. Spatial segregation between neurons that responded to two different stimuli was quantified (Extended Data Fig. 7f) using all pairwise distances between neurons responsive to the same stimulus or different stimuli, compared with data from a simulation in which neuron responses were randomly assigned based on the observed response frequency in each FOV (shuffled) to control for regional variation in neuron density. Similarly, segregation between neurons and boutons (Extended Data Fig. 8l) was quantified using all pairwise distances between the same structures (neuron-to-neuron or bouton-to-bouton) or different structures (neuron-to-bouton) and compared with data from a simulation in which the identities of neurons or boutons were randomly assigned based on the observed response frequency in each FOV (shuffled). We used a segregation index (SI) to quantify the extent of spatial segregation between neurons that responded to two different stimuli (Fig. 2e,h and Extended Data Fig. 7e,i) within the same sets of animals. The SI is calculated for each neuron as the difference between average distance to neurons with different responses and average distance to neurons with similar responses, normalized by neuron distances after response shuffling, and was calculated as follows. Within the same image, $X$, $Y$ denote the locations of neurons that respond to stimuli $A$ and $B$, respectively. For simplicity, we write $Z = (X_1, ..., X_n, Y_1, ..., Y_m)$, and let $W$ be the set of 1,000 independent random permutations of $\{1, 2, ..., n + m\}$. The SI of neurons representing stimulus $A$ against neurons representing stimulus $B$ is computed as follows:

$$\mathrm{SI} = \frac{1}{n+m}\left(\sum_{i=1}^{n}\left(\frac{1}{m}\sum_{k=1}^{m}\|X_i - Y_k\| - \frac{1}{n-1}\sum_{\substack{j=1 \\ j \neq i}}^{n}\|X_i - X_j\|\right)\right.$$
$$\left. + \sum_{k=1}^{m}\left(\frac{1}{n}\sum_{i=1}^{n}\|Y_k - X_i\| - \frac{1}{m-1}\sum_{\substack{l=1 \\ l \neq k}}^{m}\|Y_k - Y_l\|\right)\right)$$
$$\Bigg/ \frac{1}{1,000}\sum_{\sigma \in W}\frac{1}{2(n+m)}\left(\sum_{i=1}^{n}\left(\frac{1}{m}\sum_{k=1}^{m}\|Z_{\sigma(i)} - Z_{\sigma(n+k)}\|\right.\right.$$
$$\left. + \frac{1}{n-1}\sum_{\substack{j=1 \\ j \neq i}}^{n}\|Z_{\sigma(i)} - Z_{\sigma(j)}\|\right)$$
$$\left. + \sum_{k=1}^{m}\left(\frac{1}{n}\sum_{i=1}^{n}\|Z_{\sigma(n+k)} - Z_{\sigma(i)}\| + \frac{1}{m-1}\sum_{\substack{l=n+1 \\ l \neq k}}^{n+m}\|Z_{\sigma(n+k)} - Z_{\sigma(n+l)}\|\right)\right)$$

When calculating SI across multiple images, the mean real data (numerator) of neurons across all experiments were calculated before being divided by the mean of shuffled data (denominator). For spatial analysis of stimulus pairs, FOVs containing at least two neurons selectively tuned to each stimulus were used for further analysis. Because sample sizes were variable in Fig. 2b, an additional analysis was performed that involved computer-randomized selection of equal numbers of mice for each stimulus pairing, and similar results observed (Extended Data Fig. 7b).

An enrichment index was used to quantify the relative composition of Cre-defined NTS cell types responsive to each stimulus (Extended Data Fig. 6d). Only neurons responsive to any stimulus were included; $P_{Cre^+}$ is the percentage of Cre-positive neurons responsive to stimulus $A$, and $P_{Cre^-}$ is the percentage of Cre-negative neurons responsive to stimulus $A$ within the same group of animals; the enrichment index was calculated as follows:

$$\mathrm{Enrichment} = \frac{P_{Cre^+} - P_{Cre^-}}{P_{Cre^+} + P_{Cre^-}}$$

For Fig. 4a, responses were measured in the following order: stomach distension ($d_{S1}$), duodenal distension ($d_{D1}$), simultaneous stomach and duodenal distension ($d_{M1}$), second stomach distension ($d_{S2}$), second duodenal distension ($d_{D2}$) and second simultaneous stomach and duodenal distension ($d_{M2}$). A neuron was classified to be suppressed by the stimulus mixture only if the following criteria were met: (1) $d_{M1} < d_{D1}$, (2) $d_{M2} < d_{D2}$ and (3) $(d_{M1} + d_{M2})/2 < d_{D2}$ for duodenum responses; or (1) $d_{M1} < d_{S1}$, (2) $d_{M2} < d_{S2}$ and (3) $(d_{M1} + d_{M2})/2 < d_{S2}$ for stomach responses[69]. The same analyses were performed for cross-inhibition experiments involving laryngeal water and jejunum distension in Extended Data Fig. 10. Per mouse analysis involved separate response quantification of neurons that were or were not suppressed by stimulus mixtures, and included all animals in which both response types were observed.

For Extended Data Fig. 8, weak bleed-through fluorescence of H2B-jRGECO1a into the green channel was computationally removed and by manual curation. The edges of residual GFP signals were detected using the Canny edge detection algorithm, and the centroids of each GFP-positive bouton was determined and manually curated by an experimenter blinded to neuronal response properties or image orientation. Areas of innervation were manually determined by an experimenter blinded to neuronal responses and tracer injection site. Analysis was performed on the focal plane in an imaging stack with the most segregated neuronal responses. Data from multituned neurons that responded to more than one stimulus were included in all corresponding stimuli groups. All analyses were done blinded to neuronal responses in the FOV.

## Statistical analysis

Statistical analyses were performed using GraphPad Prism (GraphPad Software), with statistical tests and sample sizes reported in the figure legends. All replicates are biological, and all statistical tests are two-tailed. No statistical methods were used to predetermine sample sizes, which were based on previous expertise and publications in our field[8,14,34]. Investigators were not blinded to group allocation, except for manual determination of axon innervation (Extended Data Fig. 8), which was performed blind to group allocation and neuronal responses. No method of randomization was used to determine how animals were allocated to experimental groups in Fig. 3 and Extended Data Fig. 1d–i, and all other analyses involved stimulus comparisons in the same animals. In Figs. 1f,h and 3b, Pearson's correlation coefficients were calculated for each stimulus pair by comparing maximal $\Delta F/F$ values across all neurons responsive to any stimulus in the corresponding figures.

## Reporting summary

Further information on research design is available in the Nature Research Reporting Summary linked to this article.

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

**Acknowledgements** We thank S. Arber for providing the *pAAV-Cag-Flex-synaptophysin-Gfp* vector; R. Brust, Y. Wang, J. Zhu, G. Mahler and staff at the Boston Children's Hospital Cellular Imaging Core Facility for assistance with experiments; and Q. Zhao, W. Su, M. Schappe, S. Prescott, C. Zhang and M. Hayashi for comments on the manuscript. The work was supported by NIH grants to S.D.L. (DP1 AT009497, R01 DK122976 and R01 DK103703), the Food Allergy Science Initiative, and a Leonard and Isabelle Goldenson Postdoctoral Fellowship, a Harvard Brain Science Initiative Young Investigator Transitions Award, and an American Diabetes Association Postdoctoral Fellowship to C.R. S.D.L. is an investigator of the Howard Hughes Medical Institute.

**Author contributions** C.R. and S.D.L. designed the experiments. C.R. performed all the experiments. C.R., J.C.B. and C.E.G. analysed the cell-type-specific imaging and vagal axon tracing data. J.A.K. analysed the single-nucleus RNA sequencing data. C.R. analysed all other experiments. C.R. and S.D.L. wrote the manuscript.

**Competing interests** S.D.L. is a consultant for Kallyope. The other authors declare no competing interests.

**Additional information**
**Correspondence and requests for materials** should be addressed to Stephen D. Liberles.

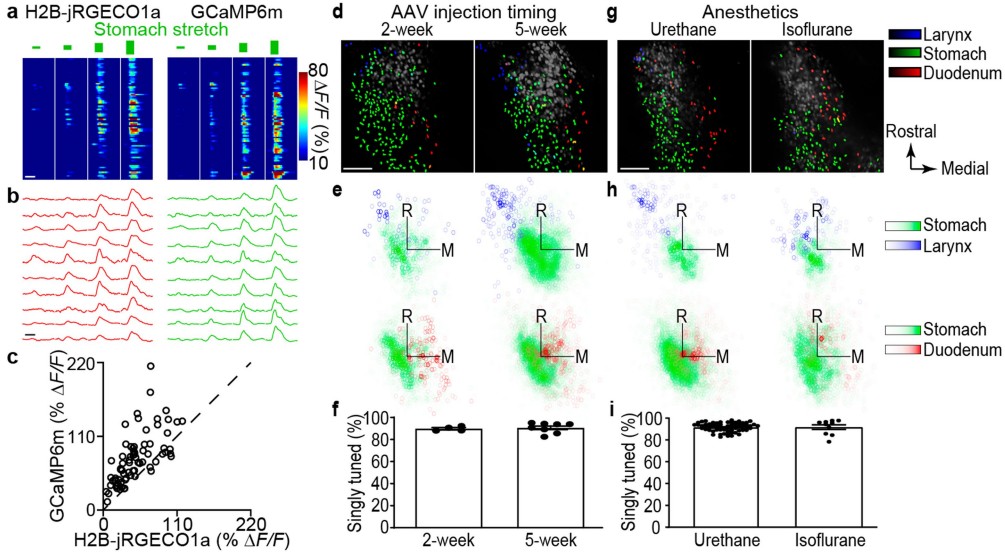

**Extended Data Fig. 1 | Characterizing the calcium sensor, AAV injection time, and anesthetic used during *in vivo* NTS imaging. a**, Mice were injected in the NTS with AAVs encoding H2B-jRGECO1a and GCaMP6m, and NTS imaging was later performed to measure gastric distension-evoked calcium transients. A heat map depicting time-resolved responses of all 74 NTS neurons (2 mice) that displayed both H2B-jRGECO1a (left) and GCaMP6m (right) responses to a series of gastric distensions (green bars of increasing thickness: 150, 300, 600, and 900 μl, white bar: 10 s. **b**, Representative traces depicting normalized $\Delta F/F$ over time for individual neurons (same neurons on left and right), scale bar: 10 s. **c**, Maximal $\Delta F/F$ of NTS neurons responsive to 900 μl gastric distension. **d**, Mice were injected in the NTS with an AAV encoding H2B-jRGECO1a at 2 (left) and 5 (right) weeks of age, and NTS imaging was later performed. Representative two-photon images of H2B-jRGECO1a fluorescence in the NTS (transverse view). Neurons are color-coded based on their relative peak response amplitudes above threshold to laryngeal water (blue, 100 μl), stomach stretch (green, peak response to 150, 300, 600, or 900 μl), or duodenal stretch (red, peak response to 90, 115, or 140 μl), scale bar: 100 μm. **e**, Positions of neurons selectively responsive to stimuli indicated are charted with axis origin corresponding to the centroid for stomach-stretch responsive neurons, top left: 695 neurons, 4 mice, bottom left: 1106 neurons, 4 mice, top right: 2331

neurons, 8 mice, bottom right: 2215 neurons, 6 mice, color scale depicts neuron density (see methods), axis length: 150 μm, R: rostral, M: medial. **f**, Quantifying the percentage of neurons singly tuned to either gastric distension, duodenum distension, or laryngeal water, dots: individual mice, n: 4 (left), 8 (right), mean ± sem. **g**, Representative two-photon images of H2B-jRGECO1a fluorescence in the NTS (transverse view) of mice anesthetized with urethane (2.2–2.6 mg/mg) or isoflurane (1.5%–2.5%). Neurons are color-coded based on their relative peak response amplitudes above threshold to laryngeal water (blue, 100 μl), stomach stretch (green, peak response to 150, 300, 600, or 900 μl), or duodenal stretch (red, peak response to 90, 115, or 140 μl), scale bar: 100 μm. **h**, Positions of neurons selectively responsive to stimuli indicated are charted with axis origin corresponding to the centroid for stomach-stretch responsive neurons, top left: 1136 neurons in 8 mice randomly selected from 49 mice, bottom left: 2525 neurons in 9 mice randomly selected from 98 mice, top right: 1010 neurons, 8 mice, bottom right: 1707 neurons, 9 mice, color scale depicts neuron density (see methods), axis length: 150 μm, R: rostral, M: medial. **i**, Quantifying the percentage of neurons singly tuned to either gastric distension, duodenum distension, or laryngeal water, dots: individual mice, n: 63 (left), 10 (right), mean ± sem.

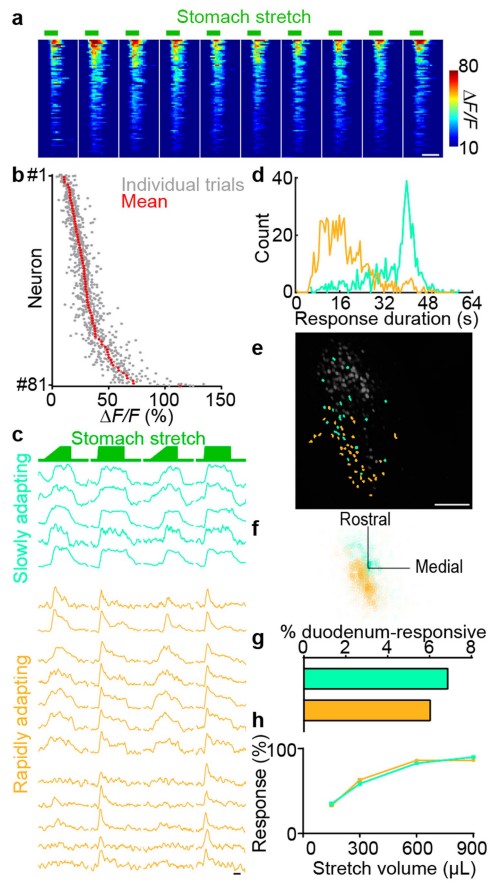

**Extended Data Fig. 2 | Two classes of NTS neurons respond to gastric distension with different adaptation rates. a**, Heat map depicting the time-resolved responses (color coded based on the percentage increase in $\Delta F/F$ of jRGECO1a fluorescence from baseline) of all 81 (1 mouse) responding NTS neurons to ten consecutive gastric distensions (green bars). Each row shows the response of an individual neuron, with rows sorted by response amplitude, white bar: 10 s. **b**, A rank-ordered plot of peak $\Delta F/F$ for all 81 neurons depicted in Fig. 1c, gray circles: peak response amplitudes of individual trials; red circles: means of peak response amplitudes across trials. **c**, Representative traces depicting normalized $\Delta F/F$ over time for individual neurons to repeated stomach stretch (600 μl, distension rate: ~25 μl/s for trials #1 and #3, and ~300 μl/s for trials #2 and #4), scale bar: 10 s. **d**, A histogram depicting response duration (full width at quarter maximum) of slowly adapting (green) and rapidly adapting (yellow) neurons in Fig. 1c. **e**, A representative two-photon image of H2B-jRGECO1a fluorescence in the NTS (transverse view) with neurons pseudocolored based on their slow (green) or rapid (yellow) adaptation to stomach stretch, scale bar: 100 μm. **f**, Positions of neurons that adapt slowly (green, 544 neurons) or rapidly (yellow, 650 neurons) to stomach stretch, with axis origin corresponding to the centroid for slowly adapting neurons, 19 mice, color scale depicts neuron density (see methods), axis length: 200 μm, R: rostral, M: medial. **g**, Percentage of slow-adapting (green, 289 neurons in 13 mice) and fast-adapting (yellow, 493 neurons in 13 mice) stomach stretch-activated neurons that also responded to duodenum stretch (90, 115, or 140 μl). **h**, Responses of slow-adapting (green, 316 neurons in 13 mice) and fast-adapting (yellow, 518 neurons in 13 mice) neurons to various magnitudes of gastric distension; maximal $\Delta F/F$ normalized to 100% for each neuron, mean ± sem.

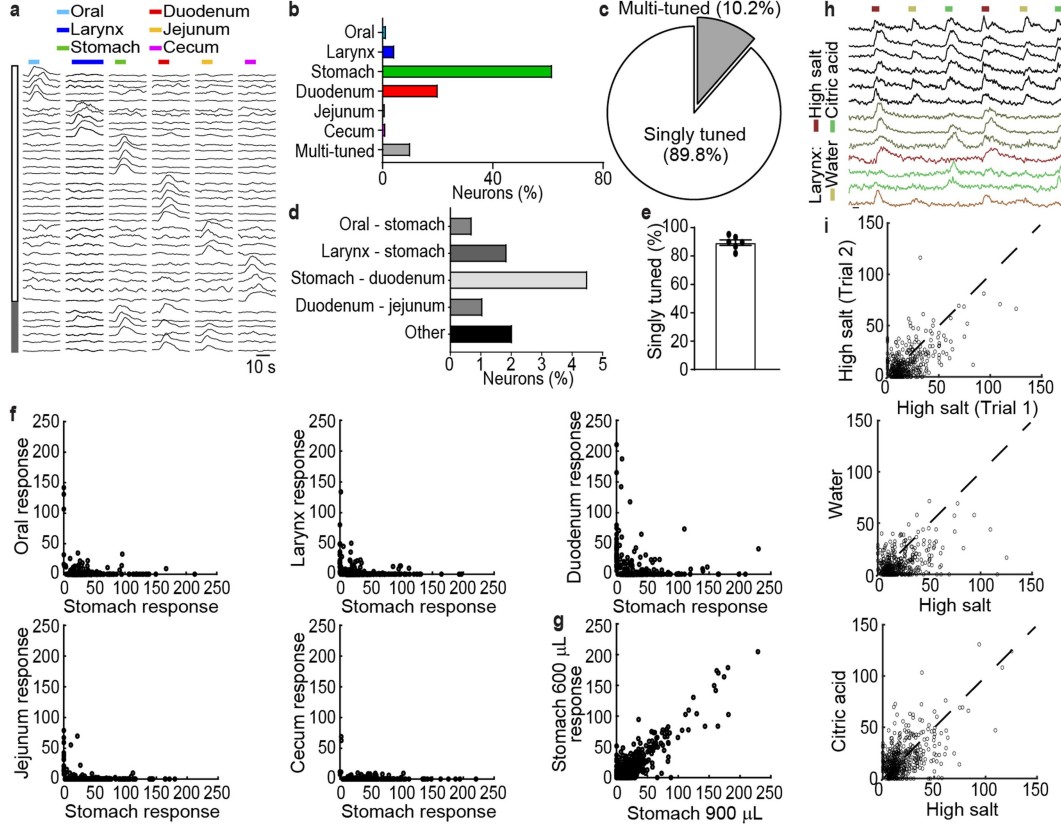

**Extended Data Fig. 3 | Response properties of NTS neurons. a**, Representative traces depicting normalized Δ*F/F* over time for individual neurons from Fig. 1e. **b**, The percentage of neurons from Fig. 1e that selectively respond to different organ inputs or display multi-tuned responses. **c**, Pie chart depicting neurons from Fig. 1e responsive to stimuli from 1 (singly-tuned) or >1 (multi-tuned) organ. **d**, The percentage of multi-tuned neurons in Fig. 1e to various stimulus pairs. **e**, The percentage of singly tuned neurons in each imaged mouse (circles) of Fig. 1e, mean ± sem. **f**, Responses (maximal Δ*F/F* above thresholds) of neurons that responded to any stomach stretch, and/or any stimulation in other organs. Each chart depicts responses of 300 neurons randomly selected from 3815 neurons, 21 mice (oral vs. stomach), from 18895 neurons, 73 mice

(larynx vs. stomach), from 35120 neurons, 113 mice (duodenum vs. stomach), from 21653 neurons, 84 mice (jejunum vs. stomach), and from 4414 neurons, 22 mice (cecum vs. stomach). **g**, Responses (maximal Δ*F/F* above thresholds) of neurons that responded to 600 ml or 900 ml stomach stretch (300 neurons randomly selected from 27125 neurons, 103 mice). **h**, Representative traces depicting normalized Δ*F/F* over time for 13 individual neurons from Fig. 1g, scale bar: 10 s. **i**, Responses (maximal Δ*F/F* above thresholds, see methods) of 362 (left), 404 (middle), and 467 (right) responsive NTS neurons from Fig. 1g. Peak responses were from one trial (left) or were the larger response from two trials (middle, right).

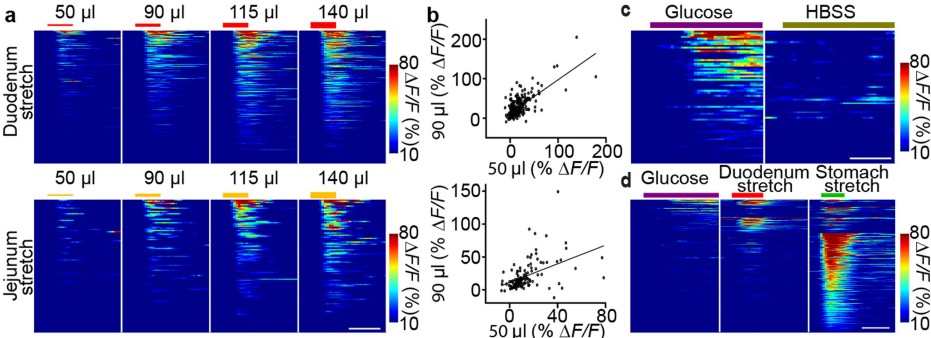

**Extended Data Fig. 4 | NTS responses to intestinal stimuli. a**, Heat maps depicting the time-resolved responses of all responding NTS neurons to various magnitudes of duodenum stretch (top, 271 neurons) and jejunum stretch (bottom, 118 neurons), 6 mice, white bar: 10 seconds. **b**, Peak $\Delta F/F$ of neurons in **a** that responded to stimuli indicated, F test for non-zero slope: \*\*\*\*$P < 0.0001$. **c**, Heat map depicting the time-resolved responses of all 74 responding NTS neurons (2 mice) to duodenal perfusion (24 s) of glucose (300 mM in HBSS, purple) and saline (HBSS, olive), scale bar: 10 s. **d**, Heat map depicting the time-resolved responses of all 492 responding NTS neurons (2 mice) to duodenal perfusion of glucose (300 mM, dark purple), duodenum stretch (red: 140 µl), and stomach stretch (600 µl, green), white bar: 10 s.

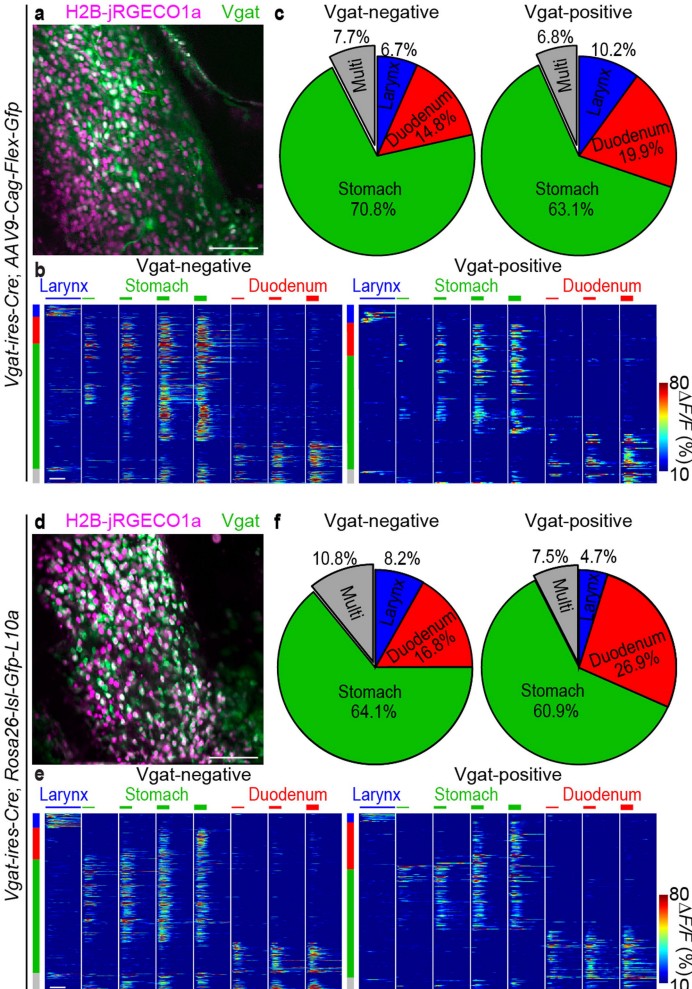

**Extended Data Fig. 5 | Responses of NTS inhibitory neurons.**

a-c, *Vgat-ires-Cre* mice were injected in the NTS with *AAV9-Cag-Flex-Gfp* and *AAV1-Syn-H2b-jRGECO1a* and prepared for *in vivo* two-photon imaging. **a**, A representative image of native fluorescence from H2B-jRGECO1a (magenta) and GFP (green, depicting Vgat neurons) is shown, scale bar: 100 µm. Nonlinear image adjustments were made to ensure visualization of all green cells. **b**, Heat map depicting the time-resolved responses of all 1933 Vgat-negative (left) and 206 Vgat-positive (right) responding NTS neurons in 7 mice to laryngeal water perfusion (dark blue), stomach stretch (green, increasing thickness: 150, 300, 600, and 900 µl) and duodenum distension (red, increasing thickness: 90, 115, and 140 µl), white bar: 10 s. **c**, Pie charts depicting the percentages of Vgat-negative and Vgat-positive neurons responsive to stimuli indicated. **d-f**, *Vgat-ires-Cre* mice were crossed with *Rosa26-lsl-Gfp-L10a* mice and prepared for *in vivo* two-photon imaging. **d**, A representative image of native fluorescence from H2B-jRGECO1 (magenta) and GFP (green, depicting Vgat neurons) is shown, scale bar: 100 µm. Nonlinear image adjustments were made to ensure visualization of all green cells. **e**, Heat map depicting the time-resolved responses of all 766 Vgat-negative (left) and 402 Vgat-positive (right) responding NTS neurons in 5 mice to laryngeal water perfusion (dark blue), stomach stretch (green, increasing thickness: 150, 300, 600, and 900 µl) and duodenum distension (red, increasing thickness: 90, 115, and 140 µl), white bar: 10 s. **f**, Pie charts depicting the percentages of Vgat-negative and Vgat-positive neurons responsive to stimuli indicated.

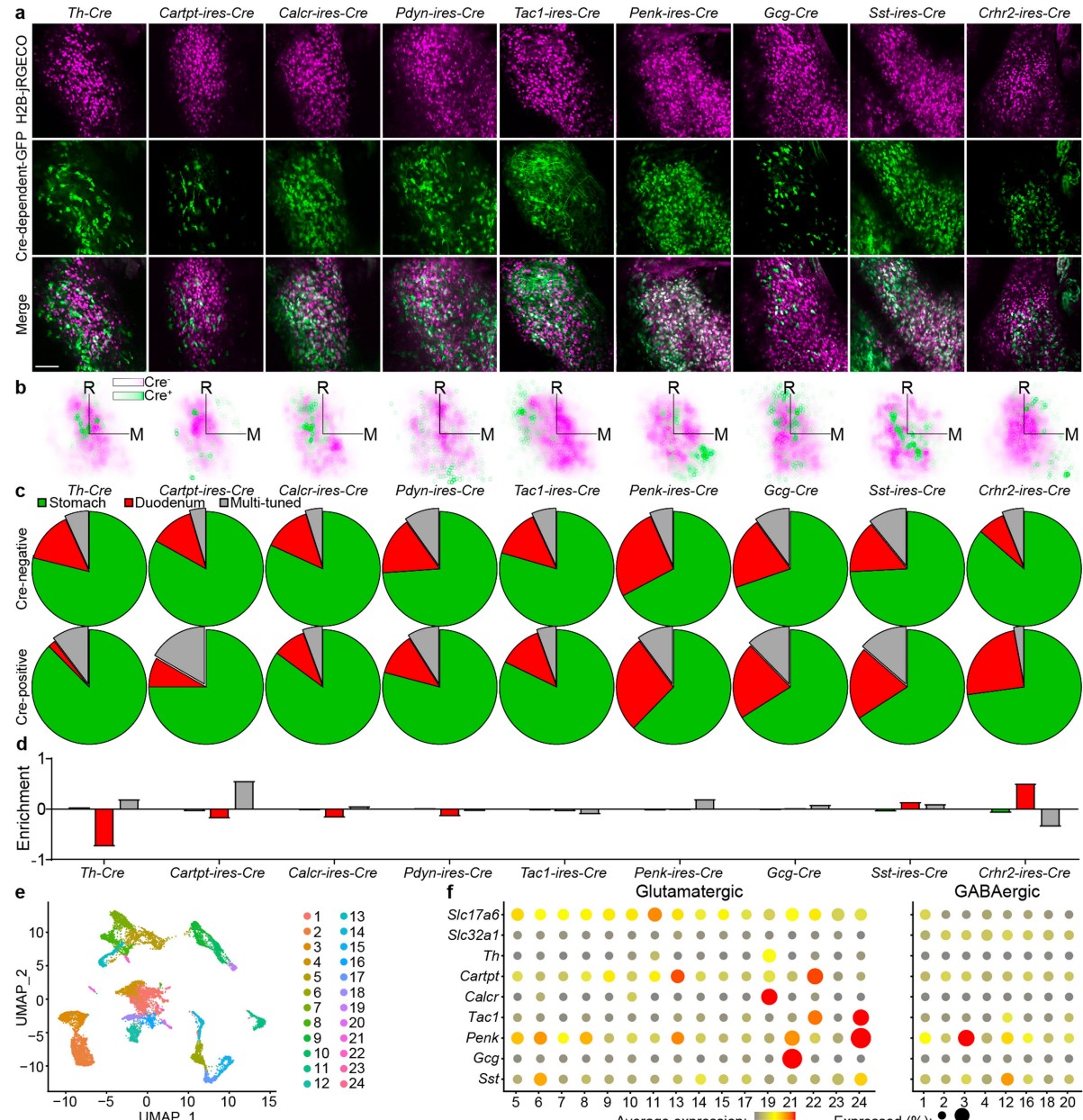

**Extended Data Fig. 6 | Responses of Cre-defined NTS neuron subtypes.**
**a**, NTS imaging was performed on mice containing a Cre-dependent *Gfp* allele and Cre alleles indicated. Representative images of native H2B-jRGECO1 (magenta) and GFP (green) fluorescence, nonlinear image adjustments made to ensure visualization of all green cells, scale bar: 100 μm. **b**, Positions of GFP-positive, H2B-jRGECO1a-positive (green) and GFP-negative, H2B-jRGECO1a-positive neurons (magenta), with axis origin corresponding to the centroid of GFP-negative, H2B-jRGECO1a-positive neurons. Numbers of green/red neurons from 5 *Th-Cre* mice: 354/11748, 3 *Cartpt-ires-Cre* mice: 106/8235, 4 *Calcr-ires-Cre* mice: 756/6911, 3 *Pdyn-ires-Cre* mice: 1/6747, 7 *Tac1-ires-Cre* mice: 456/15016, 4 *Penk-ires-Cre* mice: 1326/8635, 3 *Gcg-Cre* mice: 478/11122, 3 *Sst-ires-Cre* mice:

812/7149, and 5 *Crhr2-ires-Cre* mice: 398/14377, color scale depicts neuron density (see methods), axis length: 200 μm, R: rostral, M: medial. **c**, Pie charts depicting the percentages of responsive GFP-negative, H2B-jRGECO1a-positive (top) and GFP-positive, H2B-jRGECO1a-positive (bottom) neurons responsive to stomach stretch, duodenum stretch, or both (multi-tuned). **d**, An enrichment index quantifies the relative composition of Cre-defined NTS cell types responsive to each stimulus (see Methods). **e**, Uniform manifold approximation and projection (UMAP) plot indicating NTS neuron subtypes based on published single-cell transcriptome data[31]. **f**, Dot plots showing normalized expression of genes indicated across 24 NTS neuron subtypes from **e**.

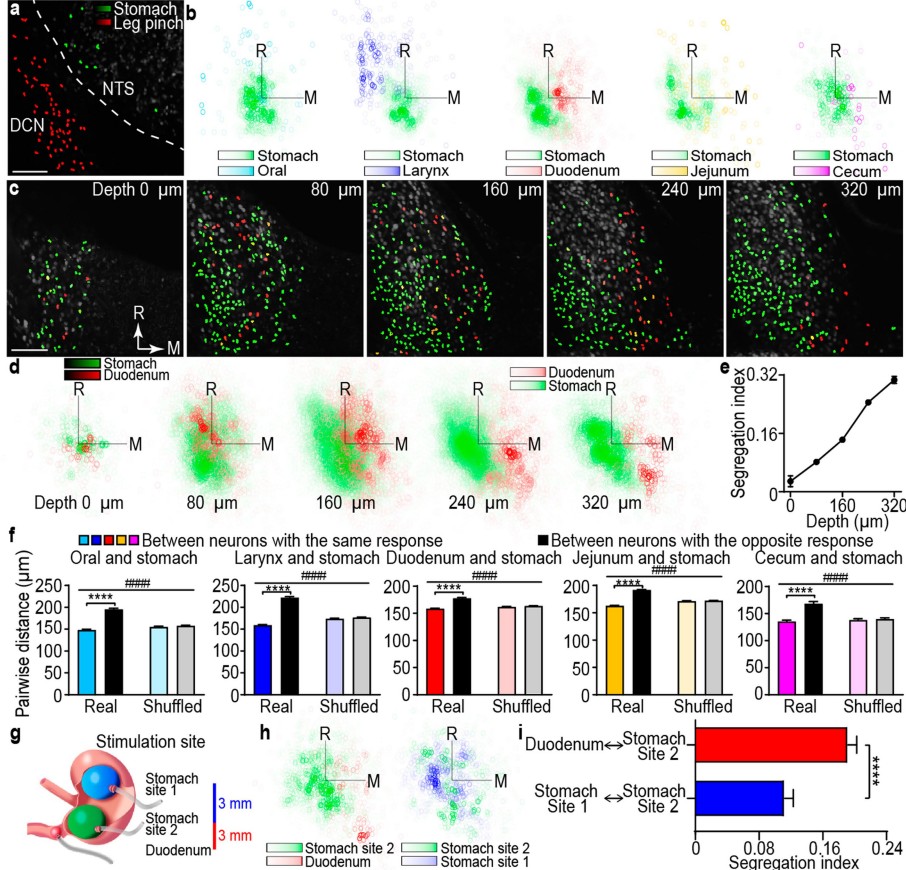

**Extended Data Fig. 7 | Spatial patterning of NTS responses to gut and upper airway stimuli. a**, Representative two-photon image of H2B-jRGECO1a fluorescence in the NTS (transverse view), with neurons color-coded based on their relative peak response amplitudes above threshold to leg pinch (red) and stomach stretch (green, successive 150, 300, 600, and 900 μl distensions), scale bar: 100 μm. **b**, Positions of neurons selectively responsive to stimuli indicated are charted as in Fig. 3b, with analysis here involving 9 randomly selected mice from Fig. 3b to equalize sample size, oral/stomach: 925 neurons, larynx/stomach: 1128 neurons, duodenum/stomach: 2347 neurons, jejunum/stomach: 1153 neurons, cecum/stomach: 415 neurons (same data as Fig. 3b), color scale depicts neuron density (see methods), axis length: 150 μm, R: rostral, M: medial. **c**, Representative two-photon images at various depths of H2B-jRGECO1a fluorescence in the NTS (transverse view), with neurons color-coded based on their relative peak response amplitudes above threshold to stomach stretch (green, successive 150, 300, 600, and 900 μl distensions) and duodenum stretch (red, successive 90, 115, or 140 μl distensions), scale bar: 100 μm. **d**, Positions of neurons at various depths selectively responsive to stimuli as described in **c** are charted with axis origin corresponding to the centroid for stomach-stretch responsive neurons, 0 μm: 455 neurons, 17 mice, 80 μm: 2517 neurons, 46 mice, 160 μm: 4909 neurons, 53 mice, 240 μm: 4445

neurons, 52 mice, 320 μm: 2204 neurons, 39 mice, color scale depicts neuron density, axis length: 150 μm, R: rostral, M: medial. **e**, Quantifying spatial segregation of neurons in **d**, mean ± sem, see methods for Segregation Index. **f**, Pairwise distances between neurons responsive to the same stimulus (colored) or different stimuli (black). Data from Fig. 3b (real) were compared with data from a simulation where neuron responses were randomly assigned based on the observed response frequency in each field of view (shuffled) to control for regional variation in neuron density, oral/stomach: 1204 neurons, 11 mice, larynx/stomach: 10610 neurons, 57 mice, duodenum/stomach: 28556 neurons, 107 mice, jejunum/stomach: 13050 neurons, 66 mice, cecum/stomach: 415 neurons, 9 mice, mean ± sem, ####P < 0.0001, significant interaction in two-way analysis of variance between responder types and shuffling, ****P < 0.0001, Šídák multiple comparisons test. **g**, Cartoon depicting sites of balloon distension in the gastrointestinal tract. **h**, Positions of neurons responsive to stimuli indicated are charted relative to the centroid (coordinate origin) of neurons responsive to distention of stomach site 2, left: 1155 neurons, 4 mice, right: 927 neurons, 4 mice, axis length: 150 μm, R: rostral, M: medial. **i**, Quantifying spatial segregation of neurons in **h** responsive to stimuli indicated, mean ± sem, ****P < 0.0001, two-tailed Mann-Whitney test, see methods for Segregation Index.

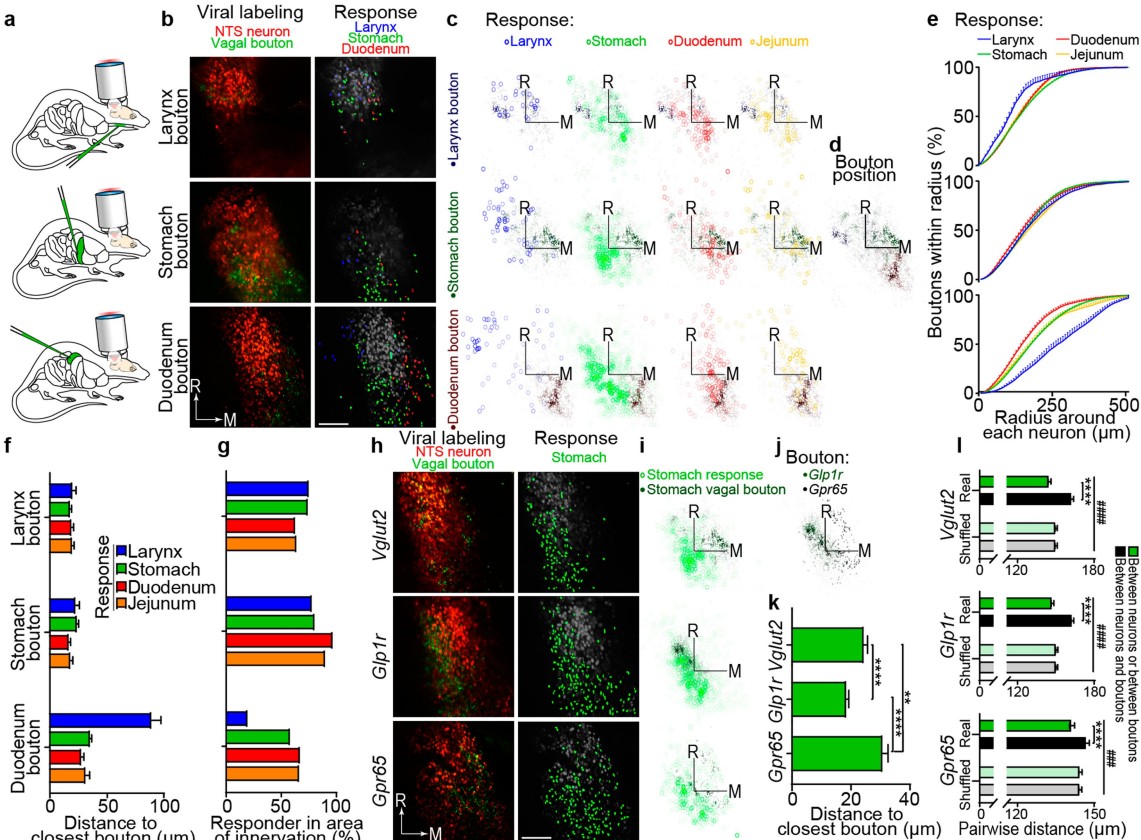

**Extended Data Fig. 8 | Comparing the positions of vagal axons and responsive NTS neurons. a**, Cartoon depicting NTS two-photon imaging in *Vglut2-ires-Cre* mice injected with *AAV1-Cag-Flex-Synaptophysin-Gfp* in the larynx (top), stomach (middle), and duodenum (bottom), and *AAV1-Syn-H2b-jRGECO1a* in the NTS. **b**, Left: representative two-photon images (transverse view) of NTS neuronal nuclei (red, jRGECO1a) and boutons of vagal axons from organs indicated (green, Synaptophysin-GFP). Right: the same images with NTS neurons color-coded based on their relative peak response amplitudes above threshold to laryngeal water (100 µl, blue), stomach stretch (green, successive 150, 300, 600, and 900 µl distensions), and duodenal stretch (red, successive 90, 115, and 140 µl distensions), R: rostral, M: medial, scale bar: 100 µm. **c**, Positions of vagal axon boutons from the larynx (top row, dark blue, 489 boutons, 3 mice), stomach (middle row, dark green, 966 boutons, 3 mice), and duodenum (bottom row, dark red, 891 boutons, 5 mice) are charted along with the positions of NTS neurons selectively responsive to 100 µl laryngeal water (left column, blue), any distension (150, 300, 600, or 900 µl) of stomach (2nd left column, green), any distension (90, 115, or 140 µl) of duodenum (2nd right column, red), or any distension (90, 115, or 140 µl) of jejunum (right column, orange). Neurons depicted top to bottom: larynx 24, 54, 57, stomach 209, 364, 555, duodenum 91, 92, 109, jejunum 69, 59, 58, axis origin: centroid for stomach-stretch responsive neurons, axis length: 150 µm, R: rostral, M: medial. **d**, Positions of vagal axon boutons from **c** with axis origin corresponding to the centroid for stomach-stretch responsive neurons, axis length: 150 µm, R: rostral, M: medial. **e**, The percentages of vagal axon boutons from the larynx (top), stomach (middle), and duodenum (bottom) in **c** that are located within a variable radius from any neuron responsive to stimuli indicated, mean ± sem. **f**, Distances between NTS neurons responsive to stimuli indicated and the closest boutons of vagal axons from the larynx (top), stomach (middle), and duodenum (bottom), mean ± sem. **g**, Percentages of NTS

neuron responsive to stimuli indicated that are located in the area of innervation (see methods) of vagal axons from different organs. **h**, Left: representative two-photon images (transverse view) of NTS neuronal nuclei (red, jRGECO1a) and boutons of vagal axons (green, Synaptophysin-GFP) visualized by injecting *AAV1-Cag-Flex-Synaptophysin-Gfp* into the stomach of *Vglut2-ires-Cre* (top), *Glp1r-ires*-Cre (middle), or *Gpr65-ires-Cre* (bottom) mice, scale bar: 100 µm. Right: the same images with NTS neurons colored in green to indicate neurons responsive to stomach distension. **i**, Positions of axonal boutons (dark green) from stomach neurons labeled in *Vglut2-ires-Cre* (top, 966 boutons, 3 mice, same data as **c**), *Glp1r-ires*-Cre (middle, 1276 boutons, 3 mice), or *Gpr65-ires-Cre* (bottom, 243 boutons, 3 mice) mice and compared to the positions of NTS neurons responsive to stomach stretch, axis origin: centroid for stomach-stretch responsive neurons, top: 364 neurons (same data as **c**), middle: 539 neurons, bottom: 244 neurons, axis length: 150 µm, R: rostral, M: medial. **j**, Positions of axonal boutons from **i**. **k**, Distances between NTS neurons responsive to stomach stretch and the closest boutons of vagal axons from the stomach of *Vglut2-ires-Cre* (same data as in **f**), *Glp1r-ires-Cre*, and *Gpr65-ires-Cre* mice, mean ± sem, **P < 0.01, ****P < 0.0001, Dunn's multiple comparisons test following Kruskal-Wallis test of significance. **l**, Pairwise distances were calculated between NTS neurons responsive to stomach stretch and axonal boutons of stomach neurons from Cre lines indicated (black) and compared with an average of bouton-bouton pairwise distances and NTS neuron-NTS neuron pairwise differences (green). Real data were compared with data from a simulation where identity of bouton or NTS neuron was randomly assigned based on observed frequency in each field of view (shuffled) to control for regional variation in neuron or bouton density, mean ± sem, ###P < 0.001, ####P < 0.0001, significant interaction in two-way analysis of variance between responder types and shuffling, ****P < 0.0001, Šídák multiple comparisons test.

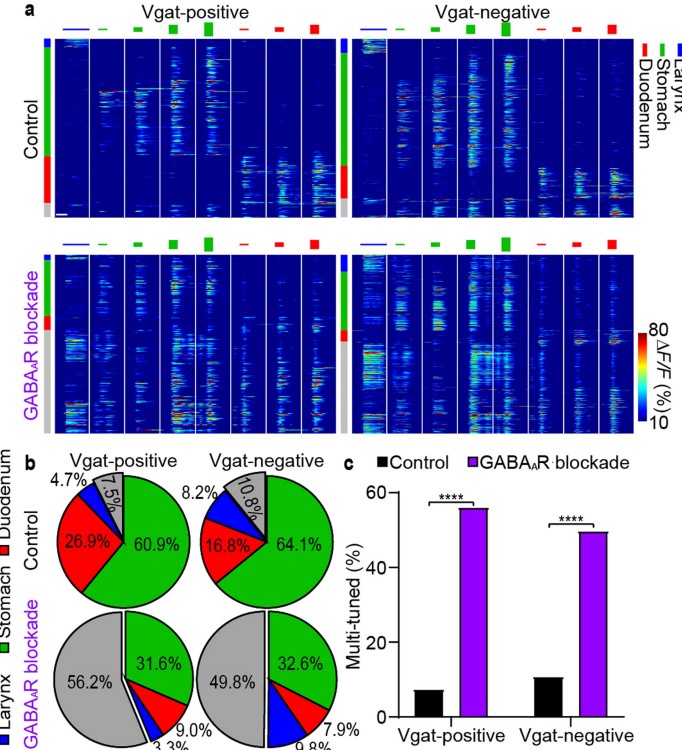

**Extended Data Fig. 9 | Inhibition shapes responses of both NTS excitatory and inhibitory neurons. a**, *Vgat-ires-Cre*; *Rosa26-lsl-Gfp-L10a* mice were injected in the NTS with *AAV1-Syn-H2b-jRGECO1a*, and *in vivo* NTS imaging was performed with (bottom) or without (top) NTS-localized administration of bicuculline. Heat maps depict the time-resolved responses of Vgat-positive (left) and Vgat-negative (right) neurons to laryngeal water (dark blue, 100 µl), stomach stretch (green, increasing thickness: 150, 300, 600, and 900 µl) and/or duodenum distension (red, increasing thickness: 90, 115, and 140 µl).

Top: 402 Vgat-positive and 766 Vgat-negative neurons, 5 mice, same data as Extended Data Fig. 5e; bottom: 365 Vgat-positive and 851 Vgat-negative neurons, 2 mice, white bar: 10 s. **b**, Pie charts depicting the percentages of Vgat-positive (left) and Vgat-negative (right) neurons from **c** responsive to stimuli indicated with (bottom) or without (top, same data as Extended Data Fig. 5f) bicuculline administration. **c**, The percentage of Vgat-positive and Vgat-negative neurons from **a** responsive to stimuli from >1 organ (multi-tuned), ****$P$ < 0.0001, two-tailed $\chi^2$ test.

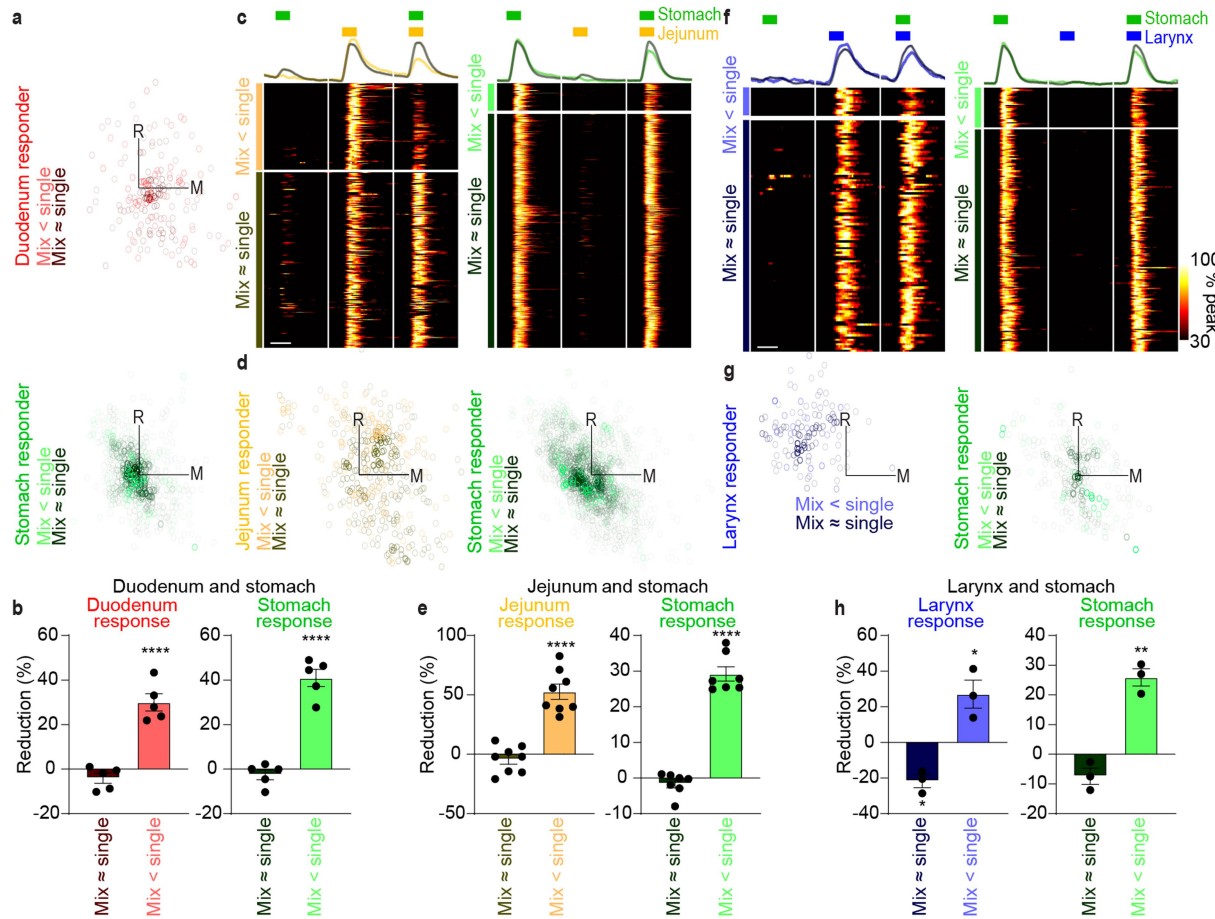

**Extended Data Fig. 10 | Characterizing NTS response suppression by multiple input pairs. a**, Positions of neurons from Fig. 4a selectively responsive to duodenum stretch (top) or stomach stretch (bottom), with differential coloring based on sensitivity to lateral inhibition, axis origin: centroid for uninhibited neurons, axis length: 150 μm, R: rostral, M: medial. **b**, Response suppression in stimulus pairs indicated measured on a per-animal basis in neurons that received dual stimulation-evoked inhibition (Mix < single, right) or did not (Mix ≈ single, left), data points: average response changes of all neurons in a single mouse, 5 mice. **c**, Average traces (top) or heat maps (bottom) of normalized, time-resolved NTS neuron responses to successive application of stomach stretch (green, 600 μl), jejunal stretch (yellow, 115 μl), and both. Neurons that selectively detected jejunum stretch (left, 378 neurons) or stomach stretch (right, 1183 neurons) were classified into two groups were dual stimulation evoked inhibition (Mix < single; left: orange, 125 neurons, right: light green, 124 neurons) or did not (Mix ≈ single; left: gray, 253 neurons, right: dark green, 1059 neurons, 8 mice), white bar: 10 s. **d**, Positions of neurons from **c** selectively responsive to jejunum stretch (left) or stomach stretch (right), with differential coloring based on sensitivity to lateral inhibition, axis origin: centroid for uninhibited neurons, axis length: 150 μm, R: rostral, M: medial.

**e**, Response suppression in stimulus pairs indicated measured on a per-animal basis in neurons that received dual stimulation-evoked inhibition (Mix < single, right) or did not (Mix ≈ single, left), data points: average response changes of all neurons in a single mouse, 7 mice. **f**, Average traces (top) or heat maps (bottom) of normalized, time-resolved NTS neuron responses to successive application of stomach stretch (green, 600 μl), laryngeal water (blue), and both. Neurons that selectively detected laryngeal water (left, 119 neurons) or stomach stretch (right, 243 neurons) were classified into two groups were dual stimulation evoked inhibition (Mix < single; left: light blue, 13 neurons, right: light green, 37 neurons) or did not (Mix ≈ single; left: dark blue, 106 neurons, right: dark green, 206 neurons, 3 mice), white bar: 10 s. **g**, Positions of neurons from **f** selectively responsive to laryngeal water (left) or stomach stretch (right), with differential coloring based on sensitivity to lateral inhibition, axis origin: centroid for uninhibited neurons, axis length: 150 μm, R: rostral, M: medial. **h**, Response suppression in stimulus pairs indicated measured on a per-animal basis in neurons that received dual stimulation-evoked inhibition (Mix < single, right) or did not (Mix ≈ single, left), data points: average response changes of all neurons in a single mouse, 3 mice, mean ± sem, *P < 0.05, **P < 0.01, ****P < 0.0001, Šídák multiple comparisons test.

# Reporting Summary

## Statistics

For all statistical analyses, confirm that the following items are present in the figure legend, table legend, main text, or Methods section.

| n/a | Confirmed | |
|---|---|---|
| ☐ | ☒ | The exact sample size (*n*) for each experimental group/condition, given as a discrete number and unit of measurement |
| ☐ | ☒ | A statement on whether measurements were taken from distinct samples or whether the same sample was measured repeatedly |
| ☐ | ☒ | The statistical test(s) used AND whether they are one- or two-sided<br>*Only common tests should be described solely by name; describe more complex techniques in the Methods section.* |
| ☒ | ☐ | A description of all covariates tested |
| ☐ | ☒ | A description of any assumptions or corrections, such as tests of normality and adjustment for multiple comparisons |
| ☐ | ☒ | A full description of the statistical parameters including central tendency (e.g. means) or other basic estimates (e.g. regression coefficient) AND variation (e.g. standard deviation) or associated estimates of uncertainty (e.g. confidence intervals) |
| ☐ | ☒ | For null hypothesis testing, the test statistic (e.g. *F*, *t*, *r*) with confidence intervals, effect sizes, degrees of freedom and *P* value noted<br>*Give P values as exact values whenever suitable.* |
| ☒ | ☐ | For Bayesian analysis, information on the choice of priors and Markov chain Monte Carlo settings |
| ☒ | ☐ | For hierarchical and complex designs, identification of the appropriate level for tests and full reporting of outcomes |
| ☐ | ☒ | Estimates of effect sizes (e.g. Cohen's *d*, Pearson's *r*), indicating how they were calculated |

*Our web collection on statistics for biologists contains articles on many of the points above.*

## Software and code

Policy information about availability of computer code

| Data collection | Olympus FluoView software was used for collecting two-photon images. |
|---|---|
| Data analysis | Seurat (4.0.5), R (4.1.1), ImageJ (1.52q), Matlab (R2020a), Processing (4.0a3), Python (3.9.4), OpenCV (4.5.1), and GraphPad Prism (9.0.2) were used for for data analysis. |

For manuscripts utilizing custom algorithms or software that are central to the research but not yet described in published literature, software must be made available to editors and reviewers. We strongly encourage code deposition in a community repository (e.g. GitHub). See the Nature Portfolio guidelines for submitting code & software for further information.

## Data

Policy information about availability of data

All manuscripts must include a data availability statement. This statement should provide the following information, where applicable:
- Accession codes, unique identifiers, or web links for publicly available datasets
- A description of any restrictions on data availability
- For clinical datasets or third party data, please ensure that the statement adheres to our policy

Source data are provided with this paper at Nature, and raw images are available upon reasonable request.

# Field-specific reporting

Please select the one below that is the best fit for your research. If you are not sure, read the appropriate sections before making your selection.

☒ Life sciences ☐ Behavioural & social sciences ☐ Ecological, evolutionary & environmental sciences

For a reference copy of the document with all sections, see nature.com/documents/nr-reporting-summary-flat.pdf

# Life sciences study design

All studies must disclose on these points even when the disclosure is negative.

| | |
|---|---|
| Sample size | Sample sizes were based on prior expertise and publications in our field, including Williams et al., Cell 2016, Zhao et al., Nature, 2022, Ichiki et al., Nature, 2022, and are disclosed in each figure legend. |
| Data exclusions | The following criteria were established prior to analysis: in experiments involving duodenal glucose, fields of view (FOVs) with at least five neurons responsive to every stimulus were used for further analysis; in all other experiments, FOVs with at least two responsive neurons to all stimuli analyzed were used. For spatial analysis of stimulus pairs, only FOVs containing at least 2 neurons selectively tuned to each stimulus were used for further analysis. Neuronal responses were all manually examined, and rarely (2.94%), responses were excluded due to motion artifacts, dramatic baseline drifting that could not be corrected by detrending, or by physical obstruction of the light path by debris. For chemogenetic activation experiment, 9 experimental mice were injected with clozapine-N-oxide. 1 mouse (out of 9) that did not display CNO-induced apnea was excluded. |
| Replication | All experiments were successfully reproduced. Extended Data Fig. 2a, b were reproduced with another set of the same stimulus in the same animal, and Extended Data Fig. 7a was reproduced in three animals in total. The numbers of animals used in all figure were reported in figure legends in detail. |
| Randomization | No method of randomization was used to determine how animals were allocated to experimental groups in Fig. 3, and all other analyses involved stimulus comparisons in the same animals. |
| Blinding | We did not perform the analyses blinded except for Extended Data Fig. 8, in which GFP-labeled boutons were determined by an experimenter blind to neuronal response properties or image orientation, and areas of innervation were determined manually by an experimenter blind to neuronal responses and tracer injection site (group allocation). Other experiments were analyzed using computer codes and were not subjected to potential experimenter bias. |

# Reporting for specific materials, systems and methods

We require information from authors about some types of materials, experimental systems and methods used in many studies. Here, indicate whether each material, system or method listed is relevant to your study. If you are not sure if a list item applies to your research, read the appropriate section before selecting a response.

## Materials & experimental systems

| n/a | Involved in the study |
|---|---|
| ☒ | ☐ Antibodies |
| ☒ | ☐ Eukaryotic cell lines |
| ☒ | ☐ Palaeontology and archaeology |
| ☐ | ☒ Animals and other organisms |
| ☒ | ☐ Human research participants |
| ☒ | ☐ Clinical data |
| ☒ | ☐ Dual use research of concern |

## Methods

| n/a | Involved in the study |
|---|---|
| ☒ | ☐ ChIP-seq |
| ☒ | ☐ Flow cytometry |
| ☒ | ☐ MRI-based neuroimaging |

## Animals and other organisms

Policy information about studies involving animals; ARRIVE guidelines recommended for reporting animal research

| | |
|---|---|
| Laboratory animals | Animals were maintained under constant temperature (23 ± 1°C) and relative humidity (46 ± 5%) with a 12-h light/dark cycle. Mice used in this study are from both sexes and of a mixed genetic background. Glp1r-ires-Cre (029283), Gpr65-ires-Cre (029282) and Crhr2-ires-Cre (033728) mice were made previously in the lab and are now available at Jackson Laboratory. Slc17a6-ires-Cre (Vglut2-ires-Cre, 028863), Slc32a1-ires-Cre (Vgat-ires-Cre, 028862), Rosa26-lsl-Gfp-L10a (024750), Tac1-ires-Cre (021877), Pdyn-ires-Cre (027958), Penk-ires-Cre (025112), Cartpt-ires-Cre (028533), Sst-ires-Cre (013044), Th-Cre (021877), and Gcg-Cre (030542) mice were purchased from Jackson Laboratory. All mice used in the experiments were older than 2 weeks. |
| Wild animals | No wild animals were used. |
| Field-collected samples | No field-collected samples were used. |

Ethics oversight | All animal procedures followed the ethical guidelines outlined in the NIH Guide for the Care and Use of Laboratory Animals, and all protocols were approved by the institutional animal care and use committee (IACUC) at Harvard Medical School.

Note that full information on the approval of the study protocol must also be provided in the manuscript.

