## [Peer Review File · Nature]

Manuscript Title: A brainstem map for visceral sensations

Reviewer Comments & Author Rebuttals

Reviewer Reports on the Initial Version:

Referees' comments:

Referee #1 (Remarks to the Author):

Overview

The manuscript “A visceral sensory homunculus in the brainstem” describes a technique to capture single neuron activity of the nucleus tractus solitarius (NTS) neurons in response to manipulation of the alimentary canal and the larynx. How the NTS integrates and separates visceral information is an important question that will be of broad interest to many fields. However, there are several major concerns regarding the technique, incomplete data presentation, and limited biological advances of the manuscript. Explanation and examples of these major concerns are outlined below.

Technique: There are important details hidden in the methods that are needed to fully interpret the data. For example, the age of the mice when AAV is injected is 2-4 weeks, at this point the circuits for feeding are not fully developed and the manipulation, ie the inflammatory response to AAV, stress of taking the pups away from their mother, breast milk vs independent feeding, could alter the neural map described. On page 4, it says “Brain motion artifacts precluded imaging in the awake mouse”. There is no reference to removal of the cerebellum until the Methods. Thus, it seems more likely that the invasive surgical technique precluded imaging in the awake mouse, not motion artifacts. Likewise, Figure 1a is an inaccurate depiction of the imaging technique as the entire cerebellum appears intact in the cartoon schematic. In addition, the authors consistently use water to simulate stretch in the larynx. The authors recently published a paper using an almost identical method in which they find that water as a stimulus induces a response in the larynx. Therefore, what they refer to as a stretch response in the larynx is actual a response to a water stimulus and is unsuitable for comparison within the “stretch” category.

Incomplete data presentation: Sample size of mice and total neurons imaged is highly variable throughout the manuscript and at times unacceptable. The most egregious example of this is Figure 2, where 2 mice and 328 neurons comprise an entire figure. Another example of this is Figure 3b and 3c where the stomach and cecum data were collected from 9 mice and 415 neurons, yet the stomach and duodenum from 56 mice and 14,252 neurons. The maps are presented as comparable in the text and figure, but clearly cannot be taken as such due to the large variability in sample size. Also, the authors repeatedly neglect to show the variability across mice. Figure 1 presents a combined total of 1133 neurons pooled from 6 different mice. From this combined set of neurons, they claim “...the vast majority of responsive neurons (89.8%, 1018/1133 neurons, 6 mice) were selectively stimulated by signals emanating from a particular organ.” However, this differs from the 70.8%, and

63.1% values stated in Extended Data 2. If 6 mice were actually imaged, the results should be stated as the mean with standard error to know the variability across mice, and the reliability of the imaging preparation.

Limited biological advances: Throughout the manuscript, the language overstates the data presented. The title of the manuscript and the titles of Figure 1 and 3 refer to a “visceral sensory homunculus” and an “internal organ representation.” These statements bring to mind all visceral or internal organs, not simply the gastrointestinal tract plus the larynx. The authors refer to cardiovascular and respiratory integration in the NTS repeatedly yet make little to no attempt to experimentally address how these functions are integrated in the brainstem. Other organs (pancreas, bladder, liver, etc.) are ignored. Simply, a visceral homunculus in the brainstem was not defined, or even interrogated here. At best, this manuscript describes the NTS integration of the gastrointestinal stretch response. From the data presented, there is limited new biological knowledge gained. The majority of the comparisons made throughout the manuscript are NTS neuron responses to mechanical stretch of distinct GI regions. As such, the outcomes are predicted by early tracing studies coupled with previous cFos staining. Many of these studies are not referenced. See work by Rinaman, Schwartz, Moran, and others.

Referee #2 (Remarks to the Author):

Our internal systems are innervated by vagal nerves that provide input used by the brain to monitor and regulate physiological processes. Numerous studies have shown how select vagal neurons to play roles in the regulation of blood pressure, breathing and feeding. Recently, a much greater appreciation has emerged regarding the transcriptional and functional diversity of vagal neurons. From these studies, it is clear that many distinct types of vagal neurons exist, and each are highly specialized in terms of which organs they target and the types of stimuli they detect (mechanical, chemical, etc). Also known is that each type of vagal neuron relays information to the brain through the selective targeting of their axons to specific regions in the nucleus of the solitary tract (NTS) of the brainstem. The NTS is a highly heterogenous structure with distinct regions dedicated to the processing of taste versus other modalities such as visceral sensation. Less well understood is the diversity of cell types within the NTS or their anatomical organization (inputs and outputs). However, recent studies have described NTS neurons that can be defined by the expression of single genes, respond to specific cues and that influence select behaviors (e.g. feeding).

In this new study, Chen and colleagues have developed a preparation to examine the responses of large numbers of NTS neurons in vivo. They reason that wide-field calcium imaging should be a powerful method to examine how sensory stimuli emanating from various organs becomes represented by ensembles of NTS neurons. They provide convincing data for regionalization within the NTS and evidence for the existence of organotypic map that is sharpened by inhibition.

Overall, this is a very technically challenging study, and the authors deserve credit for applying a two photon imaging approach in a region notoriously difficult to study in vivo. However, it is also clear that significant compromises had to be made to achieve these recordings. These limitations raise

some concerns about the conclusions that can or should be drawn.

Major points:

1. Three major technical concerns are the use of anesthetics, the choice of genetically encoded indicator and the reliance on non-physiological stimuli.
 - a. For the most part, in vivo imaging within the brain has moved away from the use of anesthetics because it is well known these drugs dramatically alter network dynamics. The effects of urethane on the response properties is not controlled for or addressed.
 - b. Red-shifted calcium indicators are significantly less sensitive and have poorer temporal resolution than GFP-based sensors. Nuclear calcium flux in the nucleus is a fraction of that typically measured in the cytosol. Hence, the performance of the h2b-jrcamp indicator needs to be benchmarked against GCaMP6 (or later). Even better would be validation of their major conclusions with electrical recordings.
 - c. The stimuli delivered are invasive and bear little resemblance to the types of physiological stimuli that the NTS neurons would be normally responding to. Can this imaging approach detect NTS responses to normal nutrient or food cues or to stimulation of specific vagal neuron types?
2. Work from the Liberles groups and others have already shown genetically distinct types of neurons have regionally specific projections in the NTS. The evidence provided in this study relies on tracing from the stomach using an AAV expressing synGFP. These new data are insufficiently detailed or quantified to support claims that the axonal organization does not predict the functional map. Co-labeling and using more selective markers on gut innervating vagal neurons that target different organs or the same organ with different ending types would significantly help.
3. The homogeneity of the evoked-calcium responses is curious. Based on the data, it appears that NTS neurons only about whether a specific organ is stimulation, irrespective of the type of stimulation. The responses seem to have very little dynamics or differences. Given the known cell type diversity of the vagal and NTS neurons, there seems to be something missing here? One possibility is the combination of the anesthesia, low resolution indicator and coarse sensory stimuli is providing highly filtered data that does not reflect the normal functional state of NTS neurons. Another concern is the representative image in Figure 1B. At baseline, most of the neurons seem already very bright and the few responding neuron are ones that are much dimmer at baseline. This may be warning about unhealthy cells or reflect a poor dynamic range of the red-shift nuclear localized indicator?
4. According to the methods, the authors used many Cre lines that have been previously shown to mark NTS neurons with functional roles in taste, feeding etc but then, somewhat unusually, lump them all together in the analyses. Does this mean that every "cell type" previously described in the NTS has the same functional properties?
5. The use of Vgat-Cre to label inhibitory neurons with GFP is fine (extended data fig 2), but there are

many types of inhibitory neurons known to have very different physiological and functional properties. Furthermore, looking at the GFP negative cells is insufficient- how efficient is the AAV viral labeling, presumably many Vgat-positive cells? The non-GFP cells will be a mixture of excitatory and inhibitory neurons.

6. In Fig 4, can the authors say something more about the circuit mechanism of inhibition? Do they believe that local inhibition from neurons that themselves respond to visceral stimuli is the predominant source of inhibition? Or do long range inputs play the role? If the former, do the authors see any evidence in their imaging data for inhibitory circuit motifs known to be involved in this kind of mapping e.g. lateral inhibition, center surround, etc.

7. Homunculus means “tiny human”, clearly there are no small people in the mouse brainstem. More seriously, this study shows there is coarse topographical representation of the digestive tract in the NTS and so the title, abstract and introduction should be toned down.

Minor comments:

1. Figure 1e. Please show more baseline before stimulation

2. There is a general lack of scale bars or bars without labels in the legends (e.g. figures 1g, 2b, 4).

3. Fig 4, it would be helpful if they could compare other organs in addition to leg pinch, (ie: lung, diaphragm, heart) to get a better sense of the boundaries

4. The two proposed models of spatial organ representation (continuous vs discrete) seem an oversimplification. They authors focus on a few organs and there remains a possibility that gradients beside the one depicted in their example cartoon exist – what would happen if other organs, position in the body axis, modality, functional role or physiological state were also considered?

5. Although reference numbers are limited, a deeper overview of what is known about the NTS from electrical recordings and other approaches should be added, perhaps at the expense of the comparisons to other sensory systems..

Referee #3 (Remarks to the Author):

The manuscript by Ran et al., aims to deduce decoding parameters of visceral interoception in the dorsal brainstem (nucleus of the solitary tract, NTS) with a focus on anatomical location. By meticulous and elegant real-time in vivo cellular Ca⁺ recordings after selective stimuli at distinct locations along the GI-tract, it is discovered that neurons responding to stimuli from different locations are spatially segregated. Most neurons uniquely respond to stimuli from a single GI-region, with little evidence of sub-patterning within major GI-region. In contrast, the same neurons respond

to different types of stimuli evoked from the same region. By comparing sensory vagal innervation and responding neuron location a mismatch was suggested. Last, inhibitory actions between NTS neurons were found pivotal to establish specific response patterns. NTS neurons activated by duodenal stimuli were silenced by inhibitory neurons when the stomach was probed and vice versa, implying that cross-inhibitory mechanisms shape final responses.

While the stringency, advanced in vivo methodologies, overall presentation, intent and findings of this study were highly appreciated, the overall conceptual advance remains limited. The focus on spatial pattern (also indicated by the title) is unfortunate as this is the least novel/interesting aspect of the study (at least without a stringent correlation to molecular or innervation patterns). In contrast, the two last sections on the possible mismatch between sensory neuron innervation and responding neurons, and the important role of inhibitory networks to shape responses, have potential strong conceptual novelty and impact on the field. Unfortunately, these parts were performed with less precision and left at a preliminary stage. Additional experiments to elevate these parts along with alignment of molecular information would be needed.

Major Concerns and Questions:

1) The authors indicate a possible mismatch between the location of responding neurons and the innervating vagal sensory axons – this was also stated as motivation for the study in the introduction. However, the hypothesis is only tested in one experiment (Extended Data Fig 7) where stomach neurites are related to the responders of other regions. There are two problems with this part – the limited scope of the experiment, as well as the precision to which it is carried out. To gain clarity of the extent of mismatch it would be important to test all vagal neurite connections with all responders, i.e. systematically label connections from larynx, duodenum, jejunum and cecum and assess the relation to responders of all these regions. The analysis also has to be performed with greater precision. The stomach innervation is claimed to be regionally patterned, yet this pattern is not presented (AP-DV-ML) either in itself or in relation to the responding neurons. Moreover, distance to closest bouton is not a very precise measure of connections. One weak bouton (possibly without a corresponding postsynaptic connection) would be equally counted as 10 strong boutons if they are at a similar distance. It would be of higher value to count the numbers of strong boutons within a set radius around each responding neuron. It would also be valuable to define the area of innervation and count how many of the responding neurons that fit that area, or correlate the area of responders to the overall area of innervating boutons. It is moreover hard to correlate the representative pictures (Ext Data Fig b) with the actual statistical table (Ext Data Fig 7c) – it appears as if many responders would be several cell diameters away from the closest bouton, while others are touched by boutons - the statistics however suggests that most neurons are connected to boutons (error bar very small). Furthermore, the conclusion that dendritic arborization of brainstem neurons allows them to respond at a distance is not verified or quantified anywhere in the paper (on the contrary the statistics indicate that all neuron cell bodies are relatively close to boutons representing the vagal innervation of a given region). To assess spatial association between dendrites and innervating axons a fluorescent response marker that marks the full morphology of the cells could be used. The question thus remains unanswered, as no attempts of comparing the location of responding neuronal dendrites and innervating axons are made.

2) The spatial segregation of responding neurons is presented as a surprising finding and contrasted to the distributed pattern of transcriptionally distinct vagal sensory neurons. However, this is the

generic difference between the central and peripheral nervous system, stemming from the very different ways in which these systems are formed. While the CNS is generated from spatially distinct patterned stem cells that differentiate into stereotypically arranged neuron types, ultimately providing the CNS a clear functional anatomy, the PNS is generated from neural crest cells that have lost the intricate layers of spatial identity and therefore form ganglionic structures consisting of intermingled subtypes. Patterned versus distributed neuron patterns are known major distinguishing features of the CNS and PNS - thus the highlighting of this difference as something novel should be avoided.

3) The spatial organization of NTS neurons is claimed to constitute an independent feature, separated from intrinsic gene expression programs - however this is never tested in the manuscript. On the contrary it would seem likely that the response pattern is genetically coded, ie absence/presence of specific receptors governing whether they would respond or not, absence/presence of specific adhesion/molecules that governs contact with other neuron types (for example the inhibitory neurons). It is thus paramount to align responding cells to their gene expression profiles. This could be achieved by in situ sequencing (using probes selected for known neuron subtypes), multiplexed IHC or RNAscope. Ideally the experiment would use spatial transcriptomics which is to be released in 2022 from 10xgenomics, allowing full genome sequencing of single cells in situ. Without information of cell identities/gene expression profiles, the presentation of spatial patterning provides little impact to the field. Molecular information of responding cells would constitute an important resource to further understanding the finetuning of each modality/organ sensation and the connectivity of NTS to other regions of the brain.

4) Both excitatory and inhibitory neurons were activated as response to all stimuli. It would be important to understand whether the excitatory and inhibitory neurons are segregated in space.

5) The observed role of inhibitory neurons to shaping the response patterns is very interesting and could form a central part of the manuscript. It will be important to expand and investigate this phenomenon in more detail. For instance, the cross-inhibition could also be investigated between all possible pairs of focus in the study, not only stomach/duodenum, to understand how general/specific the mechanism is. The mechanism of cross-inhibition indicate that the initial response of inhibitory neurons is more specific and helps to shape the response of excitatory neurons - thus greater amounts of excitatory neurons are probably able to respond by default. Please investigate whether neuron that are de-inhibited as a result of GABA-receptor blockade indeed are excitatory neurons and not inhibitory neurons. Linking back to the innervation pattern from distinct regions – assessing the spatial relation between the innervating axons of one region and only the inhibitory responders of each region could integrate different parts of this study better and reveal further insights into the decoding mechanisms of interoceptive stimuli.

6) Sections shown in Fig. 3 are confusing in relation to borders and axis, it would be helpful to include schematic drawings of the anatomical region and indicate DV and AP locations in each picture.

Minor issues

1) Fig 3b How do these graphs correspond to the anterior border of the area postrema?

2) Fig 3g Does the third graph compare duodenum to jejunum site 2 (not stomach to jejunum site 2 as indicated in the figure)?

3) Figure Legend 3 page 37; row 15. “f responses” should probably be “g responses”.

4) The methods indicate that Pdyn, Sst, Penk, Carpt and Th transgenic animals are used in the study, however they are nowhere to be found in figures nor results part. This could indicate that attempts to characterize the NTS neurons molecularly has been made and a part of the manuscript is missing, alternatively these experiments were removed at a late stage?

Author Rebuttals to Initial Comments:

Dear David,

Thank you for handling our manuscript, which is now entitled "A brainstem map for visceral organ sensations" (Manuscript Nature 2021-04-06692). We were pleased by the helpful and supportive comments of the referees, and are submitting a revised manuscript along with the point-by-point response below (our comments are in blue). We have performed a large number of new experiments and edited the text accordingly, and think the revised manuscript is substantially improved.

New experiments and revisions include:

(1) Cell type-specific responses to visceral stimuli. How molecularly defined NTS cell types respond to stimuli is an important aspect of NTS coding and attracted the interest of all three reviewers. Here, we imaged the responses of NTS neurons marked in 9 published Cre lines which have been used in the literature to denote either functionally relevant and/or transcriptome-defined NTS neuron populations, including (1) *Th-Cre*, (2) *Cartpt-Cre*, (3) *Calcr-Cre*, (4) *Pdyn-Cre*, (5) *Tac1-Cre*, (6) *Penk-Cre*, (7) *Gcg-Cre*, (8) *Sst-Cre*, and (9) *Crhr2-Cre* mice. We observed that each Cre line marked cells responsive to different visceral stimuli; it is clear that different visceral inputs engage a heterogenous assortment of cell types, and that each of these Cre-defined cell types is recruited across multiple sensory representations. In general, a similar percentage of responding cells was observed in each line, although quantitative differences were noted in a few cases. These new data allow us to put forth the concept that the NTS is comprised of distinct spatially defined modules for particular sensory inputs, with each NTS module containing diverse and mostly overlapping cell types (Extended Data Fig. 6).

(2) Comparing vagal axon innervation with NTS responses. We vastly expanded our previous studies involving visualization of vagal axons in the context of NTS imaging. First, we added new injection sites, additionally tracing all vagal neurons from the duodenum and larynx. Second, we refined vagal axon labeling by using genetic markers (*Glp1r-ires-Cre* and *Gpr65-ires-Cre*) that preferentially label gut mechanoreceptors and chemoreceptors. Third, we expanded our quantitative analysis by measuring numerous parameters suggested by reviewer 3. Together, these new findings allowed us to refine our conclusions that both vagal axon targeting and local inhibition help shape the spatial pattern of NTS responses. These new data are included in Extended Data Fig. 8.

(3) Expanding studies of lateral inhibition. We performed additional experiments to investigate lateral inhibition in the NTS. First, we expanded our stimulus repertoire, and in new data, also

observed effective cross-inhibition between jejunum and stomach stimulation, but a substantially attenuated cross-inhibition between larynx and stomach stimulation; these findings suggest that inhibitory neurons display at least some selectivity in cross-inhibition. Second, we observed that neurons prone to and insensitive to cross-inhibition are spatially intermingled within the NTS. All of these new data are included in Extended Data Fig. 10.

(4) Heterogeneous response kinetics. We performed new experiments to explore response heterogeneity in the NTS, and observed two classes of stomach stretch responses by k-means clustering that can be described as rapidly adapting and slowly adapting, and were consistent across repeated stimulation. Rapidly adapting and slowly adapting neurons occupied slightly different locations in the NTS but did not differ in response threshold or likelihood of responding to stimulation of another organ. The new results are presented in Figure 1 and Extended Data Fig. 2.

(5) Chemogenetic activation of inhibitory neurons during NTS imaging. We previously reported that inhibition in the NTS modulates responses to visceral stimulation, but the source of inhibition remained elusive. In new data, we used chemogenetic approaches to activate local NTS inhibitory neurons, and observed drastic suppression of NTS responses to visceral stimuli. These new results are included in Extended Data Fig. 10.

(6) Benchmarking of the nuclear-localized, red-shifted calcium indicator, H2B-jRGECO1a. We co-expressed H2B-jRGECO1a with the ultra-sensitive calcium indicator GCaMP6m and compared their responsiveness to stomach stretch in the same NTS neurons. Consistent with previous literature, we found that H2B-jRGECO1a reports neuronal activation with good sensitivity, which is only modestly (1.9 times) less sensitive than GCaMP6m. Based on these findings, H2B-jRGECO1a is slightly more sensitive than GCaMP6f and substantially more sensitive than GCaMP5G. These new data are presented in Extended Data Fig. 1.

(7) NTS response under isoflurane anesthesia. In the original manuscript, functional imaging was performed only under urethane anesthesia. In new data, we additionally imaged NTS responses in mice anesthetized with isoflurane, an anesthetic with a distinct mechanism of action. We observed that NTS spatial maps and neuronal tuning were remarkably similar between urethane- and isoflurane-anesthetized mice, and present these findings as a new figure (Extended Data Fig. 1) in the manuscript. In addition, before establishing the anesthetized preparation used in the manuscript, we attempted to image the NTS in awake mice. Here we demonstrate the necessity of anesthesia by comparing videos from our NTS imaging experiment before and after anesthesia in the same mouse (Supplementary Video 1).

(8) NTS responses with different AAV injection timing. At the request of Reviewer #1, we examined the effect of early age (pre-weaning) AAV injection by comparing NTS responses in mice who received AAV injections 2 weeks or 5 weeks after birth. We found that varying the age of AAV injection had no discernible impact on NTS neuronal tuning or spatial organization. These new findings are presented in Extended Data Fig. 1.

(9) Characterization of NTS inhibitory neurons. In the original manuscript, we examined the response properties of NTS inhibitory neurons by Cre-dependent viral labelling in *Vgat-Cre* mice. We agree that viral approaches may not label all neurons, so in new data, we additionally performed NTS imaging experiments in *Vgat-Cre; Rosa-IsI-Gfp-L10a* mice that do not require viral injection for neuron labeling. These experiments provided consistent data that inhibitory neurons are (1) widely distributed across the NTS and (2) exhibited response properties similar to excitatory neurons. These new data strengthened our previous data and are included in Extended data Fig. 5. We also performed additional experiments to show that both excitatory and inhibitory neurons display similarly generalized responses after disinhibition by bicuculline (Extended Data Fig. 9).

(10) A discrete map of organ identity. We previously observed that responses to stretch of the stomach and intestine (with distensions separated by only a few mm) evoked NTS responses that were more spatially segregated than stretch of rather remote regions of the intestine. Based on this observation, we concluded that organ identity is a more salient feature than absolute space. We did new experiments involving distension of different regions of the stomach which provided similar conclusions. These new findings are presented in Extended Data Fig. 7.

We also increased the sample size of several experiments, extended the range of test stimuli used, performed additional analyses demonstrating the high consistency of imaging results across individual mice, and made several textual revisions according to the reviewers' suggestions. Together, these new experiments and analyses have dramatically broadened the scope of the original manuscript and strengthened the previous conclusions.

Thank you for considering publication of our work.

Best,
Steve Liberles

Referees' comments:

Referee #1 (Remarks to the Author):

Overview

The manuscript “A visceral sensory homunculus in the brainstem” describes a technique to capture single neuron activity of the nucleus tractus solitarius (NTS) neurons in response to manipulation of the alimentary canal and the larynx. How the NTS integrates and separates visceral information is an important question that will be of broad interest to many fields. However, there are several major concerns regarding the technique, incomplete data presentation, and limited biological advances of the manuscript. Explanation and examples of these major concerns are outlined below.

Technique: There are important details hidden in the methods that are needed to fully interpret the data. For example, the age of the mice when AAV is injected is 2-4 weeks, at this point the circuits for feeding are not fully developed and the manipulation, ie the inflammatory response to AAV, stress of taking the pups away from their mother, breast milk vs independent feeding, could alter the neural map described.

Thank you for raising this possibility. We performed new experiments to compare NTS responses after AAV injection in 2- and 5-week-old mice and observed that varying the age of AAV injection had no discernible impact on NTS neuronal tuning or spatial organization. These new findings are presented in Extended Data Figure 1.

On page 4, it says “Brain motion artifacts precluded imaging in the awake mouse”. There is no reference to removal of the cerebellum until the Methods. Thus, it seems more likely that the invasive surgical technique precluded imaging in the awake mouse, not motion artifacts. Likewise, Figure 1a is an inaccurate depiction of the imaging technique as the entire cerebellum appears intact in the cartoon schematic.

Brain motion artifacts in freely behaving mice are dramatic, and we include a supplementary video (now Supplementary Video 1) showing the extent of motion artifact in awake mice. Unlike most parts of the brain that are largely stable when the mouse is head-fixed, the NTS is partly located behind the skull. Displacement can be as large as hundreds of microns when an awake mouse moves its neck, shifting NTS neurons completely out of the field of view, as shown in the video. We previously mentioned that part of the cerebellum was removed in the main text, and also now clarify this in the cartoon too. For comparison, we note that many other *in vivo* neuron imaging preparations are substantially more invasive and can involve aspiration of sizable forebrain and midbrain regions (for example, for imaging in the thalamus, hippocampus, trigeminal ganglion, etc., Dombeck et al., *Nat. Neurosci.*, 2010; Song et al., *Nat. Methods*, 2017; von Buchholtz et al., *Neuron*, 2021; Yarmolinsky et al., *Neuron*, 2016; Ghitani et al., *Neuron*, 2017; Liang et al., *Cell*, 2018.)

In addition, the authors consistently use water to simulate stretch in the larynx. The authors recently published a paper using an almost identical method in which they find that water as a stimulus

induces a response in the larynx. Therefore, what they refer to as a stretch response in the larynx is actual a response to a water stimulus and is unsuitable for comparison within the “stretch” category.

We fully agree with the reviewer that laryngeal water and stretch are fundamentally distinct sensory inputs, even if they trigger overlapping reflexive responses. Indeed, in our recent 2020 paper, we find that they are mediated by at least partially distinct sensory neuron populations. It was by no means our intention to equate them, so we have clarified the text. We also noted that distension of the larynx was not technically achievable in the context of the imaging preparation. Nevertheless, it is clear that NTS regions responsive to laryngeal water occupy a discrete NTS subregion from NTS neurons responsive to any other stimulus examined.

Incomplete data presentation: Sample size of mice and total neurons imaged is highly variable throughout the manuscript and at times unacceptable. The most egregious example of this is Figure 2, where 2 mice and 328 neurons comprise an entire figure. Another example of this is Figure 3b and 3c where the stomach and cecum data were collected from 9 mice and 415 neurons, yet the stomach and duodenum from 56 mice and 14,252 neurons. The maps are presented as comparable in the text and figure, but clearly cannot be taken as such due to the large variability in sample size.

We agree that we performed a much larger number of experiments involving stomach/duodenum stimulation, including experiments involving across-organ stimulation (Fig. 1), lateral inhibition (Fig. 4, Extended Data Fig. 10), inhibitory neuron tuning (Extended Data Fig. 5), intestinal nutrients (Extended Data Fig. 3), dorsoventral distribution of responsive neurons (Extended Data Fig. 7), vagal axon tracing (Extended Data Fig. 8), discrete/continuous maps (Fig. 3, Extended Data Fig. 7), and NTS cell types (Extended Data Fig. 6) experiments. Each of these experiments additionally provided spatial information, and we felt no reason to discard any data. Instead, we felt that all data should be combined without exclusion to provide exceptionally precise positional information for stomach and duodenum responses in Figure 3. To control for increased sample size, we now performed an experiment involving computer-randomized selection of equal numbers of mice for each stimulus pairing, and observed the same results; these findings are now presented in Extended Data Fig. 7. The incredibly large sample size of 56 mice and 14,252 neurons (now even larger with new data) goes well beyond what is reported in many high throughput neuronal imaging studies in mice (Ichiki et al., *Nature*, 2022: 2 mice in Figure 5, Gong et al., *Cell*, 2020: 2 mice in Figures 3 and 4; Xu et al., *Science*, 2020: 3 mice; Livneh et al., *Nature*, 2017: 3 mice in Figures 1 and 2; Marshall et al., *Nature*, 2020: 3 mice in Figure 2, and many others). Other sample sizes are also robust, and we performed additional experiments to increase the sample size to 4 mice in Figure 2 as suggested. Together, our findings provide compelling evidence for a stereotyped viscerotopic map in the NTS.

Also, the authors repeatedly neglect to show the variability across mice. Figure 1 presents a combined total of 1133 neurons pooled from 6 different mice... [sentences moved below]... If 6 mice were actually imaged, the results should be stated as the mean with standard error to know the variability

across mice, and the reliability of the imaging preparation.

We included new data analysis showing the percentage of singly tuned neurons per mouse, with each mouse as a single data point, as the reviewer suggests. This new figure (Extended Data Fig. 3) shows the impressive consistency of responses across each of the six imaged mice. We also note that the vast majority of large-scale imaging papers pool together all cells across mice (see Ackels et al., *Nature*, 2021; Mann et al., *Nature*, 2021; Reinert et al., *Nature*, 2021; Fustiñana et al., *Nature*, 2021; Peters et al., *Nature*, 2021; Ruder et al., *Nature*, 2021; Zhou et al., *Nature*, 2021, and numerous other examples).

From this combined set of neurons, they claim “...the vast majority of responsive neurons (89.8%, 1018/1133 neurons, 6 mice) were selectively stimulated by signals emanating from a particular organ.” However, this differs from the 70.8%, and 63.1% values stated in Extended Data 2.

We think there is a misunderstanding about Extended Data 2 (which is now Extended Data 5a-c in the revised manuscript), as these data are in extremely close agreement. To clarify this figure, 70.8/63.1% of neurons were stomach-selective, 14.8/19.9% were duodenum selective, 6.7/10.2% were larynx-selective, and 7.7/6.8% were multi-tuned. Thus, in Extended Data 2, ~93% of neurons were selective for one of three stimuli tested, while in Figure 1, 89.8% of neurons were selective for one of six stimuli tested. Both data sets strongly support the claim that the vast majority of responsive neurons were selectively stimulated by signals emanating from a particular organ.

Limited biological advances: Throughout the manuscript, the language overstates the data presented. The title of the manuscript and the titles of Figure 1 and 3 refer to a “visceral sensory homunculus” and an “internal organ representation.” These statements bring to mind all visceral or internal organs, not simply the gastrointestinal tract plus the larynx. The authors refer to cardiovascular and respiratory integration in the NTS repeatedly yet make little to no attempt to experimentally address how these functions are integrated in the brainstem. Other organs (pancreas, bladder, liver, etc.) are ignored. Simply, a visceral homunculus in the brainstem was not defined, or even interrogated here. At best, this manuscript describes the NTS integration of the gastrointestinal stretch response. From the data presented, there is limited new biological knowledge gained. The majority of the comparisons made throughout the manuscript are NTS neuron responses to mechanical stretch of distinct GI regions. As such, the outcomes are predicted by early tracing studies coupled with previous cFos staining. Many of these studies are not referenced. See work by Rinaman, Schwartz, Moran, and others.

Thank you for this thought-provoking comment. We agree with the reviewer's introductory point that understanding how the NTS integrates and separates visceral information is an important question that will be of broad interest to many fields. Different models for NTS organization have been speculated, and arguments both for or against spatial organization in the NTS have been put forward (See for example Andresen et al 2012, Llewellyn-Smith 2011, Paton 1999). Like the reviewer,

we also find many Fos studies (of Linda Rinaman, Schwartz, Moran, Berthoud, Gebhart, and others) as well as vagal axon tracing to be highly inspiring, and we have done a better job of citing this important work. Nevertheless, we would like to point out several fundamental differences between functional imaging, tracing of axonal inputs, and Fos staining.

Tracing vagal sensory neurons by organ injection provides highly useful information regarding the anatomical location of vagal axons in the NTS (Altschuler et al., *J. Comp. Neurol.*, 1989; Katz and Karten, *J. Comp. Neurol.*, 1983; Rinaman et al., *J. Neurosci.*, 1989; Shapiro and Miselis, *J. Comp. Neuro.*, 1985) but cannot reveal the response properties of post-synaptic, higher-order NTS neurons, which is the central question of our manuscript. Most NTS neurons have elaborate dendritic arbors and potentially contact sensory axons from a distance (Paton, *Exp. Physiol.*, 1999; Zhang et al., *J. Comp. Neurol.*, 1995). Many NTS neurons are higher order neurons that do not receive direct vagal inputs at all (McDougall et al., *J. Neurosci.*, 2009). Some NTS neurons are polymodal, and their responses cannot be explained by a single vagal projection field. Other limitations are that early tracing experiments involved organ dye injection and cannot differentiate functionally distinct neuronal types that innervate the same organ, such as mechanoreceptors, chemoreceptors, and osmoreceptors, cannot easily distinguish axon terminals, fibers-of-passage, and motor neuron dendrites that may be co-labeled without close scrutiny, and lack multiplexed analysis to compare the innervation patterns of numerous different organs in the same animal, as shown in Figure 1. Finally, our findings clearly demonstrate that tracing from internal organs as performed in older studies lacks sufficient power to predict NTS neuron responses (Extended Data Fig. 8). We identify a key role for local NTS inhibition in shaping the spatial response pattern, a phenomenon that was not and could not be revealed by axon tracing, and was only observed by a high-throughput analysis of real-time responses.

Fos studies have provided valuable insights into NTS responses, but also have several limitations. Only one stimulus can be examined per animal, preventing a direct comparison of whether a single neuron can (1) respond to multiple visceral cues, (2) respond to various intensities/durations of the same cue, or (3) be sensitive to acute pharmacological modulation. Furthermore, Fos responses are not analyzed in real-time but instead involve a delay of minutes to hours, during which time secondary neuronal pathways could be engaged, and not all neurons in the brain produce Fos equally following activation. We feel that our high throughput, real-time analysis of NTS responses allows us to make definitive and important conclusions about how interoceptive stimuli are encoded in the brain. In our paper, we show for the first time that the vast majority of NTS neurons preferentially respond to stimulation of one particular organ (Fig. 1, 3, Extended Data Fig. 3, 5, 6), while different types of stimuli from the same organ can sometimes activate the same neuron (Fig. 2, Extended Data Fig. 4). Furthermore, we show that neurons can respond with different dynamics and response thresholds (Fig. 1, Extended Data Fig. 2). Finally, we show a key role for fast-acting local inhibition in shaping responses (Fig. 4, Extended Data Fig. 9-10). We greatly appreciate the insights from decades of research using vagal tracing and Fos staining. However, most conclusions in the manuscript were made possible only with the high-throughput optical recording system, and we are excited to share this powerful platform with the viscerosensory community.

We fully agree with the reviewer that most of our claims are derived from studies of the gastrointestinal tract and larynx, and as such we have edited the manuscript as the reviewer suggests (including changing the title). We note that the field doesn't yet understand physiological stimuli that selectively stimulate many or even most classes of vagal neurons, including those from the liver or pancreas for example. In addition, some stimuli like airway stretch or the baroreceptor reflex could not be applied for technical reasons during in vivo imaging such as stimulation-induced motion, 3D tissue deformation for long-term imaging, and brain hemorrhage due to elevated blood pressure during surgery. For these reasons, we focused on a few well-defined vagal stimuli to reveal concepts of NTS coding. We agree that the manuscript focuses mostly on visceral stimuli of the gut-brain axis and larynx, which we already consider significant topics, and have edited the text accordingly. We fully expect that findings from gut and larynx sensory pathways will generalize to other organ systems, but discussion of lower respiratory and cardiovascular systems is toned down as suggested.

Referee #2 (Remarks to the Author):

Our internal systems are innervated by vagal nerves that provide input used by the brain to monitor and regulate physiological processes. Numerous studies have shown how select vagal neurons to play roles in the regulation of blood pressure, breathing and feeding. Recently, a much greater appreciation has emerged regarding the transcriptional and functional diversity of vagal neurons. From these studies, it is clear that many distinct types of vagal neurons exist, and each are highly specialized in terms of which organs they target and the types of stimuli they detect (mechanical, chemical, etc). Also known is that each type of vagal neuron relays information to the brain through the selective targeting of their axons to specific regions in the nucleus of the solitary tract (NTS) of the brainstem. The NTS is a highly heterogeneous structure with distinct regions dedicated to the processing of taste versus other modalities such as visceral sensation. Less well understood is the diversity of cell types within the NTS or their anatomical organization (inputs and outputs). However, recent studies have described NTS neurons that can be defined by the expression of single genes, respond to specific cues and that influence select behaviors (e.g. feeding).

In this new study, Chen and colleagues have developed a preparation to examine the responses of large numbers of NTS neurons in vivo. They reason that wide-field calcium imaging should be a powerful method to examine how sensory stimuli emanating from various organs becomes represented by ensembles of NTS neurons. They provide convincing data for regionalization within the NTS and evidence for the existence of organotypic map that is sharpened by inhibition.

Overall, this is a very technically challenging study, and the authors deserve credit for applying a two-photon imaging approach in a region notoriously difficult to study in vivo. However, it is also clear that significant compromises had to be made to achieve these recordings. These limitations raise some concerns about the conclusions that can or should be drawn.

We thank the reviewer for appreciating the quality, significance, and technical difficulty of our study. We performed new experiments to address each of the remaining comments from the reviewer as detailed below. We also extended our investigation of some questions the reviewer found

particularly interesting.

Major points:

1. Three major technical concerns are the use of anesthetics, the choice of genetically encoded indicator and the reliance on non-physiological stimuli.

a. For the most part, in vivo imaging within the brain has moved away from the use of anesthetics because it is well known these drugs dramatically alter network dynamics. The effects of urethane on the response properties is not controlled for or addressed.

Although calcium imaging can be done in awake mice in many parts of the brain, a notable exception is the caudal medulla, which includes the NTS. Brain motion artifacts in freely behaving mice are dramatic in this brain region, and we now include a supplementary video showing the extent of motion artifact in awake mice. Unlike most parts of the brain that are largely stable when the mouse is head-fixed, the NTS is partly located behind the skull. Routine neck movements in the awake mouse can cause large displacements (hundreds of microns) of the caudal medulla, shifting NTS neurons completely out of the field of view, as shown in the video. Before establishing the anesthetized preparation used in the manuscript, we attempted to image the NTS in awake mice using two different methods but could not effectively reduce motion artifacts. Indeed, the only other published medulla imaging preparation also used anesthetized mice (Lehnert et al., *Cell*, 2021). We add that simultaneous stimulation of multiple visceral organs in the same mouse also requires sophisticated surgeries and is infeasible in awake mice. Thus, we reasoned that calcium imaging in anesthetized mice would be a more realistic approach to investigate fundamental coding principles in this unexplored region of the brain, as was first done in the visual (Ohki et al., *Nature*, 2005), somatosensory (Svoboda et al., *Nature*, 1997; Stosiek et al., *PNAS*, 2003), gustatory (Chen et al., *Science*, 2011), olfactory (Charpak et al., *PNAS*, 2001), and auditory systems (Rothschild et al., *Nat. Neurosci.*, 2010). We believe the current manuscript reveals basic principles of circuit organization and provides a foundation for future studies in awake animals if they are one day technically achievable.

We also performed additional experiments to address potential effects of urethane on NTS responses. We additionally imaged NTS responses in mice anesthetized with isoflurane, an anesthetic with a distinct mechanism of action. We observed that NTS spatial maps and neuronal tuning were remarkably similar between urethane- and isoflurane-anesthetized mice, and present these findings as a new figure (Extended Data Fig. 1) in the manuscript.

b. Red-shifted calcium indicators are significantly less sensitive and have poorer temporal resolution than GFP-based sensors. Nuclear calcium flux in the nucleus is a fraction of that typically measured in the cytosol. Hence, the performance of the h2b-jrcamp indicator needs to be benchmarked against GCaMP6 (or later). Even better would be validation of their major conclusions with electrical recordings.

We performed additional experiments to compare the sensitivity of jRGECO1a to GCaMP6 as requested. We injected a mixture of AAVs encoding H2B-jRGECO1a and GCaMP6m into the NTS and compared the responses of these indicators to the same stimuli. H2B-jRGECO1a and GCaMP6m responses were highly correlated, with GCaMP6m responses modestly (1.9 times) higher than H2B-jRGECO1a responses. These data suggest that the calcium indicator we chose can faithfully and sensitively report intracellular calcium. These new findings are presented in Extended Data Fig. 1.

We also note several published findings about jRGECO1a and H2B-jRGECO1a consistent with our characterization. jRGECO1a is the most sensitive red-shifted calcium indicator developed to date. In cultured neurons, the original group (who engineered and characterized both jRGECO1a and GCaMP6) reported that “jRGECO1a response amplitudes and kinetics are comparable to GCaMP6f for brief trains of 1–10 action potentials” (Dana et al., *eLife*, 2016). When characterized *in vivo*, both the sensitivity and dynamic range of jRGECO1a are consistently higher than GCaMP6f in various preparations, except under extremely high-frequency electrostimulation (≥ 40 Hz), which is well above the normal firing rate of NTS neurons. The nuclear-localized H2B-jRGECO1a indicator used in our manuscript has recently been characterized by a separate group, who reported H2B-jRGECO1a to be 2.8-fold less sensitive than cytosolic jRGECO1a (Farhi et al., *J. Neurosci.*, 2019), yet still slightly more sensitive than GCaMP6f and approximately 4 times more sensitive than GCaMP5G (Chen et al., *Nature*, 2013).

c. The stimuli delivered are invasive and bear little resemblance to the types of physiological stimuli that the NTS neurons would be normally responding to. Can this imaging approach detect NTS responses to normal nutrient or food cues or to stimulation of specific vagal neuron types?

We agree that monitoring single neuron responses in the NTS during naturalistic feeding would be fascinating if NTS imaging were possible in freely behaving mice. Satiety is well established to be induced by both gut mechanosensation and chemosensation. Our surgical approaches allow us to deconstruct the complex signals that may arise from *ad libitum* feeding, and isolate responses due to force and particular chemicals across organs. Balloon distension and nutrient infusion are the standard procedures to selectively stimulate gut mechano- and chemoreceptors (Ichiki et al., *Nature*, 2022; Williams et al., *Cell*, 2016; Kim et al., *Nature*, 2020; Traub et al., *Neuroscience*, 1996; Willing and Berthoud, *Am. J. Physiol. Reg. I.*, 1997; Fraser et al., *Am. J. Physiol. Reg. I.*, 1995). Distension magnitudes and concentrations of applied chemicals were chosen to match physiological stimuli, and are similar to values that have been widely used in the field (Williams et al., *Cell*, 2016; Kim et al., *Nature*, 2020; Traub et al., *Neuroscience*, 1996; Willing and Berthoud, *Am. J. Physiol. Reg. I.*, 1997; Fraser et al., *Am. J. Physiol. Reg. I.*, 1995; Prescott et al., *Cell*, 2020; Smith and Hanamori, *J. Neurophysiol.*, 1991). Stomach volume is 370 μL in a comfortably full mouse but can reach as high as 550 μL (Bai et al., *Cell*, 2019; Kim et al., *Nature*, 2020; McConnell et al., *J. Pharm. Pharmacol.*, 2008). Thus, we tested the response to gastric distension volumes of 150, 300, 600, and 900 μL in our study. Intestinal volume after licking glucose solution is approximately 80 μL per 1 cm segment of intestine (Bai et al., *Cell*, 2019). We previously analyzed NTS responses to intestinal distensions of 90, 115, and 140 μL per 1 cm intestine; in the revised manuscript, we extended this range by adding a 50 μL

intestinal distension, and observed that NTS responses to 50 and 90 μ l distension were highly correlated (Extended Data Fig. 4). Similarly, nutrient concentrations were chosen to match levels in food, and intestinal infusion of salt (0.5 M NaCl) and glucose (0.5 M to 1M) have been frequently used to activate vagal chemosensory neurons (Ichiki et al., *Nature*, 2022; Williams et al., *Cell*, 2016; Tan et al., *Nature*, 2020). A 12 oz can of Coca Cola[®] contains 42 g (600 mM) sugar and glucose concentrations as high as 1.33 M are highly palatable to mice (Bai et al., *Cell*, 2019). Likewise, stimuli perfused into the larynx (like water, salt, acid) are well established to trigger airway protective reflexes that prevent aspiration events (Prescott et al., *Cell*, 2020; Smith and Hanamori, *J. Neurophysiol.*, 1991). Chemical and mechanical stimuli were carefully chosen to represent naturalistic stimuli encountered by vagal sensory neurons.

2. Work from the Liberles groups and others have already shown genetically distinct types of neurons have regionally specific projections in the NTS. The evidence provided in this study relies on tracing from the stomach using an AAV expressing synGFP. These new data are insufficiently detailed or quantified to support claims that the axonal organization does not predict the functional map. Co-labeling and using more selective markers on gut innervating vagal neurons that target different organs or the same organ with different ending types would significantly help.

We performed experiments as suggested, which led to important new observations that help clarify the relationship between vagal axon identity and NTS responsiveness. We imaged NTS responses in the context of visualized axonal projections from genetically defined vagal sensory neuron subtypes. In particular, we previously observed that GLP1R and GPR65 mark discrete populations of gut-innervating vagal sensory neurons that (1) predominantly function as mechanoreceptors and chemoreceptors respectively, and (2) display spatially discrete projections to the NTS. NTS axonal terminals of GLP1R and GPR65 neurons were visualized in the context of NTS imaging by injecting *Glp1r-ires-Cre* or *Gpr65-ires-Cre* mice with *AAV1-Flex-Synaptophysin-Gfp* in stomach and *AAV1-Syn-H2b-jRGECO1a* in the NTS. Interestingly, we observed that stomach stretch-responsive neurons were closer to axonal boutons from stomach GLP1R neurons than stomach GPR65 neurons, but that vagal axon position alone was still only partially predictive of NTS responses (Extended Data Fig. 8). These observations led us to refine our models in the manuscript. Taken together with our data involving inhibitory neurons, our findings are consistent with a role for both vagal axon targeting and local inhibition in shaping the spatial pattern of NTS responses.

3. The homogeneity of the evoked-calcium responses is curious. Based on the data, it appears that NTS neurons only about whether a specific organ is stimulation, irrespective of the type of stimulation. The responses seem to have very little dynamics or differences. Given the known cell type diversity of the vagal and NTS neurons, there seems to be something missing here? One possibility is the combination of the anesthesia, low resolution indicator and coarse sensory stimuli is providing highly filtered data that does not reflect the normal functional state of NTS neurons. Another concern is the representative image in Figure 1B. At baseline, most of the neurons seem already very bright and the few responding neuron are ones that are much dimmer at baseline. This

may be warning about unhealthy cells or reflect a poor dynamic range of the red-shift nuclear localized indicator?

This is another terrific point, and we performed new experiments into NTS response heterogeneity which led to a new important finding. In the original manuscript, mechanical stimuli were typically applied for a short time (8 seconds or 10 imaging frames), which was suboptimal to resolve the response kinetics of different neurons. Here, we extended the duration of mechanical distension of the stomach to 40 seconds (Fig. 1d and Extended Data Fig. 2), and observed two classes of responses by k-means clustering that can be described as rapidly adapting and slowly adapting, and were consistent across repeated stimulation. Response duration, measured by the full width at quarter maximum (FWQM), displayed a clear bimodal distribution corresponding to the two response classes. Rapidly adapting and slowly adapting neurons occupied slightly different locations in the NTS, but did not differ in response threshold or likelihood of responding to duodenum stretch. These data indicate that NTS neurons display heterogeneous response kinetics, and our imaging paradigm is capable of resolving these temporal differences.

The reviewer also correctly noted that there are some bright neurons in the representative images, and at least some bright neurons can indeed respond to stimuli. In the revised manuscript, we enlarged the insert in Fig. 1b to include a nearby bright neuron (baseline fluorescence ranked as top 11% of all neurons) that showed a positive response to stomach stretch, and below we also show two other bright neurons from the same mouse (baseline fluorescence ranked as top 5% and 6% of all neurons) responding to intestine distension. As discussed above, jRGECO1a generally has a better dynamic range than GCaMP6 in reporting on physiological levels (<40 Hz) of neuronal firing (Dana et al., eLife, 2016) and should not prevent response detection in these bright neurons. Moreover, we observed that many bright neurons occupied a remarkably stereotyped anterior location across experiments, and furthermore, we consistently saw that neurons in this anterior NTS region did not respond to any stimuli tested. We would expect injured cells to be more uniformly distributed across the imaging field, and think their location instead suggests that they may respond to stimulation of organs more rostral to the stomach, such as the esophagus. (Unfortunately, NTS imaging requires a ~30° downward head pitch which is not compatible with esophageal stimulation, as esophageal intubation then causes a lethal airway block.) We also attempted vagus nerve electrical stimulation to exclude neuronal death, but electrical stimulation induced field-of-view motion that precluded NTS imaging.

Reviewer Figure 1. Representative two-photon images of H2B-jRGECO1a fluorescence in the NTS at baseline and during duodenum stretch, transverse plane, R: rostral, M: medial, red arrows: responsive neurons, the percentiles of baseline fluorescence of Neuron 1 and 2 are 6% and 5%, respectively. scale bar: 100 μm .

4. According to the methods, the authors used many Cre lines that have been previously shown to mark NTS neurons with functional roles in taste, feeding etc but then, somewhat unusually, lump them all together in the analyses. Does this mean that every “cell type” previously described in the NTS has the same functional properties?

Thank you for noting this. We did indeed perform a large set of imaging experiments in which different subtypes of NTS neurons were genetically marked. We chose nine additional Cre lines which have been used in the literature to denote either functionally relevant and/or transcriptome-defined NTS neuron populations, including those marked in (1) *Th-Cre*, (2) *Cartpt-ires-Cre*, (3) *Calcr-ires-Cre*, (4) *Pdyn-ires-Cre*, (5) *Tac1-ires-Cre*, (6) *Penk-ires-Cre*, (7) *Gcg-Cre*, (8) *Sst-ires-Cre*, and (9) *Crrh2-ires-Cre* mice. We have now completed a full suite of imaging experiments in each line with associated data analysis, which allows us to make important new conclusions in the manuscript. First, we observed that cells marked in each line include neurons responsive to multiple stimuli. In general, a similar percentage of responding cells was observed in each line, although quantitative differences were noted in a few cases. It is clear from these findings that different visceral inputs engage a heterogeneous assortment of cell types, and that each of these Cre-defined cell types is recruited across multiple sensory representations. We now put forth the idea that the NTS is comprised of distinct spatially defined modules for particular sensory inputs, with each NTS module containing diverse and mostly overlapping cell types. Our data is reminiscent of the somatosensory/motor cortex and spinal cord, both of which contain diverse cell types that are horizontally distributed across cortical/spinal columns corresponding to distinct somatotopic areas (Haring et al, Nat Neuro 2018; Park et al, Nature Communications, 2021; Zhang et al, Nature 2021). Likewise, second-order neuron types of the olfactory bulb (mitral cells) share expression signatures, yet their responses are defined spatially by the glomerulus they innervate. We also note that some cell types are enriched in particular NTS modules, and in future studies it will be interesting to understand how each cell type acts in NTS circuit calculations to control physiology and behavior. The distributed representations of gastrointestinal cues across cell types nicely support recent findings that stimulating many different NTS cell types inhibit food intake (Cheng et al., Cell Metab., 2020; Aklan et al., Cell Metab., 2020; Roman et al., Nat. Commun., 2016; Roman et al., Neuroscience, 2017; Gaykema et al., J. Clin. Invest., 2017; D’Agostino et al., eLife, 2016; Jarvie and

Palmiter, Nat. Neurosci., 2017). We thank the reviewer for this suggestion, and think it has greatly improved the manuscript.

5. The use of *Vgat-Cre* to label inhibitory neurons with GFP is fine (extended data fig 2), but there are many types of inhibitory neurons known to have very different physiological and functional properties. Furthermore, looking at the GFP negative cells is insufficient- how efficient is the AAV viral labeling, presumably many *Vgat*-positive cells? The non-GFP cells will be a mixture of excitatory and inhibitory neurons.

We agree with the reviewer that AAV viral labeling may not capture all Cre-expressing NTS neurons. We performed additional experiments involving a genomic reporter which should more uniformly label inhibitory neurons. We crossed *Vgat-cre* with a GFP reporter mouse (*Rosa26-*Isl-Gfp-L10a**), and performed NTS imaging as described. The new data are remarkably consistent with previous data, did not change the conclusion that inhibitory and excitatory neurons are similarly tuned, and are provided in the revised manuscript (Extended Data Fig. 5).

We additionally examined responses of NTS neurons expressing somatostatin (SST), which marks one third of inhibitory neurons, as well as rarer excitatory neurons. SST inhibitory neurons were of particular interest as they were previously shown to provide broad inhibition of both excitatory and other inhibitory NTS neurons and to gate viscerosensory signal transmission (Thek et al., J. Neurosci., 2019). Like all other neuronal cell types examined in our manuscript, SST-expressing neurons were activated by all stimuli tested, and the tuning of SST neurons was similar to all inhibitory neurons marked in *Vgat-Cre* mice (Extended Data Fig. 5, 6).

6. In Fig 4, can the authors say something more about the circuit mechanism of inhibition? Do they believe that local inhibition from neurons that themselves respond to visceral stimuli is the predominant source of inhibition? Or do long range inputs play the role? If the former, do the authors see any evidence in their imaging data for inhibitory circuit motifs known to be involved in this kind of mapping e.g. lateral inhibition, center surround, etc.

We performed additional experiments to investigate the source of inhibition by using chemogenetic tools to activate local NTS inhibitory neurons. We crossed *Vgat-cre* with a GFP reporter mouse (*Rosa26-*Isl-Gfp-L10a**), injected AAVs encoding Cre-dependent hM3D and Cre-independent H2B-jRGECO1a into the NTS, and performed NTS imaging as described. We then recorded NTS responses to visceral stimulation before and after CNO administration in the same mouse. We observed that chemogenetic activation of NTS inhibitory neurons dramatically suppressed both the amplitude and number of neurons responsive to both stomach and duodenum stretch (Extended Data Fig. 9), consistent with the notion that local inhibition contributes to vagal input gating. Together with previous reports of broad local inhibition within the NTS (Thek et al., J. Neurosci., 2019), we believe that local inhibitory neurons are likely a predominant source of inhibition, but we cannot rule out a

contribution from long-range inhibitory inputs as well. The inhibition among parallel pathways we described is a typical example of lateral inhibition that sharpens the tuning of excitation, and we revised the text to make it clear.

7. Homunculus means “tiny human”, clearly there are no small people in the mouse brainstem. More seriously, this study shows there is coarse topographical representation of the digestive tract in the NTS and so the title, abstract and introduction should be toned down.

We removed the word homunculus from the title and abstract; we did leave one reference to it in the main body of the text as we think it is helpful to draw comparisons to the well-known 'Penfield homunculus' for external somatosensory inputs.

Minor comments:

1. Figure 1e. Please show more baseline before stimulation

The revised figure now shows a longer baseline period.

2. There is a general lack of scale bars or bars without labels in the legends (e.g. figures 1g, 2b, 4).

We added scale bars and associated labels in the legends.

3. Fig 4, it would be helpful if they could compare other organs in addition to leg pinch, (ie: lung, diaphragm, heart) to get a better sense of the boundaries

Leg pinch activates neurons located in the dorsal column nuclei (DCN) outside the NTS, and the boundary between the DCN and NTS is well documented from histological studies (Loewy and Burton, 1978). We note that some cardiovascular and respiratory stimuli (like airway stretch or the baroreceptor reflex) could not be applied for technical reasons during in vivo imaging such as stimulation-induced motion, 3D tissue deformation for long-term imaging, and brain hemorrhage due to elevated blood pressure during surgery.

4. The two proposed models of spatial organ representation (continuous vs discrete) seem an oversimplification. The authors focus on a few organs and there remains a possibility that gradients beside the one depicted in their example cartoon exist – what would happen if other organs, position in the body axis, modality, functional role or physiological state were also considered?

We performed some new experiments and edited the text in response to this comment. We previously observed that responses to stretch of the stomach and intestine (with distensions separated by only a few mm) evoked NTS responses that were more spatially segregated than stretch of rather remote regions of the intestine. We did new experiments involving distension of

different regions of the stomach (Extended Data Fig. 7) which provided similar conclusions. We can safely say that spatial organization is not completely continuous, with organ identity being more salient than absolute position. That said, we agree with the reviewer that our conclusions about continuous vs discrete NTS maps reflect two extreme models, and have toned our conclusions accordingly.

5. Although reference numbers are limited, a deeper overview of what is known about the NTS from electrical recordings and other approaches should be added, perhaps at the expense of the comparisons to other sensory systems..

We added citations to prior NTS studies throughout the manuscript.

Referee #3 (Remarks to the Author):

The manuscript by Ran et al., aims to deduce decoding parameters of visceral interoception in the dorsal brainstem (nucleus of the solitary tract, NTS) with a focus on anatomical location. By meticulous and elegant real-time in vivo cellular Ca⁺ recordings after selective stimuli at distinct locations along the GI-tract, it is discovered that neurons responding to stimuli from different locations are spatially segregated. Most neurons uniquely respond to stimuli from a single GI-region, with little evidence of sub-patterning within major GI-region. In contrast, the same neurons respond to different types of stimuli evoked from the same region. By comparing sensory vagal innervation and responding neuron location a mismatch was suggested. Last, inhibitory actions between NTS neurons were found pivotal to establish specific response patterns. NTS neurons activated by duodenal stimuli were silenced by inhibitory neurons when the stomach was probed and vice versa, implying that cross-inhibitory mechanisms shape final responses.

While the stringency, advanced in vivo methodologies, overall presentation, intent and findings of this study were highly appreciated, the overall conceptual advance remains limited. The focus on spatial pattern (also indicated by the title) is unfortunate as this is the least novel/interesting aspect of the study (at least without a stringent correlation to molecular or innervation patterns). In contrast, the two last sections on the possible mismatch between sensory neuron innervation and responding neurons, and the important role of inhibitory networks to shape responses, have potential strong conceptual novelty and impact on the field. Unfortunately, these parts were performed with less precision and left at a preliminary stage. Additional experiments to elevate these parts along with alignment of molecular information would be needed.

We thank the reviewer for appreciating the quality, novelty, and impact of our study. We are glad that the reviewer found discoveries related to inhibitory networks particularly interesting and we performed additional experiments to strengthen these conclusions. We also performed a large-scale investigation to align genetic markers with response properties, and offer our perspective on the significance of the sensory map, thanks to the reviewer's insightful comment on this topic. We believe the manuscript has significantly improved as a result.

Major Concerns and Questions:

1) The authors indicate a possible mismatch between the location of responding neurons and the innervating vagal sensory axons – this was also stated as motivation for the study in the introduction. However, the hypothesis is only tested in one experiment (Extended Data Fig 7) where stomach neurites are related to the responders of other regions. There are two problems with this part – the limited scope of the experiment, as well as the precision to which it is carried out. To gain clarity of the extent of mismatch it would be important to test all vagal neurite connections with all responders, i.e. systematically label connections from larynx, duodenum, jejunum and cecum and assess the relation to responders of all these regions. The analysis also has to be performed with greater precision. The stomach innervation is claimed to be regionally patterned, yet this pattern is not presented (AP-DV-ML) either in itself or in relation to the responding neurons. Moreover, distance to closest bouton is not a very precise measure of connections. One weak bouton (possibly without a corresponding postsynaptic connection) would be equally counted as 10 strong boutons if they are at a similar distance. It would be of higher value to count the numbers of strong boutons within a set radius around each responding neuron. It would also be valuable to define the area of innervation and count how many of the responding neurons that fit that area, or correlate the area of responders to the overall area of innervating boutons. It is moreover hard to correlate the representative pictures (Ext Data Fig b) with the actual statistical table (Ext Data Fig 7c) – it appears as if many responders would be several cell diameters away from the closest bouton, while others are touched by boutons - the statistics however suggests that most neurons are connected to boutons (error bar very small). Furthermore, the conclusion that dendritic arborization of brainstem neurons allows them to respond at a distance is not verified or quantified anywhere in the paper (on the contrary the statistics indicate that all neuron cell bodies are relatively close to boutons representing the vagal innervation of a given region). To assess spatial association between dendrites and innervating axons a fluorescent response marker that marks the full morphology of the cells could be used. The question thus remains unanswered, as no attempts of comparing the location of responding neuronal dendrites and innervating axons are made.

Thank you for these thoughts- they have inspired several additional experiments which have allowed us to refine our conclusions in important ways. We fully agree with the reviewer that understanding the link between the positions of vagal axons and responding NTS neurons is important. As suggested, we extended our analyses to label vagal neurons that arrive from additional organs, and additionally performed viral injections in the larynx, stomach, and duodenum. We also provide better depictions of the spatial distribution of organ-defined NTS axons. The reviewer also suggests several excellent ways to quantify the relationship between the positions of vagal boutons and responding NTS neurons. We performed analyses suggested for experiments involving viral injection in peripheral organs, and the conclusions were similar; new data are presented in Extended Data Fig. 8. Based on these observations, the patterns of axonal projections, as historically visualized by peripheral dye injection in organs, have limited predictive power over NTS responses. Overall, our conclusions from axon tracing were that other mechanisms could contribute to NTS response patterning, which was substantiated by our observations indicating a key role for local inhibitory circuits.

Furthermore, a limitation of prior experiments involving dye injection is that multiple terminal types (including various mechanoreceptors, chemoreceptors, and osmoreceptors) may be labeled. To achieve more precision in these experiments, we also imaged NTS responses in the context of visualized axonal projections from genetically defined vagal sensory neuron subtypes. In particular, we previously observed that GLP1R and GPR65 mark discrete populations of gut-innervating vagal sensory neurons that (1) predominantly function as mechanoreceptors and chemoreceptors respectively, and (2) display spatially discrete NTS projections. NTS axonal terminals of GLP1R and GPR65 neurons were visualized in the context of NTS imaging by injecting *Glp1r-ires-Cre* or *Gpr65-ires-Cre* mice with *AAV1-Cag-Flex-Synaptophysin-Gfp* in stomach and *AAV1-Syn-H2b-jRGECO1a* in the NTS. Interestingly, we observed that stomach stretch responsive neurons were closer to axonal boutons from stomach GLP1R neurons than stomach GPR65 neurons, but that vagal axon position alone was still only partially predictive of NTS responses (Extended Data Fig. 8I). These observations led us to refine our models in the manuscript. Taken together with our data involving inhibitory neurons, our findings are consistent with a role for both vagal axon targeting and local inhibition in shaping the spatial pattern of NTS responses.

We thank the reviewer for these helpful comments as refining the relationship between vagal axons and NTS responding neurons has significantly improved the manuscript.

2) The spatial segregation of responding neurons is presented as a surprising finding and contrasted to the distributed pattern of transcriptionally distinct vagal sensory neurons. However, this is the generic difference between the central and peripheral nervous system, stemming from the very different ways in which these systems are formed. While the CNS is generated from spatially distinct patterned stem cells that differentiate into stereotypically arranged neuron types, ultimately providing the CNS a clear functional anatomy, the PNS is generated from neural crest cells that have lost the intricate layers of spatial identity and therefore form ganglionic structures consisting of intermingled subtypes. Patterned versus distributed neuron patterns are known major distinguishing features of the CNS and PNS - thus the highlighting of this difference as something novel should be avoided.

We agree with the reviewer that the CNS in general is more structured than the PNS, and have removed the statement "with an ordered brainstem map arising from distributed representations in peripheral ganglia" from the abstract. It is important to note that spatial maps are not present in every CNS sensory region, and there are several notable exceptions. For example, odor representations are highly stereotyped in the olfactory bulb, but spatial information is discarded in the primary olfactory (piriform) cortex, where odors are represented by random and spatially distributed neuronal ensembles (Stettler and Axel. *Neuron*, 2009; Schaffer et al., *Neuron*, 2018). Similarly, in the rodent primary visual cortex, neurons encoding different stimulus orientations are intermingled in a salt-and-pepper manner (despite being neatly arranged in pinwheel-like structures in the visual cortex of some higher mammals, Ohki et al., *Nature*, 2005; Ohki et al., *Nature*, 2006). Previous studies demonstrating the existence of central maps have been highly valued in the sensory neuroscience field (for example: Garg et al., *Science*, 2019; Chen et al., *Science*, 2011; Feinberg and

Meister, *Nature*, 2014; Rothschild et al., *Nat. Neurosci.*, 2010), and it has been hotly debated if a topographic map exists in the NTS. Our study is the first one to provide a definitive answer to this important and long-standing question.

Furthermore, while sensory maps exist in several brain regions, such maps can be topographically organized to encode any of a variety of stimulus features, not necessarily stimulus location. For example, the primary auditory cortex is organized according to sound frequency, while sound location is more selectively encoded in other auditory regions (Heijden et al., *Nat. Rev. Neurosci.*, 2019). Similarly, the visual system is topographically organized according to increasingly complex stimulus features as information ascends- from light position in early visual areas, to orientation, direction, or color in primary visual cortex (Blasdel and Salama, *Nature*, 1986; Bonhoeffer et al, *Nature*, 1991; Maldonado et al, *Science*, 1997; Ohki et al, *Nature*, 2005; Ohki et al, *Nature*, 2006; Garg et al., *Science*, 2019), to object features in higher visual cortices (Bao et al., *Nature*, 2020). Understanding what sensory features govern the organization of each sensory relay, how maps are transformed, and key map features that arise or dissipate in each brain region is essential for understanding the role of that brain region in circuit computations, and more generally how the brain encodes information. Prior to our study, it has been unclear what sensory features govern the organization of the visceral NTS. For example, satiety can be induced by several signals originated from different organs, including stomach stretch, intestinal stretch, and intestinal nutrients (Phillips and Powley, *Am. J. Physiol. Reg. I.*, 1996; Powley and Phillips, *Physiol. Behav.*, 2004; Bai et al., *Cell*, 2019). By contrast, high-volume distension of the stomach and other organs can change the evoked perception to discomfort or pain. Our work shows salient features of NTS organization that include a spatial map where neuronal responses are more tightly linked to the stimulated organ than other features like evoked perception, absolute position in space, or modality. We believe these conclusions are conceptually significant for understanding how viscerosensory information is encoded by the brain.

3) The spatial organization of NTS neurons is claimed to constitute an independent feature, separated from intrinsic gene expression programs - however this is never tested in the manuscript. On the contrary it would seem likely that the response pattern is genetically coded, ie absence/presence of specific receptors governing whether they would respond or not, absence/presence of specific adhesion/molecules that governs contact with other neuron types (for example the inhibitory neurons). It is thus paramount to align responding cells to their gene expression profiles. This could be achieved by in situ sequencing (using probes selected for known neuron subtypes), multiplexed IHC or RNAscope. Ideally the experiment would use spatial transcriptomics which is to be released in 2022 from 10xgenomics, allowing full genome sequencing of single cells in situ. Without information of cell identities/gene expression profiles, the presentation of spatial patterning provides little impact to the field. Molecular information of responding cells would constitute an important resource to further understanding the finetuning of each modality/organ sensation and the connectivity of NTS to other regions of the brain.

Thank you for this comment. We have now performed a large set of imaging experiments in which different subtypes of NTS neurons were genetically marked, in order to provide a link between cell identity and response property as requested. We chose nine additional Cre lines which have been used in the literature to denote either functionally relevant and/or transcriptome-defined NTS neuron populations, including those marked in (1) *Th-Cre*, (2) *Cartpt-ires-Cre*, (3) *Calcr-ires-Cre*, (4) *Pdyn-ires-Cre*, (5) *Tac1-ires-Cre*, (6) *Penk-ires-Cre*, (7) *Gcg-Cre*, (8) *Sst-ires-Cre*, and (9) *Crhr2-ires-Cre* mice. We have now completed a full suite of imaging experiments in each line with associated data analysis, which allows us to make important new conclusions in the manuscript. First, we observed that cells marked in each line include neurons responsive to multiple stimuli. In general, a similar percentage of responding cells was observed in each line, although quantitative differences were noted in a few cases. It is clear from these findings that different visceral inputs engage a heterogeneous assortment of cell types, and that each of these Cre-defined cell types is recruited across multiple sensory representations. We now put forth the idea that the NTS is comprised of distinct spatially defined modules for particular sensory inputs, with each NTS module containing diverse and mostly overlapping cell types. Our data is reminiscent of the somatosensory/motor cortex and spinal cord, both of which contain diverse cell types that are horizontally distributed across cortical/spinal columns corresponding to distinct somatotopic areas (Haring et al, Nat Neuro, 2018; Park et al, Nat Comm 2021; Zhang et al, Nature 2021). Likewise, second-order neuron types of the olfactory bulb (mitral cells) share expression signatures, yet their responses are defined spatially by the glomerulus they innervate. We also note that some cell types are enriched in particular NTS modules, and in future studies it will be interesting to understand how each cell type acts in NTS circuit calculations to control physiology and behavior. The distributed representations of gastrointestinal cues across cell types nicely support recent findings that stimulating many different NTS cell types inhibits food intake (Cheng et al., Cell Metab., 2020; Aklan et al., Cell Metab., 2020; Roman et al., Nat. Commun., 2016; Roman et al., Neuroscience, 2017; Gaykema et al., J. Clin. Invest., 2017; D'Agostino et al., eLife, 2016; Jarvie and Palmiter, Nat. Neurosci., 2017). We thank the reviewer for this suggestion, and think it has greatly improved the manuscript.

We note that we chose to use Cre lines to link response and cell identity over a few other approaches that cannot yet be applied to the NTS. Studies involving superimposition of multiplexed *in situ* hybridization and calcium imaging have appeared very recently in studies of the somatosensory cortex, hypothalamus, and trigeminal ganglion (Condylis et al., Science, 2022; Xu et al., Science, 2020; von Buchholtz et al., Neuron, 2021). Such experiments are difficult even in regions that can be routinely imaged, given the challenges of aligning *in vivo* and *ex vivo* images, and these approaches involved analysis of 6-16 genes rather than whole-cell transcriptomes. Furthermore, the NTS poses several additional technical challenges related to registration of nuclear jRGECO fluorescence and cytosolic mRNA labeling in such a deformable brain region densely packed with small neurons. These issues make registration approaches impractical for the NTS, so we instead relied on Cre lines to link cell identity to response property, which is an established approach and here generated important new insights into NTS cellular organization.

4) Both excitatory and inhibitory neurons were activated as response to all stimuli. It would be important to understand whether the excitatory and inhibitory neurons are segregated in space.

Inhibitory neurons were visualized using *Vgat-Cre; Rosa26-IsI-Gfp-L10a* mice, and we observed consistent labeling throughout the entire NTS, with perhaps a slight enrichment in medial NTS. Furthermore, inhibitory and excitatory neurons were highly intermingled throughout the NTS. We now present a figure depicting the full distribution of inhibitory neurons (Extended Data Fig. 5), which is consistent with previous publications that reported highly dispersed inhibitory neurons throughout the NTS using immunohistochemistry and in situ hybridization methods (Fong et al., JCN, 2005; Stornetta and Guyenet, JCN, 1999).

5) The observed role of inhibitory neurons to shaping the response patterns is very interesting and could form a central part of the manuscript. It will be important to expand and investigate this phenomenon in more detail. For instance, the cross-inhibition could also be investigated between all possible pairs of focus in the study, not only stomach/duodenum, to understand how general/specific the mechanism is. The mechanism of cross-inhibition indicate that the initial response of inhibitory neurons is more specific and helps to shape the response of excitatory neurons - thus greater amounts of excitatory neurons are probably able to respond by default. Please investigate whether neuron that are de-inhibited as a result of GABA-receptor blockade indeed are excitatory neurons and not inhibitory neurons. Linking back to the innervation pattern from distinct regions – assessing the spatial relation between the innervating axons of one region and only the inhibitory responders of each region could integrate different parts of this study better and reveal further insights into the decoding mechanisms of interoceptive stimuli.

Thank you for your suggestions to expand studies on cross-inhibition in the NTS; we performed several new experiments in response.

First, we extended cross-inhibition studies by testing dual stimulation of stomach/larynx and stomach/jejunum. Previously, we observed strong cross-inhibition of some neurons responsive to stomach and duodenum distension by dual stomach/duodenum distension. Interestingly, in new data, we also observed similarly effective cross-inhibition between jejunum and stomach stimulation, but a substantially attenuated cross-inhibition between larynx and stomach stimulation. These findings suggest that inhibitory neurons display at least some selectivity in cross-inhibition, as stomach-stretch responsive inhibitory neurons more effectively suppressed duodenum and jejunum responses than larynx responses (Extended Data Fig. 10).

Second, we performed additional experiments to investigate whether de-inhibited neurons were predominantly excitatory or inhibitory neurons by performing NTS imaging in *Vgat-Cre; IsI-Gfp-L10a*

mice. As expected, we observed that both excitatory and inhibitory neurons were sensitive to disinhibition (Extended Data Fig. 9). We note that neither cross inhibition nor the sharpening of neuronal tuning requires inhibitory neurons to be more selectively tuned than excitatory neurons. Rather, such functions can be achieved by inhibition that is more broadly tuned or co-tuned with excitation (Isaacson and Scanziani, *Neuron*, 2011; Wehr and Zador, *Nature*, 2003; Kato et al., *Neuron*, 2017). For example, in the primary visual cortex, inhibitory neurons that sharpen the tuning of excitatory neurons are actually more broadly tuned (Kerlin et al., *Neuron*, 2010; Liu et al., *Neuron*, 2011). In the olfactory bulb, both inhibitory granule cells and parvalbumin-expressing cells mediate lateral inhibition, and both cell types are more broadly tuned than excitatory neurons (Kato et al., *Neuron*, 2013; Miyamichi et al., *Neuron*, 2013; Tan et al., *Neuron*, 2010). In the auditory cortex, somatostatin-expressing neurons that mediate lateral inhibition are also more broadly tuned (Kato et al., *Neuron*, 2017). We also note that previous slice electrophysiology experiments revealed that NTS inhibitory neurons act broadly, inhibiting not only excitatory neurons but also extensively inhibiting other inhibitory neurons (Thek et al., *J. Neurosci.*, 2019).

Third, we now provide positional information about cross-inhibited neurons. Neurons prone to inhibition are intermingled with other neurons responsive to the same stimulus, without apparent positional bias (Extended Data Fig. 10). Since they occupy similar positions, excitatory neurons, inhibitory neurons, and neurons sensitive to cross inhibition are also similarly located near vagal axons.

6) Sections shown in Fig. 3 are confusing in relation to borders and axis, it would be helpful to include schematic drawings of the anatomical region and indicate DV and AP locations in each picture.

We have now clarified the NTS border and axes by adding a schematic on the images, and also labeled adjacent nuclei in Fig. 3. We clarified that all images in figure panels display the same orientation.

Minor issues

1) Fig 3b How do these graphs correspond to the anterior border of the area postrema?

We now depict the location of the area postrema in all images of Fig. 3a. We note that the area postrema is only clearly discernible at superficial depths in some mice (and not present at deeper layers), so the position of the area postrema cannot be precisely indicated in Fig. 3b. In superficial layers, the anterior border of the area postrema is typically near the X axis (approximate coordinates: $x \sim 200-250 \mu\text{m}$, $y \sim 0 \mu\text{m}$).

2) Fig 3g Does the third graph compare duodenum to jejunum site 2 (not stomach to jejunum site 2 as indicated in the figure)?

Yes- thank you for catching this typo; we corrected it in the revision.

3) Figure Legend 3 page 37; row 15. “f responses” should probably be “g responses”.

Thank you for noting- this has been corrected too.

4) The methods indicate that Pdyn, Sst, Penk, Carpt and Th transgenic animals are used in the study, however they are nowhere to be found in figures nor results part. This could indicate that attempts to characterize the NTS neurons molecularly has been made and a part of the manuscript is missing, alternatively these experiments were removed at a late stage?

The reviewer is correct- we have now imaged and analyzed an additional 43 mice to complete the data set for 9 different Cre mice, as detailed in our response to point 3 above.

References

Ackels, T., Erskine, A., Dasgupta, D., Marin, A.C., Warner, T.P.A., Tootoonian, S., Fukunaga, I., Harris, J.J., and Schaefer, A.T. (2021). Fast odour dynamics are encoded in the olfactory system and guide behaviour. *Nature* 593, 558-563.

Akhan, I., Sayar Atasoy, N., Yavuz, Y., Ates, T., Coban, I., Koksalar, F., Filiz, G., Topcu, I.C., Oncul, M., Dilsiz, P., et al. (2020). NTS Catecholamine Neurons Mediate Hypoglycemic Hunger via Medial Hypothalamic Feeding Pathways. *Cell Metab* 31, 313-326 e315.

Altschuler, S.M., Bao, X.M., Bieger, D., Hopkins, D.A., and Miselis, R.R. (1989). Viscerotopic representation of the upper alimentary tract in the rat: sensory ganglia and nuclei of the solitary and spinal trigeminal tracts. *J Comp Neurol* 283, 248-268.

Andresen, M.C., Fawley, J.A., and Hofmann, M.E. (2012). Peptide and lipid modulation of glutamatergic afferent synaptic transmission in the solitary tract nucleus. *Frontiers in neuroscience* 6, 191.

Bai, L., Mesgarzadeh, S., Ramesh, K.S., Huey, E.L., Liu, Y., Gray, L.A., Aitken, T.J., Chen, Y., Beutler, L.R., Ahn, J.S., et al. (2019). Genetic Identification of Vagal Sensory Neurons That Control Feeding. *Cell* 179, 1129-1143 e1123.

Bao, P., She, L., McGill, M., and Tsao, D.Y. (2020). A map of object space in primate inferotemporal cortex. *Nature* 583, 103-108.

Blasdel, G.G., and Salama, G. (1986). Voltage-sensitive dyes reveal a modular organization in monkey striate cortex. *Nature* 321, 579-585.

Bonhoeffer, T., and Grinvald, A. (1991). Iso-orientation domains in cat visual cortex are arranged in pinwheel-like patterns. *Nature* 353, 429-431.

Charpak, S., Mertz, J., Beaurepaire, E., Moreaux, L., and Delaney, K. (2001). Odor-evoked calcium signals in dendrites of rat mitral cells. *Proc Natl Acad Sci U S A* 98, 1230-1234.

Chen, T.W., Wardill, T.J., Sun, Y., Pulver, S.R., Renninger, S.L., Baohan, A., Schreiter, E.R., Kerr, R.A., Orger, M.B., Jayaraman, V., et al. (2013). Ultrasensitive fluorescent proteins for imaging neuronal activity. *Nature* 499, 295-300.

Chen, X., Gabitto, M., Peng, Y., Ryba, N.J., and Zuker, C.S. (2011). A gustotopic map of taste qualities in the mammalian brain. *Science* 333, 1262-1266.

Cheng, W., Gonzalez, I., Pan, W., Tsang, A.H., Adams, J., Ndoka, E., Gordian, D., Khoury, B., Roelofs, K., Evers, S.S., et al. (2020). Calcitonin Receptor Neurons in the Mouse Nucleus Tractus Solitarius Control Energy Balance via the Non-aversive Suppression of Feeding. *Cell Metab* 31, 301-312 e305.

Condylis, C., Ghanbari, A., Manjrekar, N., Bistrong, K., Yao, S., Yao, Z., Nguyen, T.N., Zeng, H., Tasic, B., and Chen, J.L. (2022). Dense functional and molecular readout of a circuit hub in sensory cortex. *Science* 375, eabl5981.

D'Agostino, G., Lyons, D.J., Cristiano, C., Burke, L.K., Madara, J.C., Campbell, J.N., Garcia, A.P., Land, B.B., Lowell, B.B., Dileone, R.J., et al. (2016). Appetite controlled by a cholecystokinin nucleus of the solitary tract to hypothalamus neurocircuit. *Elife* 5.

Dana, H., Mohar, B., Sun, Y., Narayan, S., Gordus, A., Hasseman, J.P., Tsegaye, G., Holt, G.T., Hu, A., Walpita, D., et al. (2016). Sensitive red protein calcium indicators for imaging neural activity. *Elife* 5.

Dombeck, D.A., Harvey, C.D., Tian, L., Looger, L.L., and Tank, D.W. (2010). Functional imaging of hippocampal place cells at cellular resolution during virtual navigation. *Nat Neurosci* 13, 1433-1440.

Farhi, S.L., Parot, V.J., Grama, A., Yamagata, M., Abdelfattah, A.S., Adam, Y., Lou, S., Kim, J.J., Campbell, R.E., Cox, D.D., et al. (2019). Wide-Area All-Optical Neurophysiology in Acute Brain Slices. *J Neurosci* 39, 4889-4908.

Feinberg, E.H., and Meister, M. (2015). Orientation columns in the mouse superior colliculus. *Nature* 519, 229-232.

Fong, A.Y., Stornetta, R.L., Foley, C.M., and Potts, J.T. (2005). Immunohistochemical localization of GAD67-expressing neurons and processes in the rat brainstem: subregional distribution in the nucleus tractus solitarius. *J Comp Neurol* 493, 274-290.

Fustinana, M.S., Eichlisberger, T., Bouwmeester, T., Bitterman, Y., and Luthi, A. (2021). State-dependent encoding of exploratory behaviour in the amygdala. *Nature* 592, 267-271.

Garg, A.K., Li, P., Rashid, M.S., and Callaway, E.M. (2019). Color and orientation are jointly coded and spatially organized in primate primary visual cortex. *Science* 364, 1275-1279.

Gaykema, R.P., Newmyer, B.A., Ottolini, M., Raje, V., Warthen, D.M., Lambeth, P.S., Niccum, M., Yao, T., Huang, Y., Schulman, I.G., et al. (2017). Activation of murine pre-proglucagon-producing neurons reduces food intake and body weight. *J Clin Invest* 127, 1031-1045.

Ghitani, N., Barik, A., Szczot, M., Thompson, J.H., Li, C., Le Pichon, C.E., Krashes, M.J., and Chesler, A.T. (2017). Specialized Mechanosensory Nociceptors Mediating Rapid Responses to Hair Pull. *Neuron* 95, 944-954 e944.

Haring, M., Zeisel, A., Hochgerner, H., Rinwa, P., Jakobsson, J.E.T., Lonnerberg, P., La Manno, G., Sharma, N., Borgius, L., Kiehn, O., et al. (2018). Neuronal atlas of the dorsal horn defines its architecture and links sensory input to transcriptional cell types. *Nat Neurosci* 21, 869-880.

Isaacson, J.S., and Scanziani, M. (2011). How inhibition shapes cortical activity. *Neuron* 72, 231-243.

Jarvie, B.C., and Palmiter, R.D. (2017). HSD2 neurons in the hindbrain drive sodium appetite. *Nat Neurosci* 20, 167-169.

Kato, H.K., Asinof, S.K., and Isaacson, J.S. (2017). Network-Level Control of Frequency Tuning in Auditory Cortex. *Neuron* 95, 412-423 e414.

Kato, H.K., Gillet, S.N., Peters, A.J., Isaacson, J.S., and Komiyama, T. (2013). Parvalbumin-expressing interneurons linearly control olfactory bulb output. *Neuron* 80, 1218-1231.

Katz, D.M., and Karten, H.J. (1983). Visceral representation within the nucleus of the tractus solitarius in the pigeon, *Columba livia*. *J Comp Neurol* 218, 42-73.

Kerlin, A.M., Andermann, M.L., Berezovskii, V.K., and Reid, R.C. (2010). Broadly tuned response properties of diverse inhibitory neuron subtypes in mouse visual cortex. *Neuron* 67, 858-871.

Lehnert, B.P., Santiago, C., Huey, E.L., Emanuel, A.J., Renauld, S., Africawala, N., Alkisar, I., Zheng, Y., Bai, L., Koutsioumpa, C., et al. (2021). Mechanoreceptor synapses in the brainstem shape the central representation of touch. *Cell* 184, 5608-5621 e5618.

Liang, L., Fratzl, A., Goldey, G., Ramesh, R.N., Sugden, A.U., Morgan, J.L., Chen, C., and Andermann, M.L. (2018). A Fine-Scale Functional Logic to Convergence from Retina to Thalamus. *Cell* 173, 1343-1355 e1324.

Liu, B.H., Li, Y.T., Ma, W.P., Pan, C.J., Zhang, L.I., and Tao, H.W. (2011). Broad inhibition sharpens orientation selectivity by expanding input dynamic range in mouse simple cells. *Neuron* 71, 542-554.

Loewy, A.D., and Burton, H. (1978). Nuclei of the solitary tract: efferent projections to the lower brain stem and spinal cord of the cat. *J Comp Neurol* 181, 421-449.

Llewellyn-Smith, I.J.V., A. J. M. (2011). Central regulation of autonomic functions (Oxford University Press).

Loewy, A.D., and Burton, H. (1978). Nuclei of the solitary tract: efferent projections to the lower brain stem and spinal cord of the cat. *J Comp Neurol* 181, 421-449.

Ludwig, M.Q., Cheng, W., Gordian, D., Lee, J., Paulsen, S.J., Hansen, S.N., Egerod, K.L., Barkholt, P., Rhodes, C.J., Secher, A., et al. (2021). A genetic map of the mouse dorsal vagal complex and its role in obesity. *Nat Metab* 3, 530-545.

Maldonado, P.E., Godecke, I., Gray, C.M., and Bonhoeffer, T. (1997). Orientation selectivity in pinwheel centers in cat striate cortex. *Science* 276, 1551-1555.

Mann, K., Deny, S., Ganguli, S., and Clandinin, T.R. (2021). Coupling of activity, metabolism and behaviour across the *Drosophila* brain. *Nature* 593, 244-248.

McDougall, S.J., Peters, J.H., and Andresen, M.C. (2009). Convergence of cranial visceral afferents within the solitary tract nucleus. *J Neurosci* 29, 12886-12895.

Miyamichi, K., Shlomai-Fuchs, Y., Shu, M., Weissbourd, B.C., Luo, L., and Mizrahi, A. (2013). Dissecting local circuits: parvalbumin interneurons underlie broad feedback control of olfactory bulb output. *Neuron* 80, 1232-1245.

Ohki, K., Chung, S., Ch'ng, Y.H., Kara, P., and Reid, R.C. (2005). Functional imaging with cellular resolution reveals precise micro-architecture in visual cortex. *Nature* 433, 597-603.

Ohki, K., Chung, S., Kara, P., Hubener, M., Bonhoeffer, T., and Reid, R.C. (2006). Highly ordered arrangement of single neurons in orientation pinwheels. *Nature* 442, 925-928.

Park, J., Choi, W., Tiesmeyer, S., Long, B., Borm, L.E., Garren, E., Nguyen, T.N., Tasic, B., Codeluppi, S., Graf, T., et al. (2021). Author Correction: Cell segmentation-free inference of cell types from in situ transcriptomics data. *Nature communications* 12, 4103.

Paton, J.F. (1999). The Sharpey-Schafer prize lecture: nucleus tractus solitarii: integrating structures. *Exp Physiol* 84, 815-833.

Peters, A.J., Fabre, J.M.J., Steinmetz, N.A., Harris, K.D., and Carandini, M. (2021). Striatal activity topographically reflects cortical activity. *Nature* 591, 420-425.

Phillips, R.J., and Powley, T.L. (1996). Gastric volume rather than nutrient content inhibits food intake. *Am J Physiol* 271, R766-769.

Powley, T.L., and Phillips, R.J. (2004). Gastric satiation is volumetric, intestinal satiation is nutritive. *Physiol Behav* 82, 69-74.

Reinert, S., Hubener, M., Bonhoeffer, T., and Goltstein, P.M. (2021). Mouse prefrontal cortex represents learned rules for categorization. *Nature* 593, 411-417.

Rinaman, L., Card, J.P., Schwaber, J.S., and Miselis, R.R. (1989). Ultrastructural demonstration of a gastric monosynaptic vagal circuit in the nucleus of the solitary tract in rat. *J Neurosci* 9, 1985-1996.

Roman, C.W., Derkach, V.A., and Palmiter, R.D. (2016). Genetically and functionally defined NTS to PBN brain circuits mediating anorexia. *Nature communications* 7, 11905.

Roman, C.W., Sloat, S.R., and Palmiter, R.D. (2017). A tale of two circuits: CCK(NTS) neuron stimulation controls appetite and induces opposing motivational states by projections to distinct brain regions. *Neuroscience* 358, 316-324.

Rothschild, G., Nelken, I., and Mizrahi, A. (2010). Functional organization and population dynamics in the mouse primary auditory cortex. *Nat Neurosci* 13, 353-360.

Ruder, L., Schina, R., Kanodia, H., Valencia-Garcia, S., Pivetta, C., and Arber, S. (2021). A functional map for diverse forelimb actions within brainstem circuitry. *Nature* 590, 445-450.

Schaffer, E.S., Stettler, D.D., Kato, D., Choi, G.B., Axel, R., and Abbott, L.F. (2018). Odor Perception on the Two Sides of the Brain: Consistency Despite Randomness. *Neuron* 98, 736-742 e733.

Shapiro, R.E., and Miselis, R.R. (1985). The central organization of the vagus nerve innervating the stomach of the rat. *J Comp Neurol* 238, 473-488.

Song, A., Charles, A.S., Koay, S.A., Gauthier, J.L., Thiberge, S.Y., Pillow, J.W., and Tank, D.W. (2017). Volumetric two-photon imaging of neurons using stereoscopy (vTwINS). *Nat Methods* 14, 420-426.

Stettler, D.D., and Axel, R. (2009). Representations of odor in the piriform cortex. *Neuron* 63, 854-864.

Stornetta, R.L., and Guyenet, P.G. (1999). Distribution of glutamic acid decarboxylase mRNA-containing neurons in rat medulla projecting to thoracic spinal cord in relation to monoaminergic brainstem neurons. *J Comp Neurol* 407, 367-380.

Stosiek, C., Garaschuk, O., Holthoff, K., and Konnerth, A. (2003). In vivo two-photon calcium imaging of neuronal networks. *Proc Natl Acad Sci U S A* 100, 7319-7324.

Svoboda, K., Denk, W., Kleinfeld, D., and Tank, D.W. (1997). In vivo dendritic calcium dynamics in neocortical pyramidal neurons. *Nature* 385, 161-165.

Tan, J., Savigner, A., Ma, M., and Luo, M. (2010). Odor information processing by the olfactory bulb analyzed in gene-targeted mice. *Neuron* 65, 912-926.

Thek, K.R., Ong, S.J.M., Carter, D.C., Bassi, J.K., Allen, A.M., and McDougall, S.J. (2019). Extensive Inhibitory Gating of Viscerosensory Signals by a Sparse Network of Somatostatin Neurons. *J Neurosci* 39, 8038-8050.

van der Heijden, K., Rauschecker, J.P., de Gelder, B., and Formisano, E. (2019). Cortical mechanisms of spatial hearing. *Nat Rev Neurosci* 20, 609-623.

von Buchholtz, L.J., Ghitani, N., Lam, R.M., Licholai, J.A., Chesler, A.T., and Ryba, N.J.P. (2021). Decoding Cellular Mechanisms for Mechanosensory Discrimination. *Neuron* 109, 285-298 e285.

Wehr, M., and Zador, A.M. (2003). Balanced inhibition underlies tuning and sharpens spike timing in auditory cortex. *Nature* 426, 442-446.

Williams, E.K., Chang, R.B., Strohlic, D.E., Umans, B.D., Lowell, B.B., and Liberles, S.D. (2016). Sensory Neurons that Detect Stretch and Nutrients in the Digestive System. *Cell* 166, 209-221.

Xu, S., Yang, H., Menon, V., Lemire, A.L., Wang, L., Henry, F.E., Turaga, S.C., and Sternson, S.M. (2020). Behavioral state coding by molecularly defined paraventricular hypothalamic cell type ensembles. *Science* 370.

Yarmolinsky, D.A., Peng, Y., Pogorzala, L.A., Rutlin, M., Hoon, M.A., and Zuker, C.S. (2016). Coding and Plasticity in the Mammalian Thermosensory System. *Neuron* 92, 1079-1092.

Zhang, M., Eichhorn, S.W., Zingg, B., Yao, Z., Cotter, K., Zeng, H., Dong, H., and Zhuang, X. (2021). Spatially resolved cell atlas of the mouse primary motor cortex by MERFISH. *Nature* 598, 137-143.

Zhang, X., Fogel, R., and Renshan, W.E. (1995). Relationships between the morphology and function of gastric- and intestine-sensitive neurons in the nucleus of the solitary tract. *J Comp Neurol* 363, 37-52.

Zhou, J., Jia, C., Montesinos-Cartagena, M., Gardner, M.P.H., Zong, W., and Schoenbaum, G. (2021). Evolving schema representations in orbitofrontal ensembles during learning. *Nature* 590, 606-611.

Reviewer Reports on the First Revision:

Referees' comments:

Referee #1 (Remarks to the Author):

Overview statement:

Major concerns: The major concern of this paper remains that the main idea introduced in the abstract, introduction, and throughout the manuscript is not in agreement with the data presented. In fact, the abstract and introduction appear to be for a completely different study altogether. This is a manuscript about GI stretch, not broadly about visceral sensation as is stated in the writing. Without proper citation of previous work in the area, the authors appear to have omitted the entire field on which their data are based. This needs to be rectified.

This article presents a demonstration of NTS response across the entire GI tract using a difficult technique, which will be of interest to many, but the claims made here are overstated. It is acknowledged the challenge of the technique of imaging the NTS, however, this is not a techniques paper and the claims made need to be accordingly stated.

Overstating the study: This manuscript focuses on stretch in the larynx and a portion of the gastrointestinal tract, specifically the stomach, small intestine, and cecum. More precisely, the focus is on larynx, stomach, and duodenum. However, the authors continue to claim this portion represents all the very diverse visceral organs. It is incorrect. This is despite their claim in the rebuttal that the language has been toned down. Examples include:

- Title: This title is an improvement from the original title, but still misleading. A title like "A brainstem map of intestinal stretch sensing" might be a better title, that still maintains the main idea.
- Abstract: Stretch and the gastrointestinal tract are never mentioned even though this is the exact sensory mapping they are testing in this manuscript. Instead, "visceral organs" or "visceral sensing" is repeatedly used, which is misleading as the majority of visceral organs are not tested.
- Introduction: There is no background on stretch sensing in the GI tract, or larynx and no justification on why this is the sensing modality chosen. In addition, there is no introduction of the GI tract, or larynx or justification as to why they are investigating only these organs. This introduction does not introduce the actual data presented.
- Results: The figure titles are very broad and misleading with respect to what is actually shown.
- Discussion: There discussion does not incorporate the findings in Fig. 4 so the reader is left not knowing where Fig. 4 fits in the general idea of the paper. It is argued that "visceral inputs are organized into a brainstem map that takes a shape analogous to a homunculus," but as the visceral inputs are largely constrained GI stretch, this is an overstated conclusion. This is appropriate only as a hypothesis.

Unclear Data/Inappropriate Experimental Design:

- Figure 2 tests high salt, water, citric acid (pH 2.6); these experiments don't seem to fit with figures, 1, 3, and 4. The authors jump from stretch, to adding larynx irritants then back to stretch with some

glucose, and end with stretch. There is no unifying theory or background to support the decision to focus on these visceral areas or stimuli.

- Figure 3: 1M glucose is a very high concentration. Convergence of the NTS response to duodenal glucose and stretch could be an interesting result but there is concern that it may be artificial due to the high glucose concentrations recruiting non-physiological sensory signals from the duodenum. The question becomes why not simply use a lower concentration of glucose, such as 600 mM? This has already been shown to be sufficient to elicit an NTS response. And as the authors note in the rebuttal, 600 mM glucose is close to the concentration of sugar in a coca cola. It is unclear why stimuli parameters were selected that would lead to unclear interpretations of the results. Also given that there is no discussion about what is the receptor sensing osmolarity, or stretch, is it possible that the same channel is sensing both, and therefore it is not surprising that there are neurons that response to intestinal glucose and stretch (line 164-168). This is a major concern with potential to confuse the literature for years.

- Extended Data Fig. 4: The glucose concentration is 1M, and the high salt concentration is 0.5M NaCl, these are very similar osmolarities, meaning that there is no way to determine whether the neurons responding to both are responding to high osmolarity, which is noted in the text. High osmolarity can shrink and expand a cell, so it is not clear if these stimuli are acting as chemical sensory signals or mechanical sensory signals. These are inappropriate experimental parameters leading to confusing and/or inappropriate interpretations.

- It is difficult to interpret the data in Extended Data Fig. 8. There are several reasons that could account for what is seen, without using a system that specifically labeled both sides of the synapse, such as mGRASP, or some sort of tran-synaptic tracing (HSV, rabies, etc.), these data are incomplete. It will not take away from the paper to remove this figure.

Sample sizes:

- More of a comment than a concern: I understand that combining neurons across mice has been the recent way that these data are reported, however, the authors have recorded from an unprecedented number of animals for some of the experiments, giving them the opportunity to really interrogate individual variability. I was hoping that the authors would have taken this opportunity to set a new standard in the field and was disappointed in their rebuttal of the fallback statement "We note that the vast majority of large-scale imaging papers pool together all cells across mice". An example would be figure 4d stated that there is an increase in inhibition when the stomach and duodenum are simultaneously stretched, they claim to have an n=5 for this experiment, therefore, they could determine whether that is a statistically significant decrease, or whether what is being shown is simple the level of noise in the system.

Imprecise language: Qualitative language such as more, less, fewer, the majority, etc. is subjective. In a results section the actual numbers need to be stated with the corresponding n and statistical test if applicable. A few examples are listed here, however, it is of note that this list is not exhaustive:

- Line 103-106: "Responses were observed to each stimulus in the caudal NTS, with more neurons detecting stomach distension, duodenum distension, or laryngeal water, and fewer responding to distension of the oral cavity, jejunum, or cecum (Fig. 1f-g, Extended Data Fig. 3, 4).

- Figure 3b-d: Discussion of these data is all qualitative "easily located", "located apart", "generally separated"

Missing methods:

- Oral balloon distension is not described in methods.

Minor concerns:

- Extended Data Fig. 1b: no scale bars for x- or y-axis
- Extended Data Fig. 1g-h: It seems like the position of the larynx neurons is dependent on the anesthesia used. How was this statistically tested to determine there was no difference?
- Line 70-72: Reference to Extended Data Fig. 1g-h states that these data show that awake imaging is not possible, yet the data is a comparison of two different type of anesthesia.
- Extended Data Fig. 2b: no scale bars for y-axis
- Line 85: What is the definition of a “responding neuron”? How were neurons in Figure 1d selected for further analysis? Line 111-113: What is the “statistical significance” bar for indicating a response?
- Fig. 1g: no scale bars for x- and y-axis
- Extended Data Fig. 4 is out of order.
- The high salt solution in Fig. 2 is 10x PBS while, in Extended Data Fig. 4 it is 0.5M NaCl. There is no comment on why these two stimuli are so different.
- Extended Data Fig. 6 is confusing. Red and green colors are shown to differentiate both Cre+ and Cre- as well as stomach and duodenum. A different color scheme would improve clarity.
- Extended Data Fig. 8a-b: The cartoon mice do not align with the labeled images.
- It would have been nice to see the inhibitory DREADD with the stretch as a complement to the GABA-R block, since the pharmacology can be non-specific.

Referee #2 (Remarks to the Author):

This revised version of the manuscript, the authors have added new data and done an admirable job in addressing the bulk of comments of the reviewers. The additions, modifications and clarifications to the study are extensive and appreciated. Previous concerns about the sensitivity of the new nuclear localized calcium sensor have been addressed and the in-depth evaluation of inhibition adds interesting new depth to the paper. Extending the information on the Cre lines shows the limitations of considering these genetic strains as useful tool for targeting “cell types”. The more in-depth mapping of inputs is also helps a great deal. Overall, the work is significantly improved. However, the paper would still benefit from reorganization and tidying up of the text and figures prior to publication.

The two places I would urge editorial revision is in the organization of the figures and the data presentation. The main figures do not represent the main body of the very results well. Much of the new data is important and emphasized in the text yet relegated to supplementary figures. A second (more critical) issue is the heat maps. Different renderings/color schemes are used and the scales are not consistent. In some cases, the scales are selected in ways as to seem to be filtering out data in order influence readers interpretations. Please make all heat maps the same type and use a constant scale with the widest possible range- 10-100% dF/F is pretty standard.

Here are a few specific suggestions on how to improve the data presentation:

Fig. 1: (c) shows that neurons respond to repeated stimulation and could be moved to a supplementary figure. Generally, heatmaps should have the same style for increased continuity. (d) The data is shown from 30 – 100%. This seems to cut off too much of the data and it also doesn't match similar data presented elsewhere in the paper. I recommend 10-100% for this kind of data which reduces noise but maintains data integrity. (e) Similarly, the same range of dF/F values should be shown in all heatmaps. The figures currently display multiple ranges: 0-40, 0-50 and 0-80. (g) The traces seem poorly resolved and don't add any additional information - consider moving these to a supplementary figure. (h) the correlation is only shown in a range of 0 – 0.7 which skews the color representation and makes it difficult to distinguish correlations in the range of brown.

Fig. 2: (a-c) can be combined with Fig. 1. (d – e) could be moved into the supplement since it is really just showing why everything is correlated in 2c.

The part of the paper that describes that inhibition shapes NTS representation is referring heavily to Ext. figure 8 – 10 and it would be nice to see the most relevant data of these figures as main Figure 4. As it is, I spent an extensive amount of time referring to the supplemental figures to understand this key part of the study.

Referee #3 (Remarks to the Author):

The revised version of the study is substantially extended and refined, giving a completely new basis for solid conclusions. Elegant and challenging in vivo recordings demonstrate fundamental properties of visceral organ sensing in the NTS and will form basis for even more intricate investigations of circuits and molecules regulating interoception. A highlight of the study is the clear identification of spatially segregated response zones (and cells) to different gut regions (but not necessarily stimuli) that cannot be fully explained by molecular patterning nor vagal innervation pathways. The study furthermore opens the door to further investigations of the role of inhibition and cross-inhibition in fine-tuning response patterns. Only a few issues need further clarification.

Remaining comments on previous concerns:

- 1) The rebuttal answer along with the adjustments in the manuscript were satisfactory. Thank you for the extended and precise explanation.
- 2) The re-analysis of bouton-response correlation is impressive, I have no further comments or suggestions.
- 3) It is true that responses from both stomach and intestine were obtained from all types of neurons. Agreeable there is not a one-to-one correlation between marker-labelled neuron and response. However, closing the wealth of information obtained with only this conclusion does not do the data nor the understanding of the functional anatomy of the NTS justice. Based on data presented in Extended data 5, from chance one would expect roughly 20% of the cells to respond to duodenum. Extended Data Figure 6 indicate that the number of responding cells deviates significantly from this in some Cre lines. For instance, it seems like Penk+ cells are greatly over-represented from duodenal stretch, while Th+ neurons are underrepresented. Would it be possible

to determine the deviance from the percentage expected from pure chance, in relation to the observed proportions (for instance the Cre+ and Cre- could be merged as a normalized reference value of the proportions of neurons responding to either stomach or duodenal stimuli)? Based on such data, the hypothesis that NTS response is correlated to cell types/ molecularly patterned may also found partial grounds. Please test such hypothesis for each of the lines in the experiment. Moreover, Extended Data Figure 6b indicate that Cre+ cells are spatially patterned, at least some neuron types seem to be distributed far from a “salt and pepper” fashion (Calcr, Crhr2 and Penk for instance). It is also not clear what cells are labelled with Crhr2 and Pdyn as their expression is not shown in scRNA-seq clustering data. Finally, it is also an overstatement to claim that each Cre-defined cell type is recruited across multiple sensory representations, when only two conditions have been tested and compared against each other.

4-5) My questions have been answered satisfactory

6)The pictures are easier to interpret now.

Additional question:

Extended Data Fig 8l: Could you explain the reasoning behind the comparison of bouton-bouton, neuron-neuron and the neuron-boutons distances?

Minor issue.

Extended Data Fig 8:

The depictions in A appear to be scrambled in relation to b?

Label for the boutons appear twice in ©.

Author Rebuttals to First Revision:

Referees' comments:

Referee #1 (Remarks to the Author):

Overview statement:

Major concerns: The major concern of this paper remains that the main idea introduced in the abstract, introduction, and throughout the manuscript is not in agreement with the data presented. In fact, the abstract and introduction appear to be for a completely different study altogether. This is a manuscript about GI stretch, not broadly about visceral sensation as is stated in the writing. Without proper citation of previous work in the area, the authors appear to have omitted the entire field on which their data are based. This needs to be rectified.

This article presents a demonstration of NTS response across the entire GI tract using a difficult technique, which will be of interest to many, but the claims made here are overstated. It is acknowledged the challenge of the technique of imaging the NTS, however, this is not a techniques paper and the claims made need to be accordingly stated.

Overstating the study: This manuscript focuses on stretch in the larynx and a portion of the gastrointestinal tract, specifically the stomach, small intestine, and cecum. More precisely, the focus is on larynx, stomach, and duodenum. However, the authors continue to claim this portion represents all the very diverse visceral organs. It is incorrect. This is despite their claim in the rebuttal that the language has been toned down. Examples include:

- Title: This title is an improvement from the original title, but still misleading. A title like "A brainstem map of intestinal stretch sensing" might be a better title, that still maintains the main idea.

We are surprised by the tone of the reviewer; her/his suggestion that our findings are limited to 'intestinal stretch sensing' is neither fair nor accurate, as this is only one of ten stimuli tested across organs in the manuscript. We consider our systematic probing of multiple visceral organs (in single mice) to be rather more comprehensive than prior studies and a strength of the manuscript. For comparison, groundbreaking papers on coding in the olfactory system did not look at every possible odor, but instead derived general principles from a practical stimulus set. We never claim or imply analysis of all the very diverse visceral organs in the body, as that would be impossible to achieve. We also note that other reviewers appeared enthusiastic about the broad framework and interpretations of the paper. Reviewer 2 said that we "provide convincing data for regionalization within the NTS and evidence for the existence of organotypic map" in the first review, and reviewer 3 here said this study demonstrates "fundamental properties of visceral organ sensing". We would prefer not changing the title, but would reluctantly consider an alternative title in the spirit of working with the reviewer, "A brainstem map for visceral sensations from the gut and upper airways," if this is deemed a requirement.

- Abstract: Stretch and the gastrointestinal tract are never mentioned even though this is the exact

sensory mapping they are testing in this manuscript. Instead, “visceral organs” or “visceral sensing” is repeatedly used, which is misleading as the majority of visceral organs are not tested.

We added a statement in the abstract that we focused on gut and upper airway stimuli and also clarified in the abstract that we used mechanosensory and chemosensory inputs. In a few cases in the text, we followed the reviewer's suggestion and clarified that visceral inputs analyzed are from the gut and larynx; we do not think remaining usage of the word visceral implies that all or a majority of visceral organs were tested.

- Introduction: There is no background on stretch sensing in the GI tract, or larynx and no justification on why this is the sensing modality chosen. In addition, there is no introduction of the GI tract, or larynx or justification as to why they are investigating only these organs. This introduction does not introduce the actual data presented.

We added several sentences introducing gastrointestinal and laryngeal stimuli in the introduction according to the reviewer's suggestion.

- Results: The figure titles are very broad and misleading with respect to what is actually shown.

We changed the titles to Figures 1, 2, 4, and Extended Data Figures 3, 7, 9, and 10.

- Discussion: There discussion does not incorporate the findings in Fig. 4 so the reader is left not knowing where Fig. 4 fits in the general idea of the paper. It is argued that “visceral inputs are organized into a brainstem map that takes a shape analogous to a homunculus,” but as the visceral inputs are largely constrained GI stretch, this is an overstated conclusion. This is appropriate only as a hypothesis.

We add several sentences to discuss the role of inhibition (now Figures 3 and 4), and qualified the sentence about the homunculus to state that this refers to inputs from the gastrointestinal tract and upper airways.

Unclear Data/Inappropriate Experimental Design:

- Figure 2 tests high salt, water, citric acid (pH 2.6); these experiments don't seem to fit with figures, 1, 3, and 4. The authors jump from stretch, to adding larynx irritants then back to stretch with some glucose, and end with stretch. There is no unifying theory or background to support the decision to focus on these visceral areas or stimuli.

As the reviewer points out, there is an enormous array of interoceptive stimuli to consider that vary by location and modality. As clearly stated in the manuscript, our unifying approach was to first isolate contributions from location and modality. The prior Figure 1 kept modality constant when possible while varying stimulus location. The prior Figure 2 (and associated Extended Data) kept location constant while varying modality, and we did this in two organ systems, the larynx and intestine. Then, in subsequent figures we analyzed the location of responsive neurons, revealing a map, and then the role of inhibition in shaping responses. At the suggestion of reviewer 2, we have now combined prior Figures 1 and 2 in the same figure, and this should help simplify the logical progression of the paper.

- Figure 3: 1M glucose is a very high concentration. Convergence of the NTS response to duodenal glucose and stretch could be an interesting result but there is concern that it may be artificial due to the high glucose concentrations recruiting non-physiological sensory signals from the duodenum. The question becomes why not simply use a lower concentration of glucose, such as 600 mM? This has already been shown to be sufficient to elicit an NTS response. And as the authors note in the rebuttal, 600 mM glucose is close to the concentration of sugar in a coca cola. It is unclear why stimuli parameters were selected that would lead to unclear interpretations of the results. Also given that there is no discussion about what is the receptor sensing osmolarity, or stretch, is it possible that the same channel is sensing both, and therefore it is not surprising that there are neurons that response to intestinal glucose and stretch (line 164-168). This is a major concern with potential to confuse the literature for years.

We add a statement in the text that clarifies choice of stimulus concentration. Prior studies from our lab and other labs have clearly shown that 1M glucose and intestinal stretch activate non-overlapping populations of vagal sensory neurons (Zhao et al., *Nature*, 2022; Williams et al., *Cell*, 2016). We also previously did a careful titration analysis, and showed that the same chemosensory neurons (marked by GPR65) respond to both 300 mM and 1M glucose (Williams et al., *Cell*, 2016). We note that 1M-1.5 M glucose is a standard concentration used in numerous intestinal nutrient sensing and feeding studies (Zhao et al., *Nature*, 2022; Goldstein et al., *Cell Metab.*, 2021; Bai et al., *Cell*, 2019; Su et al., *Cell Metab.*, 2017; Williams et al., *Cell*, 2016). Mechanosensory and chemosensory inputs from the gastrointestinal tract clearly activate different vagal sensory neurons but the same NTS neurons, indicating high-level convergence in the brainstem rather than that the same peripheral channels are sensing both signals.

- Extended Data Fig. 4: The glucose concentration is 1M, and the high salt concentration is 0.5M NaCl, these are very similar osmolarities, meaning that there is no way to determine whether the neurons responding to both are responding to high osmolarity, which is noted in the text. High osmolarity can shrink and expand a cell, so it is not clear if these stimuli are acting as chemical sensory signals or mechanical sensory signals. These are inappropriate experimental parameters leading to confusing and/or inappropriate interpretations.

0.5M NaCl was precisely chosen because it has an identical osmolarity to 1M glucose and can properly serve as a control to distinguish neurons responsive to osmolarity or glucose alone. We stated in the text: "intestinal glucose may activate both a dedicated response pathway as well as neurons that are more broadly tuned, for example to osmotic stimuli", which we think fairly reflects the data. As referenced above, prior studies from our lab and other labs have shown that neither 1M glucose nor 0.5M NaCl activates vagal mechanosensory neurons that detect intestinal stretch (Zhao et al., Nature, 2022; Ichiki et al., Nature, 2022; Williams et al., Cell, 2016), so osmolarity-based activation of stretch-sensitive neurons is not a concern.

- It is difficult to interpret the data in Extended Data Fig. 8. There are several reasons that could account for what is seen, without using a system that specifically labeled both sides of the synapse, such as mGRASP, or some sort of tran-synaptic tracing (HSV, rabies, etc.), these data are incomplete. It will not take away from the paper to remove this figure.

To date, the principal insights into NTS coding have relied on the organization of arriving vagal axons. Thus, the goal of this figure was only to compare imaging responses with historical data involving vagal sensory axons alone, and was not about connectivity. Other reviewers found these studies interesting, and the data were substantially extended in response to previous comments from Reviewer 3. The conclusion from this figure was that NTS representations required additional brainstem processing (including through local inhibition) beyond what is achieved by vagal axon sorting alone. Experiments proposed are not only beyond the goal of the figure, but are infeasible in the context of imaging, and thus cannot provide a way to link anatomical inputs and NTS response properties.

Sample sizes:

- More of a comment than a concern: I understand that combining neurons across mice has been the recent way that these data are reported, however, the authors have recorded from an unprecedented number of animals for some of the experiments, giving them the opportunity to really interrogate individual variability. I was hoping that the authors would have taken this opportunity to set a new standard in the field and was disappointed in their rebuttal of the fallback statement "We note that the vast majority of large-scale imaging papers pool together all cells across mice". An example would be figure 4d stated that there is an increase in inhibition when the stomach and duodenum are simultaneously stretched, they claim to have an n=5 for this experiment, therefore, they could determine whether that is a statistically significant decrease, or whether what is being shown is simply the level of noise in the system.

We already provided an animal-by-animal analysis for Fig 1 as requested in the prior review, and have now provided a similar analysis of data in Figure 4d as requested (analysis is shown in Extended Data Fig. 10). Responses in data sets are remarkably consistent across animals. As mentioned in the prior rebuttal letter, pooled analysis across animals is standard in the field.

Imprecise language: Qualitative language such as more, less, fewer, the majority, etc. is subjective. In a results section the actual numbers need to be stated with the corresponding n and statistical test if applicable. A few examples are listed here, however, it is of note that this list is not exhaustive:

Use of 'more, less, fewer, majority' is always based on data presented in figures. As detailed below, we added quantitative statements in the results section as needed; in one case, use of the word 'majority' was quantified through a correlation coefficient provided in the text.

- Line 103-106: "Responses were observed to each stimulus in the caudal NTS, with more neurons detecting stomach distension, duodenum distension, or laryngeal water, and fewer responding to distension of the oral cavity, jejunum, or cecum (Fig. 1f-g, Extended Data Fig. 3, 4).

This comment is directly based on data in Fig 1f-g. We added statements in the results section indicating the percentage of responding neurons.

- Figure 3b-d: Discussion of these data is all qualitative "easily located", "located apart", "generally separated"

All statements about neuron positions in Fig 3b-d are rigorously quantified and supported by statistical analyses of a segregation index (as detailed in the text) and pairwise distances between observed responders, as compared to shuffled responders (Extended Data Fig. 7).

The phrase 'easily located' was used once in the following sentence: 'the relative positions of neurons responsive to stomach and duodenum stretch were stereotyped and easily located across animals.' We agree that the phrase 'easily located' relates to the technical difficulty of finding the right NTS location in a given experiment and that it is a non-quantifiable statement. Nevertheless, it is an accurate description of our experimental experience. We would prefer to keep this phrase in the text, but can delete it if deemed necessary.

Missing methods:

- Oral balloon distension is not described in methods.

We added this description in the methods.

Minor concerns:

- Extended Data Fig. 1b: no scale bars for x- or y-axis

We clarified that the x-axis scale bar was the same for 1a and 1b; the Y-axis is normalized and has no absolute scale.

- Extended Data Fig. 1g-h: It seems like the position of the larynx neurons is dependent on the anesthesia used. How was this statistically tested to determine there was no difference?

Using either anesthetic, neurons responsive to laryngeal water are consistently located laterally and rostrally to neurons responsive to stomach stretch. We were careful to avoid making quantitative comparisons of response distributions when different stimuli were applied to different mice, which would be inappropriate. Instead, throughout the manuscript, we only performed direct comparisons of spatial distributions with stimuli applied to the same mice, which is essential to control for animal-to-animal technical variation. For example, while we did our best to keep surgical access to the NTS constant, small changes in the orientation or position of the mouse's head are possible and could change the absolute distance between organ representations. Our findings in Extended Data Fig 1g-i clearly reveal a segregation of laryngeal and gastric inputs under both anesthetics, strongly supporting our conclusions that organ representations in the NTS reflect their physical positions within the body, with more anterior organs represented in more rostralateral area of the NTS.

- Line 70-72: Reference to Extended Data Fig. 1g-h states that these data show that awake imaging is not possible, yet the data is a comparison of two different type of anesthesia.

The callout for Extended Data Fig. 1g-i was moved later in the text.

- Extended Data Fig. 2b: no scale bars for y-axis

The Y-axis is normalized and has no absolute scale.

- Line 85: What is the definition of a “responding neuron”? How were neurons in Figure 1d selected for further analysis? Line 111-113: What is the “statistical significance” bar for indicating a response?

We have a detailed explanation of how responding neurons were classified in the methods, which reads:

Baseline fluorescence (F_0) was determined by averaging jRGECO1a fluorescence over a 24 second period prior to stimulation onset, and the standard deviation of this pre-stimulus baseline period (s_0) was determined. $\Delta F/F$ was calculated as:

$$\frac{\Delta F}{F} = \frac{(F_t - F_0)}{F_0}$$

The response threshold (ϑ) of each neuron was set as $2.5 \times s_0 + F_0$. The neuron was considered to display a positive response to a mechanical stimulus if $\Delta F/F$ during the stimulation period was 1) above ϑ for more than three continuous frames and 2) higher than two times standard deviation (s_0') above the averaged fluorescence intensity (F_0') of the seven frames prior to stimulation onset and ϑ for at least two continuous frames. A neuron was considered to display a positive response to a chemical stimulus if $\Delta F/F$ during the stimulation period plus the 20 frames after stimulus offset was 1) above ϑ for more than four continuous frames and 2) higher than two times standard deviation (s_0'') above the averaged fluorescence intensity (F_0'') of the 25 frames prior to stimulation onset and ϑ for at least two continuous frames. Classification of response types in Fig. 1d (Fig. 1c in the revised manuscript) based on adaptation property uses k-means clustering of normalized responses that is already stated in the main text and figure legend.

- Fig. 1g: no scale bars for x- and y-axis

The scale bar for the X-axis was on the left side of the traces; we since moved the figure to Extended Data and enhanced visibility of the X-axis scale bar. The Y-axis is normalized and has no absolute scale.

- Extended Data Fig. 4 is out of order.

We first cite Extended Data Fig 4 together with Extended Data Fig 3 on page 6, so we think the figures are in order. Please let us know if we are misunderstanding the comment.

- The high salt solution in Fig. 2 is 10x PBS while, in Extended Data Fig. 4 it is 0.5M NaCl. There is no comment on why these two stimuli are so different.

These stimuli are used in different organs, and in one case but not the other is a control. The 0.5M NaCl was chosen to match the osmolarity of 1M glucose in the intestine, while the high salt solution in Fig 2 (10x PBS) is an established stimulus that evokes airway protective responses in the larynx.

- Extended Data Fig. 6 is confusing. Red and green colors are shown to differentiate both Cre+ and Cre- as well as stomach and duodenum. A different color scheme would improve clarity.

We changed the color scheme as requested.

- Extended Data Fig. 8a-b: The cartoon mice do not align with the labeled images.

Thank you for catching this- we changed the images so they match.

- It would have been nice to see the inhibitory DREADD with the stretch as a complement to the GABA-R block, since the pharmacology can be non-specific.

Proof-of-principle experiments to establish use of inhibitory DREADDs in NTS inhibitory neurons did not work. While DREADD expression was achieved in inhibitory neurons, CNO (at a high dose of 5 mg/kg) did not silence the responses of inhibitory neurons to visceral stimulation, indicating that this approach was not efficient enough to be used to silence inhibitory neurons. Bicuculline is widely used in the neuroscience field, and we feel it provides compelling loss-of-function data.

Referee #2 (Remarks to the Author):

This revised version of the manuscript, the authors have added new data and done an admirable job in addressing the bulk of comments of the reviewers. The additions, modifications and clarifications to the study are extensive and appreciated. Previous concerns about the sensitivity of the new nuclear localized calcium sensor have been addressed and the in-depth evaluation of inhibition adds interesting new depth to the paper. Extending the information on the Cre lines shows the limitations of considering these genetic strains as useful tool for targeting “cell types”. The more in-depth mapping of inputs is also helps a great deal. Overall, the work is significantly improved. However, the paper would still benefit from reorganization and tidying up of the text and figures prior to publication.

The two places I would urge editorial revision is in the organization of the figures and the data presentation. The main figures do not represent the main body of the very results well. Much of the new data is important and emphasized in the text yet relegated to supplementary figures. A second (more critical) issue is the heat maps. Different renderings/color schemes are used and the scales are not consistent. In some cases, the scales are selected in ways as to seem to be filtering out data in order influence readers interpretations. Please make all heat maps the same type and use a constant scale with the widest possible range- 10-100% dF/F is pretty standard.

We appreciate the thoughtful and helpful comments of the reviewer, and in response, have made all requested changes as detailed below.

Here are a few specific suggestions on how to improve the data presentation:

Fig. 1: (c) shows that neurons respond to repeated stimulation and could be moved to a supplementary figure.

We moved this panel to Extended Data as suggested.

Generally, heatmaps should have the same style for increased continuity. (d) The data is shown from 30 – 100%. This seems to cut off too much of the data and it also doesn't match similar data presented elsewhere in the paper. I recommend 10-100% for this kind of data which reduces noise but maintains data integrity. (e) Similarly, the same range of dF/F values should be shown in all heatmaps. The figures currently display multiple ranges: 0-40, 0-50 and 0-80.

We have now standardized all heat maps to the same scale. All heat maps reporting $\Delta F/F$ are displayed in a blue background. Based on the observed distribution of jRGECO responses in NTS neurons (Extended Data Fig. 2b), we chose a range from 10-80% $\Delta F/F$, which captures the full range of neuronal responses. In some cases, displaying normalized responses is essential, such as for analysis of adaptation rates (Fig. 1c), the extent of cross-inhibition (Fig. 4), and response selectivity (Fig. 3). Showing normalized responses in standard in the field, as without normalization, neurons with low response amplitudes would be visually overwhelmed by strongly responding neurons. Normalized responses are displayed with an orange/brown hue, and a range of 30-100% was chosen and consistently used throughout the manuscript as it provides a clear visualization of the extent of adaptation and cross-inhibition.

(g) The traces seem poorly resolved and don't add any additional information - consider moving these to a supplementary figure.

We moved this panel to Extended Data as suggested.

(h) the correlation is only shown in a range of 0 – 0.7 which skews the color representation and makes it difficult to distinguish correlations in the range of brown.

We now show the full correlation range of 0 to 1.0 as suggested.

Fig. 2: (a-c) can be combined with Fig. 1. (d – e) could be moved into the supplement since it is really just showing why everything is correlated in 2c.

We made the changes suggested.

The part of the paper that describes that inhibition shapes NTS representation is referring heavily to Ext. figure 8 – 10 and it would be nice to see the most relevant data of these figures as main Figure 4. As it is, I spent an extensive amount of time referring to the supplemental figures to understand this key part of the study.

We moved data from Extended Data to main figures as suggested. Figure 3 now includes studies involving 1) chemogenetic activation of inhibitory neurons and 2) GABA_AR blockade, while Figure 4 focuses on cross-inhibition studies.

Referee #3 (Remarks to the Author):

The revised version of the study is substantially extended and refined, giving a completely new basis for solid conclusions. Elegant and challenging in vivo recordings demonstrate fundamental properties of visceral organ sensing in the NTS and will form basis for even more intricate investigations of circuits and molecules regulating interoception. A highlight of the study is the clear identification of spatially segregated response zones (and cells) to different gut regions (but not necessarily stimuli) that cannot be fully explained by molecular patterning nor vagal innervation pathways. The study furthermore opens the door to further investigations of the role of inhibition and cross-inhibition in fine-tuning response patterns. Only a few issues need further clarification.

Remaining comments on previous concerns:

- 1) The rebuttal answer along with the adjustments in the manuscript were satisfactory. Thank you for the extended and precise explanation.
- 2) The re-analysis of bouton-response correlation is impressive, I have no further comments or suggestions.
- 3) It is true that responses from both stomach and intestine were obtained from all types of neurons. Agreeable there is not a one-to-one correlation between marker-labelled neuron and response. However, closing the wealth of information obtained with only this conclusion does not do the data nor the understanding of the functional anatomy of the NTS justice. Based on data presented in Extended data 5, from chance one would expect roughly 20% of the cells to respond to duodenum. Extended Data Figure 6 indicate that the number of responding cells deviates significantly from this in some Cre lines. For instance, it seems like Penk+ cells are greatly over-represented from duodenal stretch, while Th+ neurons are underrepresented. Would it be possible to determine the deviance from the percentage expected from pure chance, in relation to the observed proportions (for instance the Cre+ and Cre- could be merged as a normalized reference value of the proportions of neurons responding to either stomach or duodenal stimuli)? Based on such data, the hypothesis that NTS response is correlated to cell types/ molecularly patterned may also found partial grounds. Please test such hypothesis for each of the lines in the experiment. Moreover, Extended Data Figure 6b indicate that Cre+ cells are spatially patterned, at least some neuron types seem to be distributed far from a “salt and pepper” fashion (Calcr, Crhr2 and Penk for instance). It is also not clear what cells are labelled with Crhr2 and Pdyn as their expression is not

shown in scRNA-seq clustering data. Finally, it is also an overstatement to claim that each Cre-defined cell type is recruited across multiple sensory representations, when only two conditions have been tested and compared against each other.

4-5) My questions have been answered satisfactory

6)The pictures are easier to interpret now.

Thank you for taking the time to provide helpful comments on our manuscript. We agree with the point raised in comment #3 that some Cre-defined cell types are disproportionately engaged by particular stimuli. As suggested, we now provide a quantification of response enrichment by each Cre-defined cell type (Extended Data Fig. 6d), with use of Cre-negative neurons as an internal reference as suggested. We add a statement in the text which clarifies that different stimuli recruit heterogeneous ensembles of Cre-defined cell types, but that the frequencies of recruited cell types vary across stimuli. We also agree that some cell types display spatial patterning within the NTS, and thus would be differentially recruited by stimuli which engage different NTS regions. In two most obvious cases, neurons labeled in *Crhr2-ires-Cre* mice were enriched medially and responded more frequently to duodenum stretch, while neurons labeled in *Th-Cre* mice were enriched laterally and responded less frequently to duodenum stretch. The coordinate position of each Cre-positive cell type is provided in Extended Data Fig. 6b, and the positional difference of some Cre-defined cell types is now discussed in the text. We removed the word multiple as suggested for precision. We did not detect high level expression of *Crhr2* or *Pdyn* in the single cell data set, even though the Cre driver lines clearly label a subset of NTS neurons. Neurons labeled in *Crhr2-ires-Cre* mice do display enriched responses for intestinal distension. We debated whether to remove data for these lines, but feel that the Cre line reveals the concept of cell type-selective responses and provides a genetic handle for a potentially interesting NTS cell type, even if the marker gene is not well expressed or only expressed during development.

Additional question:

Extended Data Fig 8I: Could you explain the reasoning behind the comparison of bouton-bouton, neuron-neuron and the neuron-boutons distances?

Extended Data Fig. 8I follows your previous suggestion for a 2D correlation analysis between boutons and NTS responders (neurons). If boutons perfectly predict the positions of NTS neurons, boutons should have similar pairwise distances to both NTS neurons and other boutons, like the shuffled data. Contrarily, if boutons and neurons are clustered separately, a bouton would be closer to other boutons than neurons, which is what we observe in this Figure. Thus, while stomach GLP1R boutons are closer to stomach stretch-responsive NTS neurons than stomach GPR65 boutons, vagal axon position alone was still only partially predictive of NTS responses, as indicated by the significantly greater pairwise distance between boutons and neurons (Extended Data Fig. 8).

Minor issue.

Extended Data Fig 8:

The depictions in A appear to be scrambled in relation to b?

Thank you for catching this! We fixed the figure.

Label for the boutons appear twice in ©.

We removed the extra bouton label.

Reviewer Reports on the Second Revision:

Referees' comments:

Referee #1 (Remarks to the Author):

There are two major concerns:

- Claims are overstated. Changes made to the text are satisfactory and the new title offered by the authors in the recent rebuttal seems more appropriate.
- The 1M concentration of nutrient solution used is a flaw for the conclusions drawn.

The use of glucose concentrations at 1M and above have been used by the authors in previous of intestinal studies. However, above 300mM glucose has an osmolarity effect rather than nutrient sensing effect. If this issue is not addressed, misinterpreted results will continue to appear in the literature affecting the field and eventually the authors as well.

The manuscript could be salvaged if the 1M glucose data are removed entirely or replaced using a lower concentration (100-500mM) and amending conclusions. In the rebuttal to previous comments, the authors note that vagal nodose neurons respond to both 300mM and 1M glucose. Why not use the more osmotic-appropriate concentration of 300mM here? In this way, conclusions can be as sound as they should be for a research manuscript of otherwise high potential. Below is offered an explanation for the issue.

Vagal nodose neurons respond to glucose in the duodenum, of which a subset also respond to hypertonic stimuli. Tan et al (Nature 2020) demonstrate that ~25% of glucose-responsive (500mM) vagal nodose neurons also respond to equi-osmotic mannitol solution. Moreover, 1M fructose (equi-osmotic sugar that does not engender preference behavior or activate vagal neurons at lower osmolarities) stimulates vagal neuron responses equivalent to that of a 1M mannitol control. [See Extended Data Figure 8]. This is important to note as 1M fructose, 1M mannitol, and 1M glucose have the same osmolarity.

Recently, Ichiki et al (Nature 2022) showed that 500mM NaCl, which is equivalent osmolarity to 1M glucose, activates vagal nodose neurons when perfused through the duodenum. Moreover, 38.5% of vagal nodose neurons that respond to 300mM glucose also respond to 500mM NaCl (equivalent osmolarity to 1M glucose). [See Figure 2]. These data agree well with those of Tan et al. Although a higher concentration than used in the current manuscript, Bai et al (2019 Cell) demonstrate that 1.33M glucose is indeed hypertonic in the intestine. [See Figure S7].

Solutions of 1 Osm (i.e., 1M glucose or 500 mM NaCl) in the duodenum activate overlapping vagal nodose neurons due to the osmolarity of the solutions. Thus, in the current manuscript, the impact of glucose on NTS neuron response cannot be solely attributed to chemosensation of the duodenum. It is indeed an effect of osmolarity, which changes completely the conclusions and in its current form the manuscript is misleading. If the authors wish to make this conclusion, a lower concentration of glucose must be used.

There is a slight chance that the conclusions made about the convergence of stretch and chemosensation in the NTS are correct. However, without the proper experimentation, there is a larger chance that incorrect conclusions make its way into the literature jeopardizing the field, the journal, and the authors themselves.

Referee #2 (Remarks to the Author):

The authors have fully addressed specific comments and I appreciate their implementing many suggestions as well. This is an important and well executed study.

Referee #3 (Remarks to the Author):

The additional analysis of potential cell type specificity based on Cre-mouse lines in relation to response patterns and locations along with the changed conclusions are totally acceptable in the revised version. My only remaining concern is the analysis made in response to my initial comment on the correlation between sensory neuron boutons and responsive neurons. I still do not understand the reasoning and the parameters used in Extended Data Fig 8l. As I read it, your green bar indicates the pair-wise difference between neurons OR between boutons. Why are neuron-neuron and bouton-bouton information not separated? I can see a vague point in comparing responsive neuron – bouton to bouton-bouton, however, why neuron-neuron distances are baked in with the same bar as bouton-bouton is not clear. Furthermore, what pairwise distances are you including – do you only consider the closest pairs, or do you compare all responsive neurons (or bouton) to all others? To me this part is confusing. Consider to either explain the reasoning and parameters used better or remove 8l. The information provided in 8a-k is already sufficient.

Author Rebuttals to Second Revision:

We thank all referees for the time spent working on the manuscript.

Response to Referee #1:

We removed all data in the manuscript involving 1 M glucose, and repeated experiments with 300 mM glucose. Conclusions are unchanged as these new experiments match our prior observations about overlapping duodenal glucose responses and duodenal stretch responses in the NTS. These findings reveal an interesting convergence between glucose-sensing and stretch-sensing pathways that first arises in the brainstem. Updated figure panels involving 300 mM glucose are now provided in Figure 2 and Extended Data 4.

Response to Referee #3:

We added clarifying statements for Extended Data Fig. 8I in both the text and methods related to reasoning and parameters. This figure panel shows that the positions of vagal axons and responsive NTS soma are not perfectly intermingled. Mathematically, if two groups occupy the same positions, distances between elements from opposing groups (A to B) and similar groups (A to A and B to B) would be identical. The measurement provided is sufficient to rule out perfect overlap, which is the only conclusion made related to these data. We have now edited the text to clarify that the goal of this figure panel was to show that the positions of vagal axons and responsive NTS soma were not perfectly aligned and that segregation was quantified across all responders and boutons rather than closest pairs.